# A Training-Free Sub-quadratic Cost Transformer Model Serving Framework with Hierarchically Pruned Attention

**Heejun Lee,**[*] **Geon Park,**[*] **Youngwan Lee,**[*] **Jaduk Suh,**[*] **Jina Kim**
Graduate School of Artificial Intelligence (AI)
Korea Advanced Institute of Science and Technology (KAIST)
Seoul, South Korea
`{ainl,geon.park,ywlee88,jaduksuh,jinakim}@kaist.ac.kr`

**Wonyong Jeong, Bumsik Kim, Hyemin Lee**
LLMOps Team
DeepAuto.ai
Seoul, South Korea
`{young,liam,hailey}@deepauto.ai`

**Myeongjae Jeon**
Graduate School of AI
POSTECH
Pohang, South Korea
`mj.jeon@postech.ac.kr`

**Sung Ju Hwang**
Graduate School of AI
KAIST, DeepAuto.ai
Seoul, South Korea
`sjhwang@kaist.ac.kr`

## ABSTRACT

In modern large language models (LLMs), increasing the context length is crucial for improving comprehension and coherence in long-context, multi-modal, and retrieval-augmented language generation. While many recent transformer models attempt to extend their context length over a million tokens, they remain impractical due to the quadratic time and space complexities. Although recent works on linear and sparse attention mechanisms can achieve this goal, their real-world applicability is often limited by the need to re-train from scratch and significantly worse performance. In response, we propose a novel approach, Hierarchically Pruned Attention (HiP), which reduces the time complexity of the attention mechanism to $O(T \log T)$ and the space complexity to $O(T)$, where $T$ is the sequence length. We notice a pattern in the attention scores of pretrained LLMs where tokens close together tend to have similar scores, which we call "attention locality". Based on this observation, we utilize a novel tree-search-like algorithm that estimates the top-$k$ key tokens for a given query on the fly, which is mathematically guaranteed to have better performance than random attention pruning. In addition to improving the time complexity of the attention mechanism, we further optimize GPU memory usage by implementing KV cache offloading, which stores only $O(\log T)$ tokens on the GPU while maintaining similar decoding throughput. Experiments on benchmarks show that HiP, with its training-free nature, significantly reduces both prefill and decoding latencies, as well as memory usage, while maintaining high-quality generation with minimal degradation. HiP enables pretrained LLMs to scale up to millions of tokens on commodity GPUs, potentially unlocking long-context LLM applications previously deemed infeasible.

## 1 INTRODUCTION

Large Transformer-based generative language models (LLM) trained on huge datasets have recently demonstrated remarkable abilities in various problem domains, such as natural language understanding (Touvron et al., 2023), code generation (Rozière et al., 2024), and multi-modal question answering (Liu et al., 2023a). This is made possible by the effectiveness of the attention mechanism, which learns $T^2$ pairwise relationships between all tokens in a sequence of $T$ tokens. Despite their success, the quadratic complexity of the attention mechanism makes it increasingly challenging to meet growing resource demands when processing longer sequences.

---

[*]Equal contributors

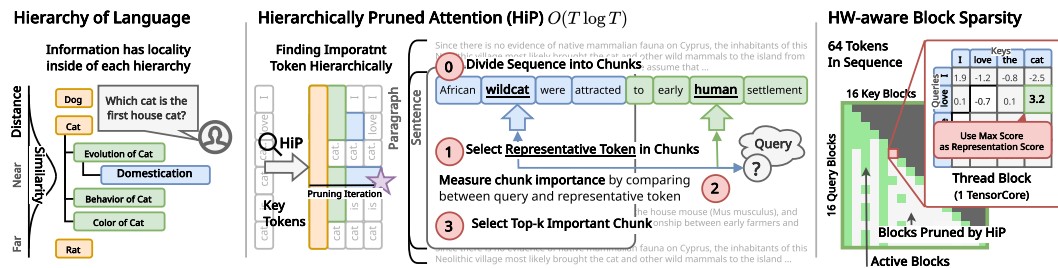

Figure 1: **HiP Attention.** HiP dynamically prunes block sparse attention depending on a given query token in sub-quadratic cost by utilizing the hierarchy and locality of natural language.

Various approaches have been suggested to handle longer sequences efficiently to overcome this limitation. FlashAttention (Dao et al., 2022; Dao, 2023) has reduced the space complexity to $O(T)$ by fusing the component computations to avoid storing $T^2$ attention scores at one time. However, its time complexity remains $O(T^2)$, making it less applicable to inference tasks with long contexts. Many other methods (Lee et al., 2023; Beltagy et al., 2020; Zaheer et al., 2020b; Tay et al., 2020; Kitaev et al., 2019; Tay et al., 2021; Liu et al., 2021) tackle the issue by sparsifying the attention matrix or approximate the attention mechanism using kernel methods to reduce its quadratic complexity. However, these works are not widely employed in real-world LLM serving frameworks because they often lead to performance degradation due to drastic changes in the computation flow and are too complex to implement efficiently for actual speedups. Moreover, they often require extensive fine-tuning or even pre-training from scratch, which can be prohibitively expensive and prevent the timely deployment of production-ready pre-trained models.

In this paper, we define and achieve three fundamental objectives for frameworks tailored to long-context transformer serving frameworks: (1) minimizing the algorithmic complexity of attention mechanisms, (2) enhancing GPU compute efficiency, particularly through TensorCore utilization, and (3) maximizing the effective use of limited GPU memory capacity.

First, to serve long sequence in a timely manner, we propose **Hie**rarchically **P**runed Attention (HiP), an efficient training-free attention mechanism reducing the quadratic time complexity to $O(T \log T)$ by approximating the top-$k$ key tokens in a sequence. HiP exploits "attention locality", where neighboring tokens often have similar attention scores, as shown in Figure 1 (Left). Therefore, as shown in Figure 1 (Center), HiP divides the input sequence into $2k$ chunks, and the center token in each chunk is chosen to represent its neighbors, driven by the attention locality within the chunk. HiP computes the attention scores of these representative tokens to approximate the importance of each chunk for a given query. HiP iteratively refines its selection by starting with the top-$k$ most important chunks and progressively narrowing them down until each chunk contains a single token. This hierarchical top-$k$ key estimation takes $O(T \log T)$ time, which is used for sparse attention computation that costs $O(T)$, making the overall complexity of our attention mechanism log-linear. We provide mathematical proof demonstrating that our HiP outperforms random selection, supported by empirical evidence from attention score statistics in Section 4.

Second, we introduce hardware-aware optimizations to enhance GPU compute efficiency for our HiP through block-wise key sparsity, as illustrated in Figure 1 (Right). Specifically, our top-k approximation is implemented in a tiled manner (Tillet et al., 2019) so that it can fully utilize matrix multiplier units (MMUs; e.g., TensorCores (Nvidia, 2024)) and achieve the highest possible token-processing throughput. Additionally, we integrate our attention mechanism into throughput-optimized LLM serving frameworks, such as vLLM (Kwon et al., 2023) and SGlang (Zheng et al., 2024), further enhancing deployment efficiency.

Lastly, to serve extremely long sequences within the limited GPU memory, we propose a KV cache management strategy that stores only $O(\log T)$ tokens in GPU memory (HBM) and offloads the remaining tokens to host memory (DRAM). The $O(\log T)$ tokens stored in GPU memory are the ones accessed most frequently and are meant to provide quick access for the GPU's MMUs. In contrast, other less frequently accessed tokens reside in main memory and are transferred to GPU memory only upon token access misses. With a high access hit ratio in HiP, our memory management scheme effectively meets the demand for limited HBM capacity while leveraging the larger DRAM capacity, preventing token access from becoming a bottleneck.

We validate HiP on various benchmarks by applying it to Llama3.1-8B (Meta, 2024). In LongBench (Bai et al., 2023), HiP maintains 96% of its relative performance while achieving almost $2.7\times$ speedup in the prefill stage and $16.5\times$ speedup attention computation in the decode stage with 32k context length compared to Flash Attention. Additionally, in passkey retrieval tasks such as

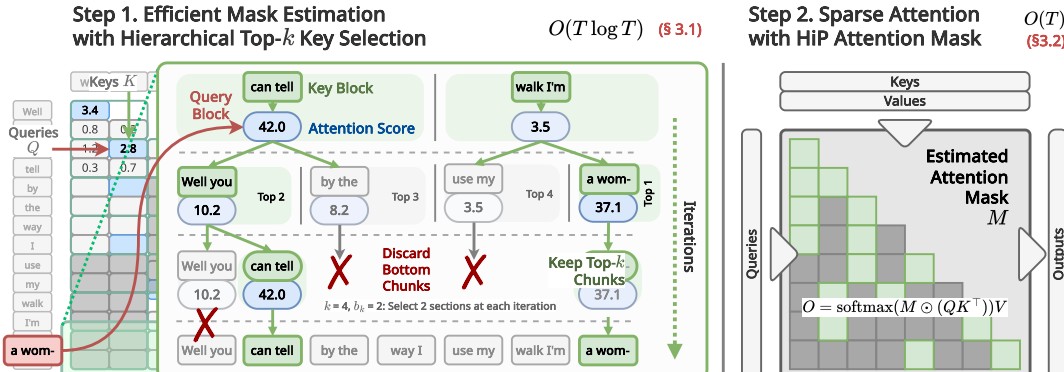

Figure 2: **Overview of our HiP attention mechanism.** In HiP, the model dynamically decides which $k$ number of key tokens to attend to for each query by generating a sparse attention mask. The sparse attention mask is generated in a tree search-like manner. At each iteration, the top-$k$ blocks with the largest attention scores are selected, and the rest of the branches are discarded. The final mask becomes an accurate approximation of the top-$k$ blocks of the true attention map. Please refer to Figure 19 for a more detailed illustration.

RULER (Hsieh et al., 2024), HiP preserves its original effective context length, while all baselines fail to do so. We also evaluate the effectiveness of the proposed KV cache offloading framework. On a machine capable of serving up to a 16k context length with Flash Attention, our method extends the context length up to 64k by offloading the KV cache without significant throughput degradation.

In conclusion, by integrating the three proposed solutions, we present a single long-context serving framework that efficiently manages compute and memory resources while being transparent and easily usable. This extension of serving context length, achieved within the constraints of limited space and compute budgets, delivers substantial benefits for long-context applications, such as question answering with long texts (Kryściński et al., 2022), multi-agent chatbots (Hu et al., 2024), enhanced retrieval-augmented reasoning, and long video data summarization. Furthermore, since our approach is training-free, HiP can be seamlessly applied to pretrained LLMs without requiring additional training. As a result, we expect our method to be highly practical for a wide range of long-context LLM applications.

Our contributions within the proposed framework can be summarized as follows:

- We propose a novel, training-free hierarchically pruned attention mechanism that uses hierarchical score-locality-aware top-$k$ approximation to accelerate LLM serving, reducing the quadratic cost of the attention mechanism to $O(T \log T)$ time and $O(T)$ space complexity (Section 3.1).

- We further optimize our HiP mechanism with a hardware-aware block-wise tiled optimization using OpenAI Triton, achieving up to speed up to $6.83\times$ speedup in end-to-end decoding for 128k context. (Section 3.2, Table 5)

- We implement KV cache offloading to reduce GPU memory efficiency further, increasing serving context from 16k up to 64k tokens in an RTX 4090 with 8B model (Section 3.3).

## 2   RELATED WORKS

Previous studies proposed several attention approximations with linear complexity using either kernel methods or sparse attention. Low-rank approximations of softmax attention via kernel methods (Choromanski et al., 2022; Qin et al., 2022) achieve faster inference speeds but significantly alter the data flow, leading to performance degradation that is hard to mitigate. In contrast, sparse attention methods, which use attention pruning to preserve trained attention scores, allow for simple replacement of pre-trained mechanisms. However, they often require additional fine-tuning to adapt to static attention patterns (Beltagy et al., 2020; Zaheer et al., 2020a; Xiao et al., 2024) or the training of an attention estimator (Lee et al., 2023; Liu et al., 2021). These methods are generally less efficient than fused attention techniques (Dao et al., 2022; Dao, 2023) due to their fine-grained sparsity, which prevents optimal MMU utilization. For more details, see Appendix E.1.

## 3   METHODOLOGY

Given query, key, and value sequences $Q, K, V \in \mathbb{R}^{T \times d}$, the conventional single-head attention output $O$ is computed as $S = QK^\top \in \mathbb{R}^{T \times T}$, $P = \mathrm{softmax}(S) \in \mathbb{R}^{T \times T}$, $O = PV \in \mathbb{R}^{T \times d}$, where $d$ denotes embedding dimension, and softmax is applied row-wise. The causal masking and

constant scaling are omitted for brevity. The $\boldsymbol{S}$ and $\boldsymbol{P}$ matrices are respectively called the *attention scores* and *probabilities*. We focus on the fact that, due to the nature of the softmax function, only the highest attention scores significantly impact the output. Therefore, a promising approach to approximating $\boldsymbol{S}$ in a sparse format and reducing the complexity from $O(T^2)$ is to retain only its top-$k$ elements, as detailed in the following equations:

$$\boldsymbol{M} = \text{top\_}k\text{\_mask}\left(\boldsymbol{QK}^\top\right) \in \{0,1\}^{T \times T}, \tag{1}$$

$$\widehat{\boldsymbol{S}} = \text{mask}_{\boldsymbol{M}}(\boldsymbol{QK}^\top) \in \mathbb{R}^{T \times T}, \quad \widehat{\boldsymbol{P}} = \text{softmax}(\widehat{\boldsymbol{S}}) \in \mathbb{R}^{T \times T}, \quad \widehat{\boldsymbol{O}} = \widehat{\boldsymbol{P}}\boldsymbol{V} \in \mathbb{R}^{T \times d}, \tag{2}$$

$$\text{where } [\text{mask}_{\boldsymbol{M}}(\boldsymbol{S})]_{i,j} := \begin{cases} S_{i,j} & \text{if } M_{i,j} = 1 \\ -\infty & \text{if } M_{i,j} = 0 \end{cases}, \tag{3}$$

where $\text{top\_}k\text{\_mask}(\cdot)$ denotes a binary mask which selects the top-$k$ largest elements for each row of the given matrix. Since $\widehat{\boldsymbol{S}}$ is a sparse matrix with only $kT$ valid elements, $\widehat{\boldsymbol{S}}$ and $\widehat{\boldsymbol{O}}$ in Equation (2) can be computed in $O(T)$ time using sparse matrix operations.

However, obtaining the binary mask $\boldsymbol{M}$ in sub-quadratic time is no easy task. To address this challenging problem, we exploit what we call "attention locality". Observation of attention scores reveal that the scores tend to exhibit local similarity, a phenomenon we refer to as attention locality. We exploit this observation by performing a tree-based search for the top-$k$ tokens. We divide the sequence into $2k$ chunks, and then select a representative token from each chunk. Due to attention locality, a representative token have similar scores to other tokens in its chunk - thereby "representing" that chunk. We select the top-$k$ most important chunks based on the attention scores of the representative tokens. By repeating this process, we refine the tokens until we can no longer divide chunks. Exact details of our method are shown in Section 3.1. We only cover the single-head non-causal case here, but note that our method can easily be extended to causal multi-head attention.

## 3.1 Hierarchical Score-Locality-Aware Top-$k$ Estimation

As shown in Equation (1), our goal is to select the top-$k$ largest elements of each row of pre-trained attention score $S$ without computing the entire matrix. To this end, we use a greedy binary tree search algorithm, as illustrated in the left side of Figure 2. The complete algorithm for mask estimation is presented in Algorithm 1.

For a given query $\boldsymbol{q} \in \mathbb{R}^d$, at the first iteration, we divide the key sequence $\boldsymbol{K} \in \mathbb{R}^{T \times d}$ along the time dimension into $k$ equal-sized chunks $(f_1^{(1)} : l_1^{(1)}), (f_2^{(1)} : l_2^{(1)}), \ldots, (f_k^{(1)} : l_k^{(1)})$, where $f_j^{(1)} = \left\lfloor \frac{(j-1)\cdot T}{k} \right\rfloor + 1$ and $l_j^{(1)} = \left\lfloor \frac{j \cdot T}{k} \right\rfloor$ are the first and last indices of the $j$th chunk, each.[1] The superscripts denote the iteration number. At each iteration $i$, we further divide each of the $k$ chunks into two equal-sized *branches*:

$$\mathcal{B}_{2j-1}^{(i)} = (f_j^{(i)}, m_j^{(i)} - 1), \ \ \mathcal{B}_{2j}^{(i)} = (m_j^{(i)}, l_j^{(i)}), \text{ where } m_j^{(i)} = \left\lfloor (f_j^{(i)} + l_j^{(i)})/2 \right\rceil, \text{ for } j = 1 \mathrel{..} k.$$

A representative key index $r_j^{(i)}$ is the center key token index for each branch $\mathcal{B}_j^{(i)}$. We assume that this representative key represents the entire branch. Thus, among the $2k$ branches, the top $k$ branches whose representative key's scores are the highest are chosen for the next iteration:

$$(f_j^{(i+1)}, l_j^{(i+1)}) := \mathcal{B}_{t_j}^{(i)} \text{ for } j = 1 \mathrel{..} k, \text{ where } \{t_1, \ldots, t_k\} := \underset{j \in [1 \mathrel{..} 2k]}{\text{argtop}_k} \left[ \boldsymbol{q}^\top \boldsymbol{K}_{r_j^{(i)}, :} \right]. \tag{4}$$

We repeat the above iteration $n_{it} := \lceil \log_2 T \rceil$ times, i.e., until the length of each branch all becomes 1. In the end, we obtain a set of indices $\mathcal{I} = \{f_1^{(n_{it})}, \ldots, f_k^{(n_{it})}\}$, which is our estimation of the top-$k$ indices of $\boldsymbol{K}$ which have the largest attention scores with the query $\boldsymbol{q}$. Thus, we obtain $\widehat{m}$, an estimation of a row of the attention mask $\boldsymbol{M}$[2]:

$$\widehat{m} = \text{estimate\_attn\_mask}_k(\boldsymbol{q}, \boldsymbol{K}) := [\mathbb{1}_{\mathcal{I}}(1), \mathbb{1}_{\mathcal{I}}(2), \ldots, \mathbb{1}_{\mathcal{I}}(d)]. \tag{5}$$

In conclusion, this algorithm takes $O(T \log T)$ time in total because the total number of iterations is $\log_2 T$ where each iteration takes constant time $O(k)$, and we do this for each of the $T$ queries.

---

[1] $\lfloor \cdot \rceil$ denotes rounding to the nearest integer.
[2] $\mathbb{1}_{\mathcal{A}}(x)$, where $\mathcal{A}$ is a set, denotes the indicator function: $\mathbb{1}_{\mathcal{A}}(x) = 1$ if $x \in \mathcal{A}$, and otherwise $\mathbb{1}_{\mathcal{A}}(x) = 0$.

## 3.2 BLOCK APPROXIMATION OF TOP-$k$ ESTIMATION

Despite the log-linear complexity, obtaining competitive latency to the state-of-the-art implementations of dense attention on an accelerator (e.g., GPU) is difficult. This is because the matrix multiplier unit (MMU) inside accelerators is optimized for dense attention, where they compute fixed-size blocks of matrix multiplication in a few clock cycles. In contrast, the attention score computation in the top-$k$ estimation of HiP cannot be performed with traditional matrix multiplication because a different key matrix is used to compute the dot product for each query vector. To utilize MMU, we use a technique called *block approximation* during top-$k$ estimation, illustrated in Figure 2 (Right).

In top-$k$ estimation, we replace $\boldsymbol{K} \in \mathbb{R}^{T \times d}$ with its tiled version $\mathbf{K} \in \mathbb{R}^{T/b_k \times b_k \times d}$, and $\boldsymbol{Q}$ with its tiled version $\mathbf{Q} \in \mathbb{R}^{T/b_q \times b_q \times d}$, where $b_k$ and $b_q$ are the size of a key block and a query block. The top-$k$ estimation iterations are done similarly to before, except that the division and branching of the key sequence are done block-wise (using the first dimension of $\mathbf{K}$). Importantly, instead of $k$, $k/b_k$ chunks are maintained at each iteration in order to select $k$ tokens, and the score calculation in Equation (4) is replaced with $\max_{m \in [1:b_q], n \in [1:b_k]} \left( \boldsymbol{q}_{m,:}^{\top} \mathbf{K}_{l_j^{(i)}, n,:} \right)$, where $\boldsymbol{q} \in \mathbb{R}^{b_q \times d}$ is the given query block. While this modification enables HiP to reduce the cost further, we internally sample the blocks with stride $b_{sq}$ in the query dimension and $b_{sk}$ in the key dimension instead of using the full $b_q \times b_k$ block.

As a result of this optimization, the estimated mask $\widehat{\boldsymbol{M}}$ becomes block-sparse. Therefore, each $(b_q/b_{sq}) \times d$-block of the query can be matrix-multiplied with the same $(k/b_{sk}) \times d$ key matrix to obtain $(b_q/b_{sq}) \times (k/b_{sk})$ elements of $\widehat{\boldsymbol{S}}$. Thus, $b_q$ and $b_{sq}$ are critical for the most efficient utilization of the MMU: we can achieve a considerable latency reduction if we set $b_q/b_{sq}$ to a multiple of 16 or 32, as shown in Appendix E.4. While the choice of $b_k$ and $b_{sk}$ is irrelevant to the MMU utilization, it helps reduce the number of top-$k$ estimation iterations.

## 3.3 KV CACHE OFFLOADING

Thanks to our top-$k$ estimation algorithm, HiP only accesses $(k/b_{sk}) \log T$ key states per attention head. Moreover, the algorithm's memory access pattern exhibits strong temporal locality. Using this fact, we can further enhance efficiency by exploiting the memory hierarchy: we offload less frequently accessed key-value (KV) states from the GPU to the main memory. This involves caching frequently accessed KV states (hot tokens) by tracking state access patterns of top-$k$ estimation and sparse attention using the estimated HiP mask.

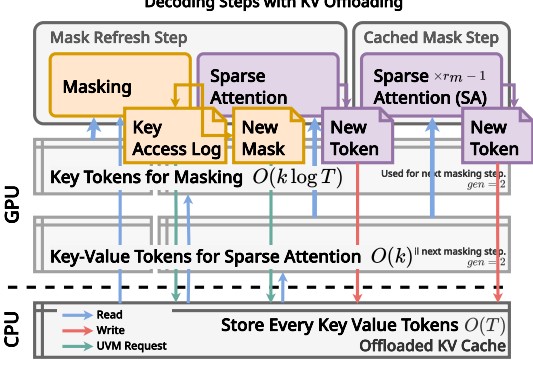

Figure 3: **Flow of KV Cache Offloading with HiP.**

Our GPU cache that holds the hot tokens consists of two components: a *token bank* containing the actual KV states and a *page table* with the token-bank index mapping, as shown in Figure 4. One straightforward implementation for the page table would be a vector map: a simple length-$T$ array of pointers. While this approach is practical for typical sequence lengths (e.g., 128k - 1M), its space complexity is $O(T)$. We employ a linear probing hash table to reduce the space complexity, achieving $O(\log T)$ space complexity. However, empirical results show that GPU hash map lookups introduce additional latency compared to using a simpler vector-based page table.

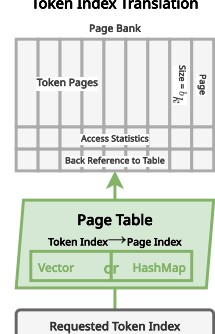

Figure 4: **KV Token Index Translation**

Given the distinct memory access patterns in top-$k$ estimation and in sparse attention, we maintain two separate offloading contexts, each containing a page table and a set of GPU-resident hot tokens, as illustrated as two separate GPU loaded KV caches in Figure 3. For the top-$k$ estimation stage, $k_{\text{cache}} := c \cdot (k/b_{sk}) \log T$ key states are held in VRAM, where $c$ is a hyperparameter determining the cache size. For sparse attention, $k$ key and value states are held. In summary, we need to hold $(k_{\text{cache}}/2 + k)$ tokens' equivalent of KV states in the GPU. The kernel first queries

the GPU cache when accessing key or value tokens. Upon a cache miss (which is unavoidable due to the dynamic nature of the attention access pattern), the system attempts to retrieve tokens from the main memory. By using our cache, we can significantly speed up memory access compared to directly accessing CPU memory from the GPU.

In conclusion, we reduce the GPU memory footprint for KV tokens from $O(T)$ to $O(\log T)$, but this comes with page table overhead that can range between $O(T)$ and $O(\log T)$ depending on the data structure used. The overall space complexity is thus determined by the type of page table, allowing for a configurable trade-off between GPU memory efficiency and latency. However, we suggest that users use vector maps in many practical long-context ranges (32-512k) to achieve competitive latency compared to Flash attention. Please refer to Section 5.5 for detailed benchmarks.

## 4 THEORETICAL ANALYSIS

In this section, we justify the design choices of our HiP's approximate top-$k$ key selection algorithm by answering the following questions: (1) Is HiP's key selection algorithm better than the random selection baseline at finding keys with the biggest scores? (2) How should the representative token in each branch be chosen? We answer these questions by providing a probabilistic analysis of HiP's key selection algorithm in a simplified setting ($k = 1$), based on the assumption of attention locality.

**Observation: keys closer together exhibit higher similarity in attention scores.** In each attention head of a layer in an LLM, a key sequence $K \in \mathbb{R}^{T \times d}$ is used for computing the attention mechanism. Given a query vector $q \in \mathbb{R}^d$, the scores for each key $s = Kq \in \mathbb{R}^T$ can be computed. We investigate how much locality these scores exhibit by studying the correlation between their distance $\Delta := |i - j|$ and the score difference $\delta_\Delta := s_i - s_j$ for every $i, j \in [1..T]$, with a sample natural language data. As shown in Figure 6, our empirical observation shows that $\delta_\Delta$ generally follows a normal distribution, whose mean is almost zero and the standard deviation is an increasing function of distance $\Delta$. More details regarding this observation are provided in Appendix A.3.

**Analysis.** Based on this observation, we assume that we can approximate the difference in attention scores between two keys separated by $\Delta$ tokens as a scalar random variable $\delta_\Delta \sim \mathcal{N}\left(0, \sigma(\Delta)^2\right)$, where $\sigma(\Delta)$ is an increasing function of $\Delta$. This can be interpreted as keys that are closer together are more likely to have a similar attention score, which fits well with our observation and attention locality assumption.

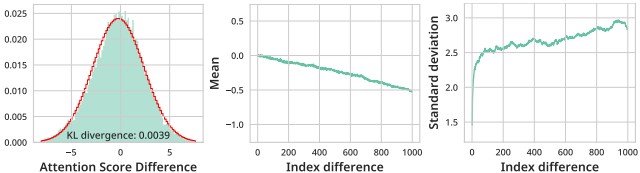

Figure 6: **Score Difference Distribution.** We collect the attention score statistics from the 17th layer and second attention head of Llama3.1-8B. The left figure shows the raw distribution when $\Delta = 500$. The middle and right figures show the mean and standard deviation as a function of $\Delta$.

With this assumption, the following Theorem 1 can be shown.

**Theorem 1** (Informal). *Consider the case of finding the location of the top-1 key token with the maximum attention score in a context of $T$ tokens. Suppose that our locality assumption holds true. We divide the context into two branches with $T/2$ keys each. Then, the branch whose center token has the bigger attention score is more likely to contain the top-1 key token.*

The above shows the effectiveness of one iteration of HiP's key selection algorithm. By recursive application of HiP's key selection iterations, we can intuitively see that the probability of HiP's key selection algorithm finding the location of the top-1 key would be higher than that of uniform random selection as well. Therefore, under the attention locality assumption, on average, HiP's key selection algorithm on average finds the best key tokens more often than random selection. This is also the basis for choosing the center key token as the representative in our algorithm. See Appendix A.1 for the proof sketch and Appendix A.2 for the formal statement and proof of the theorem.

## 5 EXPERIMENTS

### 5.1 EXPERIMENT SETTINGS

Large Language Models (LLMs) are one of the most prominent models that utilize the attention mechanism. Thus, we first apply our proposed HiP to Llama3.1-8B (Touvron et al., 2023), a pre-trained LLM that is reported to perform well on various long-context natural language understanding

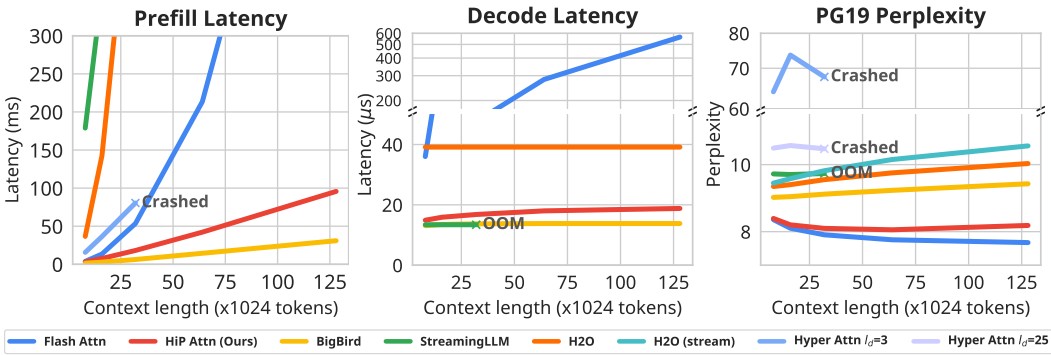

Figure 7: **Latency and Perplexity Evaluation with Various Context Lengths.** We evaluate our proposed HiP and baselines in PG19 (Rae et al., 2019) with various context length on Llama3.1-8B (Meta, 2024). See Appendix D for experiment details.

Table 1: **Passkey Results.** We evaluate our proposed HiP and baselines using passkey retrieval which is a needle in a haystack style context utilization benchmark.

| | | Dense Prefill | | | | | | | | | Sparse Prefill | | | | | | | | | | | | |
| | | Dense | Sparse Decode | | | | | | | | Dense Decode | | | | | | | Sparse Decode | | | | | | |
| | Attention Method | FlashAttn | A 2k | A 4k | BigBird 1k | BigBird 2k | H₂O 256 | H₂O 512 | HiP 512 | HiP 1k | A 2k | AVD 1k | BigBird 1k | BigBird 2k | HiP 512 | HiP 1k | HiP1k +VD | A 2k | AVD 1k | BigBird 1k | BigBird 2k | HiP 512 | HiP 1k | HiP1k +VD |
|---|---|---|---|---|---|---|---|---|---|---|---|---|---|---|---|---|---|---|---|---|---|---|---|---|
| Context Length | 128k | 100 | 62.7 | 64.0 | 64.6 | 67.4 | 80.2 | 87.2 | 79.4 | 82.0 | 11.1 | 97.8 | 8.6 | 10.5 | 27.2 | 27.6 | 92.9 | 7.4 | 9.9 | 6.2 | 9.2 | 19.3 | 32.7 | 68.0 |
| | 64k | 100 | 64.3 | 66.3 | 72.9 | 77.3 | 84.6 | 99.0 | 90.7 | 94.0 | 23.2 | 95.0 | 15.0 | 20.2 | 57.5 | 77.2 | 100 | 18.4 | 12.7 | 16.7 | 16.7 | 50.0 | 68.2 | 80.4 |
| | 32k | 100 | 64.5 | 65.3 | 82.2 | 87.1 | 91.2 | 98.0 | 99.7 | 99.6 | 16.5 | 100 | 26.8 | 42.8 | 96.9 | 100 | 100 | 14.6 | 14.8 | 16.1 | 38.1 | 96.3 | 100 | 95.3 |
| | 16k | 100 | 66.2 | 67.5 | 90.3 | 98.0 | 93.7 | 98.5 | 94.7 | 100 | 25.3 | 98.6 | 43.9 | 59.5 | 100 | 100 | 100 | 19.0 | 13.9 | 29.8 | 55.6 | 98.7 | 100 | 91.3 |
| | 8k | 100 | 69.5 | 73.5 | 98.8 | 99.3 | 95.7 | 99.0 | 100 | 100 | 32.3 | 100 | 61.9 | 98.4 | 100 | 100 | 100 | 30.8 | 16.9 | 66.8 | 95.8 | 100 | 100 | 100 |
| | 4k | 100 | 77.3 | 83.1 | 100 | 100 | 98.6 | 99.7 | 100 | 100 | 56.8 | 100 | 90.4 | 100 | 97.8 | 100 | 100 | 56.8 | 24.7 | 94.1 | 100 | 98.0 | 100 | 100 |
| | Avg. | 100 | 67.4 | 70.0 | 84.8 | 88.2 | 90.7 | **96.9** | 94.1 | 95.9 | 27.5 | 98.6 | 41.1 | 55.2 | 79.9 | 84.1 | **98.8** | 24.5 | 15.5 | 38.3 | 52.6 | 77.0 | 83.5 | **89.2** |
| Speedup (128k) | Prefill | 1.0 | 1.0 | 1.0 | 1.0 | 1.0 | 1.0 | 1.0 | 1.0 | 1.0 | 14.6 | 12.1 | 15.3 | 8.4 | 9.1 | 4.9 | 8.5 | 14.6 | 12.1 | 15.3 | 8.4 | 9.1 | 4.9 | 8.5 |
| | Decode | 1.0 | 29.1 | 14.8 | 21.8 | 11.3 | 29.1 | 14.5 | 29.9 | 15.1 | 1.0 | 1.0 | 1.0 | 1.0 | 1.0 | 1.0 | 1.0 | 29.1 | 45.0 | 21.8 | 11.3 | 29.9 | 15.1 | 63.9 |

tasks up to 128k context tokens, to evaluate the effectiveness of our HiP mechanism. We replace all, but the initial $l_d$ attention layers with HiP in the pretrained LLM, where $L$ is the total number of layers, and $l_d$ denotes the remaining dense attention layers. We choose $l_d$ through an ablation study (Appendix E.5). During LLM decoding, we cache the sparse attention mask from the previous step and refresh it every $r_m$ step to reduce the decoding latency. The latency-performance tradeoff of $r_m$ is discussed in Section 5.4. For a detailed description of HiP's decoding process, see Algorithm 2 in the appendix. Further details on the hyperparameters are in Appendix D.

**Baselines.** We use several sparse attention baselines: A, StreamingLLM (SLLM) (Xiao et al., 2024), AVD (Jiang et al., 2024; Li et al., 2024), BigBird (Zaheer et al., 2020a), HyperAttention (Han et al., 2024), and H₂O (Zhang et al., 2023), chosen for their training-free and sub-quadratic properties. Both StreamingLLM and A use a combination of global sink tokens and sliding window (Beltagy et al., 2020), with StreamingLLM additionally using rolling RoPE indexing (Xiao et al., 2024). AVD retains key vertical and diagonal lines in the prefill attention mask based on snapshot scores on top of A. As it is a prefill-oriented method, A is used for decoding. BigBird uses random masking along with the A pattern. HyperAttention is a token-clustering-style (Kitaev et al., 2019) attention mechanism. Finally, H₂O retains the top-$k$ high-scoring KV tokens for the next step's KV cache.

## 5.2 LANGUAGE MODELING PERFORMANCE EVALUATION

We evaluate HiP on the PG19 (Rae et al., 2019) datasets. We measure latency in two stages: (1) the initial pass (prefill), where the forward pass covers the entire prompt, and (2) subsequent passes (decode), which process one token at a time with a KV cache. In Figure 7, HiP attention is $9.00\times$ faster in prompt latency and $29.99\times$ faster in decoding latency on Llama3.1-8B, with only a +0.5348 increase in perplexity on PG19 ($8.1151 \rightarrow 8.6499$). Our method leverages block approximation to maximize MMU efficiency, outperforming quadratic baselines and achieving near-linear decoding latency. Further details on experimental settings are in Appendix D.

Table 3: **LongBench Results.** We evaluate HiP and baselines. We measure the speedup of prefill and decode on 32k context length, the maximum context length of LongBench.

| | | Dense Prefill | | | | | | | Sparse Prefill | | | | | | | | | | | | | | | | |
| | | Dense | Sparse Decode | | | | | | Dense Decode | | | | | | | | Sparse Decode | | | | | | | | | |
| Attention Method | | FlashAttn | H₂O 256 | H₂O 512 | BigBird 512 | BigBird 1k | HiP 512 | HiP 1k | A 1k | AVD 512 1k+k | AVD 1k 2k+2k | BigBird 512 | BigBird 1k | HiP 512 | HiP 1k | HiP 256 +VD | SLLM 512 | A 1k | AVD 512 1k+k | AVD 1k 2k+2k | BigBird 512 | BigBird 1k | HiP 512 | HiP 1k | HiP HEAL 1k | HiP 256 +VD |
|---|---|---|---|---|---|---|---|---|---|---|---|---|---|---|---|---|---|---|---|---|---|---|---|---|---|---|
| Subset | NarrativeQA | 29.5 | 15.8 | 15.3 | 25.9 | 26.0 | 29.0 | 28.9 | 20.2 | 22.0 | 25.5 | 23.5 | 21.4 | 23.9 | 26.8 | 27.1 | 11.4 | 17.4 | 18.8 | 20.6 | 14.9 | 19.4 | 21.4 | 25.1 | 26.9 | 26.4 |
| | Qasper | 44.3 | 19.4 | 21.6 | 32.3 | 35.6 | 42.1 | 43.2 | 28.6 | 41.4 | 42.9 | 44.3 | 39.8 | 45.1 | 44.2 | 43.1 | 7.8 | 21.4 | 26.2 | 26.3 | 26.9 | 34.9 | 41.7 | 43.6 | 43.9 | 43.0 |
| | HotpotQA | 54.6 | 17.3 | 17.8 | 45.7 | 50.9 | 53.9 | 54.5 | 27.8 | 45.6 | 53.4 | 49.0 | 40.7 | 51.6 | 56.1 | 53.8 | 9.9 | 27.1 | 39.5 | 42.5 | 38.6 | 39.0 | 50.4 | 55.0 | 53.6 | 53.7 |
| | 2WikiMQA | 39.5 | 19.3 | 20.6 | 34.7 | 33.7 | 38.9 | 39.5 | 21.6 | 34.6 | 41.7 | 45.0 | 34.1 | 45.8 | 45.1 | 41.2 | 8.6 | 22.2 | 28.9 | 34.5 | 25.1 | 33.0 | 45.2 | 44.0 | 42.4 | 39.8 |
| | GovReport | 35.0 | 26.7 | 28.2 | 24.9 | 27.3 | 30.3 | 32.4 | 33.8 | 34.2 | 35.0 | 34.5 | 34.0 | 34.6 | 34.4 | 34.6 | 23.2 | 25.9 | 23.2 | 23.1 | 25.2 | 27.3 | 30.1 | 31.2 | 34.9 | 31.7 |
| | MultiNews | 27.4 | 25.1 | 25.6 | 24.1 | 26.4 | 26.5 | 26.8 | 27.1 | 27.2 | 27.4 | 27.1 | 27.0 | 27.1 | 27.1 | 27.2 | 22.6 | 25.5 | 22.8 | 22.5 | 24.9 | 26.1 | 26.0 | 26.7 | 27.9 | 26.7 |
| | Avg. Scores | **38.4** | 20.6 | 21.5 | 31.3 | 33.3 | 36.8 | **37.6** | 26.5 | 34.1 | 37.7 | 37.2 | 32.8 | 38.0 | **38.9** | 37.8 | 13.9 | 23.3 | 26.6 | 28.3 | 25.9 | 29.9 | 35.8 | **37.6** | **38.3** | 36.9 |
| | Rel. Scores (%) | **100** | 53 | 56 | 81 | 87 | 96 | **98** | 69 | 89 | 98 | 97 | 86 | 99 | **101** | 99 | 36 | 61 | 69 | 74 | 68 | 78 | 93 | **98** | **100** | 96 |
| Speedup (32k) | Prefill | 1.0 | 1.0 | 1.0 | 1.0 | 1.0 | 1.0 | 1.0 | **5.6** | **3.2** | **2.1** | **8.5** | **4.6** | **3.0** | **1.7** | **2.7** | **0.1** | **5.6** | **3.2** | **2.1** | **8.5** | **4.6** | **3.0** | **1.7** | **1.7** | **2.7** |
| | Decode | 1.0 | **7.3** | **3.7** | **10.4** | **5.6** | **8.5** | **4.3** | 1.0 | 1.0 | 1.0 | 1.0 | 1.0 | 1.0 | 1.0 | 1.0 | **10.6** | **12.8** | **12.8** | **6.4** | **10.4** | **5.6** | **8.5** | **4.3** | **4.3** | **16.5** |

## 5.3 LONG CONTEXT PERFORMANCE

In this section, we investigate the performance of our HiP, comparing its latency and accuracy against baselines on various benchmarks. Mainly, we build two kinds of benchmark sets: (1) long-context utilization to verify our method can retrieve the information in a given context using a needle in a haystack (NIAH) and (2) long-context natural language understanding to show that our method can preserve reasoning and text generation performance of original long-context LLM. We apply the efficient attention method to mimic various deployment settings by replacing prefill, decode, or prefill-decode flash attention. We can find our HiP performs robustly in every scenario compared to baselines, by applying efficient attention methods in different phases separately.

**Passkey and RULER.** First, we analyze the result of long-context utilization performance using passkey retrieval in Tables 1 and 2. Our passkey retrieval test is a simple test to find a five-digit passkey in a repeated haystack sentence. RULER (Hsieh et al., 2024) is a more complex benchmark containing NIAH tests, such as finding multiple passkeys and tracking variable changes inside complicated essay-style haystack sentences. In Table 1, our method is the strongest in every deployment setting. Dense prefill in general scores high in this benchmark because the model has no chance of overlooking the passkey tokens. However, interestingly, AVD shows an almost perfect score with sparse prefill + dense decode. We think this is because the snapshot

Table 2: **RULER Results.** We compare the effective context lengths of HiP and baselines with Llama3.1-8B. Accuracies surpassing 80% are marked with bold font.

| | | Dense Prefill | Sparse Prefill | | | | | | | | | |
| | | Dense | Sparse | | Dense Decode | | | | Sparse Decode | | | |
| Attention Method | | FlashAttn | BigBird 4k | HiP 2k | BigBird 4k | AVD 2k 4k+k | HiP 2k | HiP 2k +VD | BigBird 4k | AVD 2k 4k+k | HiP 2k | HiP 2k +VD |
|---|---|---|---|---|---|---|---|---|---|---|---|---|
| Effective Length | | 32k | 4k | 32k | 8k | 16k | 32k | 32k | 4k | <4k | 16k | 16k |
| Context Length | 128k | 77.0 | 13.9 | 38.9 | 31.3 | 19.1 | 52.0 | 58.2 | 11.0 | 8.3 | 21.1 | 26.5 |
| | 64k | 84.7 | 15.3 | 68.6 | 41.8 | 66.0 | 73.7 | 79.9 | 11.8 | 12.9 | 53.5 | 63.7 |
| | 32k | **87.4** | 16.9 | **82.9** | 58.1 | 77.0 | **86.5** | **89.7** | 12.5 | 15.9 | 77.5 | **84.2** |
| | 16k | **91.6** | 27.2 | **92.4** | 76.3 | **89.5** | **92.1** | **94.1** | 24.0 | 22.6 | **90.0** | **93.9** |
| | 8k | **93.8** | 54.5 | **94.3** | **89.5** | **94.1** | **94.6** | **94.7** | 46.1 | 38.6 | **94.4** | **94.6** |
| | 4k | **95.5** | **88.3** | **95.9** | **95.3** | **95.9** | **95.9** | **96.0** | **87.1** | 65.1 | **95.7** | **95.9** |
| | Avg. | **88.3** | 36.0 | **78.8** | 65.4 | 73.6 | **82.5** | **85.4** | 32.1 | 27.2 | **72.0** | **72.6** |
| Speedup (128k) | Prefill | 1.00 | 1.00 | 1.00 | **4.53** | **4.80** | **2.44** | **1.70** | **4.53** | **4.80** | **2.44** | **1.70** |
| | Decode | 1.00 | **5.90** | **7.05** | 1.00 | 1.00 | 1.00 | 1.00 | **5.90** | **27.01** | **7.05** | **7.75** |

heuristic that captures important tokens during prefill is a perfect fit for this benchmark. However, because of this aspect, it performs poorly on more complex tasks such as RULER and LongBench. The combination of HiP and AVD slightly increases the performance from regular HiP, achieving 100% accuracy in passkey up to 64k context length.

**LongBench.** We then use the LongBench benchmark (Bai et al., 2023) to evaluate the long context prompt and decoding performance of HiP in Table 3. We believe that this benchmark is the most important because it shows both long context generation performance and knowledge retrieval performance, which are critical in many LLM applications, such as multi-turn assistants and in-context learning. Compared to passkey, the dense decode setting scores higher because this benchmark is much more decoding-heavy. This means that real-world natural language question answering and long context text generation rely more on decoding accuracy rather than prefill. Therefore, we can see non-decode-friendly baselines such as StreamingLLM, AVD and A failing to recover long-generation performance in GovReport and MultiNews subtasks, which decode 512 tokens. Interestingly, AVD completely fails on those two subsets while it works moderately well on some QA tasks. We think this is because AVD fails to capture complex reasoning and long-term context due to its restrictive attention mask patterns. In Appendix E.2, we illustrate this long context knowledge retrieval ability by using an example from LongBench. HiP outperforms every baseline, and

Table 4: **Benchmark Performance on Long-Booksum Task.** We evaluate the book summarization task. 2k tokens are generated for the summary of each book, whose lengths are between 32k-128k tokens. For the 'Half' and 'Quarter window' settings, the context window size of each sparse attention method is adjusted accordingly. The speedups are measured on the Normal setting.

| | Method | KVCache footprint (tokens) | Decoding context window (tokens) | Llama3.1-8B-Instruct | | | | | | | | | Decode Speedup |
| | | | | Normal Window (×1) | | | Half Window (×.5) | | | Quarter Window (×.25) | | | |
| | | | | ROUGE-1 | ROUGE-2 | ROUGE-L | ROUGE-1 | ROUGE-2 | ROUGE-L | ROUGE-1 | ROUGE-2 | ROUGE-L | |
| Un-limited VRAM | FlashAttn | ∞ | ∞ | 41.63% | 10.16% | 23.58% | 41.63% | 10.16% | 23.58% | 41.63% | 10.16% | 23.58% | 1.00x |
| | BigBird 4k | ∞ | 4K | 36.05% | 7.59% | 19.81% | 34.20% | 7.02% | 18.90% | 31.97% | 6.23% | 18.71% | 5.90x |
| Restricted VRAM Size | FlashAttn TRUNC | 8K | 8K | 36.62% | 8.33% | 20.97% | 36.62% | 8.33% | 20.97% | 36.62% | 8.33% | 20.97% | 15.68x |
| | BigBird 2k TRUNC | 8K | 4K | 35.44% | 7.57% | 19.65% | 34.60% | 7.20% | 19.32% | 32.36% | 6.46% | 18.62% | 5.90x |
| | AVD 8k + 8k + 8k | 8K | 8K | 37.18% | 8.28% | 21.62% | 36.07% | 7.72% | 20.60% | 35.72% | 7.67% | 20.81% | 7.51x |
| | HiP 2k (Ours) | 7K | 3K | **38.84%** | **9.11%** | **21.92%** | **38.47%** | **8.57%** | **21.50%** | 37.34% | **8.52%** | 21.53% | 7.75x |
| | HiP 2k + V2k D1k | 7K | ~4K | **39.98%** | **9.61%** | **22.61%** | **39.44%** | **9.09%** | **22.05%** | **38.00%** | **8.70%** | **22.04%** | 7.05x |

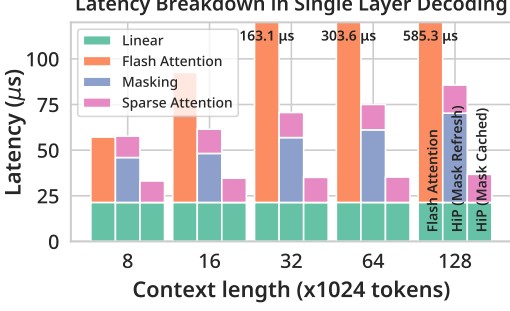

Figure 8: **End-to-end Decoding Latency.** We show two phases of HiP decoding: mask refreshing step triggered in every $r_m$ decoding step and mask cached sparse attention.

Table 5: **End-to-end Decoding Speedup and Quality for each $r_m$.** We show the trade-off between end-to-end decoding speedup and decoding quality using Long-Bench. Metrics are the comparison score to Flash Attention. In LongBench, we merge four tasks into 'QA' and merge two tasks into the 'Summary' column in the table. The latency is measured with $l_d=0$.

| | End-to-End Decoding Speedup | | | | | LongBench | | |
| Seq. Length | 8k | 16k | 32k | 64k | 128k | QA | Summary | Avg. |
| --- | --- | --- | --- | --- | --- | --- | --- | --- |
| HiP ($r_m$=1) | 0.99 | 1.51 | 2.31 | 4.05 | 6.83 | **95.9** | **96.1** | **96.0** |
| HiP ($r_m$=2) | 1.26 | 1.93 | 3.09 | 5.51 | 9.57 | 94.7 | 95.5 | 95.0 |
| HiP ($r_m$=4) | 1.58 | 2.44 | 4.02 | 7.28 | 12.97 | 94.2 | 93.3 | 93.9 |
| HiP ($r_m$=8) | **1.65** | **2.55** | **4.31** | **7.89** | **14.30** | 93.4 | 90.5 | 92.4 |

with a small amount of fine-tuning with an unrelated dataset, it even recovers the original model's performance ('HiP HEAL'). See Appendix D for more details and discussion about healing.

**BookSum.** We use the BookSum benchmark (Kryściński et al., 2022) to assess the long-context and long-response generation capabilities of HiP. We report the average ROUGE F1-scores (Lin, 2004) for the generated summaries in Table 4. To simulate a realistic long-context decoding scenario and demonstrate the effectiveness of KV cache offloading, we put a limit on the GPU KV memory size to 8K tokens. This represents a practical context length on a 24GB GPU with an 8B model without KV offloading. Specifically, for FlashAttention and BigBird, we truncate the context to 8K tokens, and AVD uses an 8K token length sliding window. With our method, with KV cache offloading, we can expand the effective context length only limited by the main memory's capacity, which is much cheaper. HiP outperforms all other baselines in this VRAM-limited setting while maintaining high decoding speed: over 7× faster than regular FlashAttention. Although FlashAttention with a truncated context is faster, it suffers from significant performance degradation and, most importantly, breaks the user's expectation that the model can access the entire context. We observe that HiP with a context window of only 512 still outperforms AVD with an 8k window.

### 5.4 LATENCY BREAKDOWN AND END-TO-END DECODING SPEEDUP

We evaluate the trade-off between attention latency and the model performance with HiP in Figure 7. We observe that our HiP's latency-optimized setting shows about 9.00× speedup of attention decoding latency but only increases the perplexity by 0.5348 in PG19 (Rae et al., 2019), compared to FlashAttention2. In Figure 8, we show the latency breakdown of the HiP-applied transformer model. Our proposed method contains two major components that contribute to the overall latency: (1) top-$k$ estimation iterations and (2) fused sparse attention. We observe that the HiP top-$k$ estimation kernel is the only scaling part as the sequence grows; the sparse attention and linear layer shows constant time for each decoding step. Since the top-$k$ estimation iteration results can be cached and reused $r_m$ times, the latency of the HiP method is dominated by fused sparse attention in most practical scenarios, as shown in Figure 8. On the other hand, the $r_m$ hyperparameter trades off the generation quality for latency, especially for long decoding, as shown in Table 5. HiP achieves 6.83 times end-to-end decoding speedup with 128k context while maintaining 96.0% relative performance in LongBench. We can speed up further to 14.30× when we allow a moderate amount of performance degradation (-3.6%p).

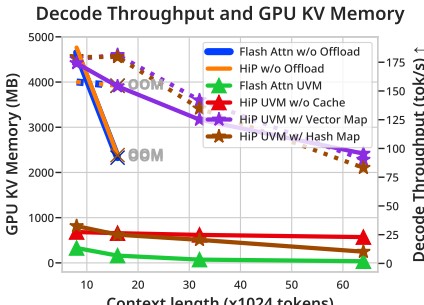

Table 6: Detailed additional data of KV cache offloading performance on RTX 4090 24GB and A100 80GB.

| Throughput (tok/s) | RTX4090 24GB, $T$=64k | | | A100 80GB, $T$=512k | | |
|---|---|---|---|---|---|---|
| | Prefill | Decode | VRAM (MB) | Prefill | Decode | VRAM (MB) |
| FA2 | OOM | OOM | OOM | OOM | OOM | OOM |
| HiP no offload | OOM | OOM | OOM | OOM | OOM | OOM |
| FA2 UVM | 5678 | 1.91 | N/A | 1382 | 0.18 | N/A |
| HiP UVM | 9002 | 22.91 | N/A | 8225 | 20.94 | N/A |
| HiP w/ VectorMap | **6386** | **95.45** | 2283 | **4127** | **85.11** | 18404 |
| HiP w/ HashMap | 359 | 10.15 | **2104** | 48 | 2.74 | **10890** |

Figure 9: **KV Cache Offloading Performance (left).** We measure batched prefill and decoding throughput (tokens/s) with our novel KV cache offloading framework with an on-device offloading cache. Straight lines show the latency, and dashed lines show the GPU memory usage by KV caches.

## 5.5 KV CACHE OFFLOADING BENCHMARK

In Figure 9, we evaluate the latency and memory usage of our KV offloading framework. The UVM variants use the CUDA unified virtual memory API to offload the whole KV cache to the main memory. Our HiP has two variants that depend on the type of cache implementation. We use Llama3.1-8B with 16-bit weights, and the KV states are stored in 8-bit floats. We use a single RTX 4090 24GB for the graph on the left, and to additionally test our method up to 512k tokens, we also test on a single A100 80GB GPU. We set $l_d = 0$, and choose the last token for the representative key to reduce the memory access in this test. See Appendix D for details.

As shown in Figure 9, with UVM, both ours and Flash Attention slow down decoding about 5 to 7 times compared to full GPU runtime. However, we could serve until 64k context, while the same machine can serve only 16k at maximum. Since memory access is significantly more costly with UVM, the trend of logarithmic scaling of decode throughput is clearer than when working with pure GPU memory. So, at 64k context length, ours is more than 50 times faster than Flash Attention with UVM. However, UVM slows down both methods too much compared to full GPU runtime.

We test two types of cache implementation: vector map and hash map. A vector map uses a $T$-sized vector of pointers pointing to the allocated bank to store the mapping between a token index and a bank index. Our GPU-loaded KV offloading cache (Vector Map) shines by achieving 93% decoding throughput compared to no KV offloading at all. Without a significant slowdown, we could extend the serving context from 16k to 64k on an RTX 4090, which is 4.17× higher decoding throughput compared to HiP$_{UVM}$ and 49.97× higher decoding throughput compared to Flash Attention$_{UVM}$, as shown in Table 6. However, with the vector map, the space complexity is $O(T)$. To reduce the space complexity to $O(\log T)$, we use a linear probing hash map to store the index mapping. This way, we can reduce the GPU memory consumption by 40.8% on 512k context length. However, since the hash map lookup is not friendly to the GPU, it slows down token accesses more than naive UVM.

We present our KV offloading framework on a standard gaming PC equipped with a single RTX 4090. Our experiments confirm that the PCIe 4.0x8 bandwidth is sufficient to manage offloading traffic through KV accesses using UVM. Furthermore, when scaled up to a single A100 80GB, our framework demonstrates its ability to extend serving context length, even on server-grade hardware. We anticipate that our HiP's KV offloading framework will effectively increase serviceable context length across a wide range of deployments, from on-device setups to cloud-based environments.

## 6 CONCLUSION

In this study, we present HiP Attention, a novel framework for accelerating pretrained Transformer-based models without any training, with a focus on the acceleration of LLMs for long-context tasks. Our proposed HiP rapidly estimates the top-$k$ context keys for computing sparse attention, drastically reducing the computation required for long context inference and fine-tuning from $O(T^2)$ to $O(T \log T)$. Our HiP attention is a drop-in replacement for the core of any Transformer-based model, such as language and multimodal models, and does not require modifying the existing weights. This is a practical and meaningful improvement as it allows pre-trained models to be fine-tuned and executed much more efficiently in long sequences without sacrificing quality. We are looking forward to contributing to open-source LLM serving frameworks by combining various efficient decoding strategies with HiP attention.

## REPRODUCIBILITY STATEMENT

We provide every experiment code and kernel code in the attached supplementary file. We also provide detailed instructions on how to run experiments in readme markdown files, so please read those files. And we put detailed experiment settings in Appendix D. We will try our best to resolve further reproducibility problems. Inside the HiP library, we have multiple versions of HiP kernels, all written with OpenAI Triton. The upstream kernel path is `hip / models / hip_attention / attention2_draft_prefetch.py`. Additionally, you can see the evolution of our HiP from the very first HiP implementation `hip / models / hip_attention / attention1.py`; please feel free to enjoy our codebases. We left them all for research purposes when someone needs various settings, such as dynamic retention ratios, that are only supported by old versions. Our main experiment entry file is `hip / main / model_eval.py`. Please execute `--help` option to gather further information. Our offloading experiment entry file is `hip / models / hip_attention / offload_runner / offload_runner.py`. For Long-bench and RULER, we modified the official code to run our method with vLLM. Please refer to `HiPAttentionArgs` class to investigate full settings, including every subtle configuration. A, AVD and BigBird are using the same HiP kernel since they are the same block sparse attention. We just modify the block masks that passed to block sparse attention. StreamingLLM is implemented in `hip/models/sink_attention/sink_attention.py`. About HiP-related environment variables of vLLM and SGlang, please refer to `HiPAttentionEnvs` in vLLM and SGlang attention backend implementations.

## ACKNOWLEDGEMENTS

This work was supported by the Institute for Information & communications Technology Planning & Evaluation(IITP) grant funded by the Korea government (MSIT) (RS-2019-II190075, Artificial Intelligence Graduate School Program (KAIST); No.RS-2019-II191906, Artificial Intelligence Graduate School Program (POSTECH); No.RS-2024-00459797, Development of ML compiler framework for on-device AI; No.RS-2022-II220713, Meta-learning Applicable to Real-world Problems), the National Research Foundation of Korea (NRF) grant funded by the Korean government (MSIT) (No. RS-2024-00354947; No. RS-2023-00256259) and the NAVER-Intel Co-Lab. The work was conducted by KAIST and reviewed by both NAVER and Intel. Artificial intelligence industrial convergence cluster development project funded by the Ministry of Science and ICT (MSIT, Korea) & Gwangju Metropolitan City.

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

# Appendices

# A THEORETICAL ANALYSIS

## A.1 PROOF SKETCH

This section provides sketches of the proofs and derivations for the theorems and lemmas mentioned in the main section of the paper. The full formal proofs are presented in the next subsection.

Using the insight obtained from our observations, we model the random variable $\delta_\Delta$ as follows.

**Axiom 1.** *The attention score difference between two tokens distance $\Delta$ apart is defined as a random variable $\delta_\Delta$ with the following distribution.*

$$\delta_\Delta \sim \mathcal{N}(0, \sigma(\Delta)^2), \ \Delta > 0.$$

Where the standard deviation $\sigma(\Delta)$ is an increasing function of $\Delta$, note that both $\delta_\Delta$ and $\sigma(\Delta)$ are defined only when $\Delta > 0$, and cases when $\Delta = 0$ will be dealt separately. We also need to define the problem setting precisely in terms of math. Without losing generality, we model the tokens as a list of indices of length $2n$. For simplicity, we simplify the problem into the case of top-1 selection. The indices of tokens range from 1 to $2n$, where indices 1 to $n$ belong to the first section, and indices $n+1$ to $2n$ belong to the second section. The number $k$ represents the location of the representative token inside each section, and hence, indices $k$ and $n+k$ are the representative tokens of the first and second sections, respectively. We visualize each token position in Figure 10.

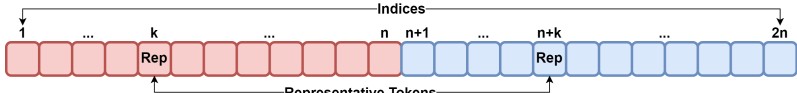

Figure 10: **Visualization of Problem Setting.**

For the ensuing theorems, we define the following random variables and events: t is the random variable regarding the index of the maximum token, and it is assumed to be a uniform random variable. $\mathbb{A}$ and $\mathbb{B}$ are the events where the maximum token is located in the first and second section, respectively. $\mathbb{X}$ is the event where the first representative token is selected, due to the attention score at index $k$ being larger than the score at index $n+k$. The event $\mathbb{Y}$ is the exact opposite of $\mathbb{X}$, where the second representative token is selected.

What is the probability of a token whose distance from the maximum token is $\alpha$, having a greater attention score than a token distance $\beta$ away from the maximum token? Using the random variable $\delta_\Delta$ from earlier, since the tokens cannot have larger scores than the maximum token, the scores can be calculated as $s_{max} - |\delta_\alpha|$ and $s_{max} - |\delta_\beta|$, where $s_{max}$ denotes the maximum attention score. Thus, this problem is equivalent to calculating the probability $\mathbf{P}(|\delta_\alpha| < |\delta_\beta|)$, which can be calculated as follows.

**Lemma 1.**
$$\mathbf{P}(|\delta_\alpha| < |\delta_\beta|) = 4 \int_0^\infty \Phi\left(\frac{\sigma(\beta)}{\sigma(\alpha)}x\right) \cdot \frac{1}{\sqrt{2\pi}} e^{-\frac{x^2}{2}} \, dx - 1.$$

A detailed derivation of the above lemma can be found in Appendix A.2. For convenience, in the proofs to follow, we will abbreviate $\mathbf{P}(|\delta_\alpha| < |\delta_\beta|)$ as $\phi(\sigma(\alpha), \sigma(\beta))$. Please note that $\phi(\sigma(\alpha), \sigma(\beta))$ is a decreasing function of $\alpha$, and an increasing function of $\beta$. Intuitively, this means that the probability of the token with distance $\alpha$ being larger than one with distance $\beta$ gets larger as the first token gets closer and the second token gets further away from the maximum token. This intuition agrees with the assumption of locality.

Using Lemma 1, we can calculate the probabilities for the following four cases — the case of the first or second representative token is selected, either when the maximum token lies in the first or section. The detailed derivation processes are provided in Appendix A.2.

**Claim 1.**
$$\mathbf{P}(\mathbb{X} \cap \mathbb{A}) = \frac{1}{2n}\left(\sum_{i=1}^{k-1} \psi(\sigma(k-i), \sigma(n+k-i)) + 1 + \sum_{i=k+1}^{n} \psi(\sigma(i-k), \sigma(n+k-i))\right),$$

**Claim 2.**

$$\mathbf{P}(\mathbb{Y} \cap \mathbb{A}) = \frac{1}{2n} \left( \sum_{i=1}^{k-1} \psi(\sigma(n+k-i), \sigma(k-i)) + 0 + \sum_{i=k+1}^{n} \psi(\sigma(n+k-i), \sigma(i-k)) \right),$$

**Claim 3.**

$$\mathbf{P}(\mathbb{X} \cap \mathbb{B}) = \frac{1}{2n} \left( \sum_{i=1}^{k-1} \psi(\sigma(n+i-k), \sigma(k-i)) + 0 + \sum_{i=k+1}^{n} \psi(\sigma(n+i-k), \sigma(i-k)) \right),$$

**Claim 4.**

$$\mathbf{P}(\mathbb{Y} \cap \mathbb{B}) = \frac{1}{2n} \left( \sum_{i=1}^{k-1} \psi(\sigma(k-i), \sigma(n+i-k)) + 1 + \sum_{i=k+1}^{n} \psi(\sigma(i-k), \sigma(n+i-k)) \right).$$

From these claims, we now appraise HiP's ability to make the correct choice. In particular, if HiP is indeed better than random token selection, then it should select the first representative token with a higher probability if the maximum token is in the first section, and vice versa. The following theorems specify the conditions in which HiP performs better than random selection. Similarly, more detailed proof can be found in the appendix section.

**Lemma 2.** *If $k \geq n/2$, then $\mathbf{P}(\mathbb{X} \cap \mathbb{A}) > \mathbf{P}(\mathbb{Y} \cap \mathbb{A})$.*

*Proof (sketch).* If $k \geq n/2$, then the following three inequalities hold.

$$\sum_{i=1}^{k-1} \psi(\sigma(k-i), \sigma(n+k-i)) > \sum_{i=1}^{k-1} \psi(\sigma(n+k-i), \sigma(k-i)), \quad 1 > 0,$$

$$\sum_{i=k+1}^{n} \psi(\sigma(i-k), \sigma(n+k-i)) > \sum_{i=k+1}^{n} \psi(\sigma(n+k-i), \sigma(i-k)).$$

Therefore, if $k \geq n/2$, $\mathbf{P}(\mathbb{X} \cap \mathbb{A}) > \mathbf{P}(\mathbb{Y} \cap \mathbb{A})$. $\qquad\square$

**Lemma 3.** *If $k \leq n/2 + 1$, then $\mathbf{P}(\mathbb{Y} \cap \mathbb{B}) > \mathbf{P}(\mathbb{X} \cap \mathbb{B})$.*

*Proof (sketch).* If $k \leq n/2 + 1$, then the three inequalities hold.

$$\sum_{i=1}^{k-1} \psi(\sigma(k-i), \sigma(n+i-k)) > \sum_{i=1}^{k-1} \psi(\sigma(n+i-k), \sigma(k-i)), \quad 1 > 0,$$

$$\sum_{i=k+1}^{n} \psi(\sigma(i-k), \sigma(n+i-k)) > \sum_{i=k+1}^{n} \psi(\sigma(n+i-k), \sigma(i-k)).$$

Therefore, if $k \leq n/2 + 1$, $\mathbf{P}(\mathbb{Y} \cap \mathbb{B}) > \mathbf{P}(\mathbb{X} \cap \mathbb{B})$. $\qquad\square$

The final theorem follows directly from the above two lemmas.

**Theorem 1.** *If $k = n/2$, then $\mathbf{P}(\mathbb{X} \cap \mathbb{A}) > \mathbf{P}(\mathbb{Y} \cap \mathbb{A})$ and $\mathbf{P}(\mathbb{Y} \cap \mathbb{B}) > \mathbf{P}(\mathbb{X} \cap \mathbb{B})$.*

This theorem tell us that if $k = n/2$, then HiP consistently outperforms random token selection in terms of making the correct choice. Thus, just by setting the representative token as the middle token, the hierarchical structure of HiP consistently outperforms random token selection! We now have the answers for the questions mentioned at the beginning of section Section 4. Through mathematical analysis, we have shown that by using the middle token as the representative token, the hierarchical approach of HiP consistently guarantees better performance than random token selection. Therefore, throughout our paper, we always use the middle token as the representative token of a section.

A.2 DETAILED PROOFS

This section provides more detailed proofs and derivations for the theorems and lemmas mentioned in the main section of the paper.

**Lemma 1.**

$$\mathbf{P}(|\delta_\alpha| < |\delta_\beta|) = 4 \int_0^\infty \Phi\left(\frac{\sigma(\beta)}{\sigma(\alpha)}x\right) \cdot \frac{1}{\sqrt{2\pi}} e^{-\frac{x^2}{2}} \, dx - 1.$$

*Proof.*

$$\mathbf{P}(|\delta_\alpha| < |\delta_\beta|) = \mathbf{P}(-|\delta_\beta| < \delta_\alpha < |\delta_\beta|) = \int_{-\infty}^\infty \int_{-|y|}^{|y|} \mathbf{P}(\delta_\alpha = x, \delta_\beta = y) dx dy$$

Assuming that $\delta_\alpha$ and $\delta_\beta$ are independent, we can expand the equation as follows.

$$\mathbf{P}(\delta_\alpha = x, \delta_\beta = y) = \mathbf{P}(\delta_\alpha = x)\mathbf{P}(\delta_\beta = y)$$

$$\therefore \mathbf{P}(|\delta_\alpha| < |\delta_\beta|) = \int_{-\infty}^\infty \int_{-|y|}^{|y|} \mathbf{P}(\delta_\alpha = x)\mathbf{P}(\delta_\beta = y) dx dy$$

From Axiom 1, note that $\delta_\alpha \sim \mathcal{N}(0, \sigma(\alpha)^2)$ and $\delta_\beta \sim \mathcal{N}(0, \sigma(\beta)^2)$.

$$\int_{-\infty}^\infty \int_{-|y|}^{|y|} \mathbf{P}(\delta_\alpha = x)\mathbf{P}(\delta_\beta = y) dx dy$$

$$= \int_{-\infty}^\infty \int_{-|y|}^{|y|} \frac{1}{\sqrt{2\pi\sigma(\alpha)^2}} e^{-\frac{x^2}{2\sigma(\alpha)^2}} \frac{1}{\sqrt{2\pi\sigma(\beta)^2}} e^{-\frac{y^2}{2\sigma(\beta)^2}} \, dx dy$$

$$= \int_{-\infty}^\infty \frac{1}{\sqrt{2\pi\sigma(\beta)^2}} e^{-\frac{y^2}{2\sigma(\beta)^2}} \int_{-|y|}^{|y|} \frac{1}{\sqrt{2\pi\sigma(\alpha)^2}} e^{-\frac{x^2}{2\sigma(\alpha)^2}} \, dx dy$$

$$= \int_{-\infty}^\infty \frac{1}{\sqrt{2\pi\sigma(\beta)^2}} e^{-\frac{y^2}{2\sigma(\beta)^2}} \int_{-|y|/\sigma(\alpha)}^{|y|/\sigma(\alpha)} \frac{1}{\sqrt{2\pi}} e^{-\frac{x^2}{2}} \, dx dy$$

$$= \int_{-\infty}^\infty \left(2\Phi\left(\frac{|y|}{\sigma(\alpha)}\right) - 1\right) \frac{1}{\sqrt{2\pi\sigma(\beta)^2}} e^{-\frac{y^2}{2\sigma(\beta)^2}} \, dy$$

$$= 2 \int_{-\infty}^\infty \Phi\left(\frac{|y|}{\sigma(\alpha)}\right) \frac{1}{\sqrt{2\pi\sigma(\beta)^2}} e^{-\frac{y^2}{2\sigma(\beta)^2}} \, dy - 1$$

Note that the symbol $\Phi$ represents the cumulative distribution function (CDF) of the standard normal distribution.

$$2 \int_{-\infty}^\infty \Phi\left(\frac{|y|}{\sigma(\alpha)}\right) \frac{1}{\sqrt{2\pi\sigma(\beta)^2}} e^{-\frac{y^2}{2\sigma(\beta)^2}} \, dy - 1$$

$$= 4 \int_0^\infty \Phi\left(\frac{y}{\sigma(\alpha)}\right) \frac{1}{\sqrt{2\pi\sigma(\beta)^2}} e^{-\frac{y^2}{2\sigma(\beta)^2}} \, dy - 1$$

$$= 4 \int_0^\infty \Phi\left(\frac{\sigma(\beta)}{\sigma(\alpha)}y\right) \frac{1}{\sqrt{2\pi}} e^{-\frac{y^2}{2}} \, dy - 1$$

$$= 4 \int_0^\infty \Phi\left(\frac{\sigma(\beta)}{\sigma(\alpha)}x\right) \frac{1}{\sqrt{2\pi}} e^{-\frac{x^2}{2}} \, dx - 1$$

$\square$

As mentioned in the main section, for convenience, we will denote $\mathbf{P}(|\delta_\alpha| < |\delta_\beta|)$ as $\psi(\sigma(\alpha), \sigma(\beta))$. Please remember that $\psi(\sigma(\alpha), \sigma(\beta))$ is a decreasing function of $\alpha$, the first argument, and an increasing function of $\beta$, the second argument. Note that in our derivation, we assumed that $\delta_\alpha$ and $\delta_\beta$ are independent. More discussion on this assumption will be provided in the ensuing subsections.

**Claim 1.**

$$\mathbf{P}(\mathbb{X} \cap \mathbb{A}) = \frac{1}{2n} \left( \sum_{i=1}^{k-1} \psi(\sigma(k-i), \sigma(n+k-i)) + 1 + \sum_{i=k+1}^{n} \psi(\sigma(i-k), \sigma(n+k-i)) \right)$$

*Proof.* Recall that t is a uniform random variable denoting the index of the maximum token. Since the number of tokens is $2n$, event $\mathbb{A}$ is actually the union of events $t = 1 \ldots n$. Therefore,

$$\mathbf{P}(\mathbb{X} \cap \mathbb{A}) = \sum_{i=1}^{n} \mathbf{P}(\mathbb{X} \cap t = i) = \sum_{i=1}^{n} \mathbf{P}(t = i)\mathbf{P}(\mathbb{X}|t = i) = \frac{1}{2n} \sum_{i=1}^{n} \mathbf{P}(\mathbb{X}|t = i)$$

Using Lemma 1, $\mathbf{P}(\mathbb{X}|t = i)$ can be calculated as follows.

$$\mathbf{P}(\mathbb{X}|t = i) = \begin{cases} \psi(\sigma(k-i), \sigma(n+k-i)) & \textbf{if } i < k \\ 1 & \textbf{if } i = k \\ \psi(\sigma(i-k), \sigma(n+k-i)) & \textbf{if } i > k \end{cases}$$

A detailed explanation for the above derivation is as follows. When the maximum token index is smaller than the first representative token (i.e. $i < k$), the distances between the maximum token and the two representative tokens are $k - i$ and $n + k - i$. Thus, using Lemma 1, $\mathbf{P}(\mathbb{X}|t = i) = \psi(\sigma(k-i), \sigma(n+k-i))$. Similarly, $\mathbf{P}(\mathbb{X}|t = i) = \psi(\sigma(i-k), \sigma(n+k-i))$ when the maximum token index is larger than the first representative token (i.e. $i > k$). When the maximum token is the first representative token (i.e. $i = k$), we cannot use Lemma 1 as $\sigma(\Delta)$ is defined only when $\Delta > 0$. However, if the maximum token is the first representative token, then it will always be larger than the second representative token - hence, $\mathbf{P}(\mathbb{X}|t = i) = 1$.

Using the above derivation, we can rewrite the original summation as follows.

$$\mathbf{P}(\mathbb{X} \cap \mathbb{A}) = \frac{1}{2n} \sum_{i=1}^{n} \mathbf{P}(\mathbb{X}|t = i)$$

$$= \frac{1}{2n} \left( \sum_{i=1}^{k-1} \psi(\sigma(k-i), \sigma(n+k-i)) + 1 + \sum_{i=k+1}^{n} \psi(\sigma(i-k), \sigma(n+k-i)) \right)$$

$\square$

**Claim 2.**

$$\mathbf{P}(\mathbb{Y} \cap \mathbb{A}) = \frac{1}{2n} \left( \sum_{i=1}^{k-1} \psi(\sigma(n+k-i), \sigma(k-i)) + 0 + \sum_{i=k+1}^{n} \psi(\sigma(n+k-i), \sigma(i-k)) \right)$$

*Proof.* The derivation is almost the same as Claim 1, except that the second representative token needs to have a larger attention score than the first one. Therefore, the arguments in the $\psi$ function are switched for cases $i < k$ and $i > k$. When $i = k$, the maximum token is the first representative token, so $\mathbf{P}(\mathbb{X}|t = i)$ is zero. Therefore,

$$\mathbf{P}(\mathbb{Y}|t = i) = \begin{cases} \psi(\sigma(n+k-i), \sigma(k-i)) & \textbf{if } i < k \\ 0 & \textbf{if } i = k \\ \psi(\sigma(n+k-i), \sigma(i-k)) & \textbf{if } i > k \end{cases}$$

$$\therefore \mathbf{P}(\mathbb{Y} \cap \mathbb{A}) = \frac{1}{2n} \sum_{i=1}^{n} \mathbf{P}(\mathbb{Y}|t = i)$$

$$= \frac{1}{2n} \left( \sum_{i=1}^{k-1} \psi(\sigma(n+k-i), \sigma(k-i)) + 0 + \sum_{i=k+1}^{n} \psi(\sigma(n+k-i), \sigma(i-k)) \right)$$

$\square$

**Claim 3.**

$$\mathbf{P}(\mathbb{X} \cap \mathbb{B}) = \frac{1}{2n} \left( \sum_{i=1}^{k-1} \psi(\sigma(n+i-k), \sigma(k-i)) + 0 + \sum_{i=k+1}^{n} \psi(\sigma(n+i-k), \sigma(i-k)) \right)$$

*Proof.* Recall that t is a uniform random variable denoting the index of the maximum token. Since the number of tokens is $2n$, event $\mathbb{B}$ is actually the union of events $t = n+1 \dots 2n$. Therefore,

$$\mathbf{P}(\mathbb{X} \cap \mathbb{B}) = \sum_{i=n+1}^{2n} \mathbf{P}(\mathbb{X} \cap t = i) = \sum_{i=n+1}^{2n} \mathbf{P}(t = i)\mathbf{P}(\mathbb{X}|t = i) = \frac{1}{2n} \sum_{i=n+1}^{2n} \mathbf{P}(\mathbb{X}|t = i)$$

Similarly to Claim 1, using Lemma 1, $\mathbf{P}(\mathbb{X}|t = i)$ is derived as follows.

$$\mathbf{P}(\mathbb{X}|t = i) = \begin{cases} \psi(\sigma(i-k), \sigma(n+k-i)) & \textbf{if } i < n+k \\ 0 & \textbf{if } i = n+k \\ \psi(\sigma(i-k), \sigma(i-n-k)) & \textbf{if } i > n+k \end{cases}$$

$$\therefore \mathbf{P}(\mathbb{X} \cap \mathbb{B}) = \frac{1}{2n} \sum_{i=n+1}^{2n} \mathbf{P}(\mathbb{X}|t = i)$$

$$= \frac{1}{2n} \left( \sum_{i=n+1}^{n+k-1} \psi(\sigma(i-k), \sigma(n+k-i)) + 0 + \sum_{i=n+k+1}^{2n} \psi(\sigma(i-k), \sigma(i-n-k)) \right)$$

$$= \frac{1}{2n} \left( \sum_{i=1}^{k-1} \psi(\sigma(n+i-k), \sigma(k-i)) + 0 + \sum_{i=k+1}^{n} \psi(\sigma(n+i-k), \sigma(i-k)) \right)$$

$\square$

**Claim 4.**

$$\mathbf{P}(\mathbb{Y} \cap \mathbb{B}) = \frac{1}{2n} \left( \sum_{i=1}^{k-1} \psi(\sigma(k-i), \sigma(n+i-k)) + 1 + \sum_{i=k+1}^{n} \psi(\sigma(i-k), \sigma(n+i-k)) \right)$$

*Proof.* The derivation is similar to Claim 3, except that the second representative token needs to have a larger attention score than the first one. Therefore, the parameters of the $\psi$ function are switched compared to Claim 3. Also, when $i = n+k$, then the second representative token is the maximum token. Therefore, $\mathbf{P}(\mathbb{Y}|t = i)$ is derived as follows.

$$\mathbf{P}(\mathbb{X}|t = i) = \begin{cases} \psi(\sigma(n+k-i), \sigma(i-k)) & \textbf{if } i < n+k \\ 1 & \textbf{if } i = n+k \\ \psi(\sigma(i-n-k), \sigma(i-k)) & \textbf{if } i > n+k \end{cases}$$

$$\therefore \mathbf{P}(\mathbb{Y} \cap \mathbb{B}) = \frac{1}{2n} \sum_{i=n+1}^{2n} \mathbf{P}(\mathbb{Y}|t = i)$$

$$= \frac{1}{2n} \left( \sum_{i=n+1}^{n+k-1} \psi(\sigma(n+k-i), \sigma(i-k)) + 1 + \sum_{i=n+k+1}^{2n} \psi(\sigma(i-n-k), \sigma(i-k)) \right)$$

$$= \frac{1}{2n} \left( \sum_{i=1}^{k-1} \psi(\sigma(k-i), \sigma(n+i-k)) + 1 + \sum_{i=k+1}^{n} \psi(\sigma(i-k), \sigma(n+i-k)) \right)$$

$\square$

**Lemma 2.** *If $k \geq n/2$, then $\mathbf{P}(\mathbb{X} \cap \mathbb{A}) > \mathbf{P}(\mathbb{Y} \cap \mathbb{A})$*

*Proof.* From Claim 1 and Claim 2,

$$\mathbf{P}(\mathbb{X} \cap \mathbb{A}) = \frac{1}{2n} \left( \sum_{i=1}^{k-1} \psi(\sigma(k-i), \sigma(n+k-i)) + 1 + \sum_{i=k+1}^{n} \psi(\sigma(i-k), \sigma(n+k-i)) \right)$$

$$\mathbf{P}(\mathbb{Y} \cap \mathbb{A}) = \frac{1}{2n} \left( \sum_{i=1}^{k-1} \psi(\sigma(n+k-i), \sigma(k-i)) + 0 + \sum_{i=k+1}^{n} \psi(\sigma(n+k-i), \sigma(i-k)) \right)$$

The following inequalities trivially hold.

$$\sum_{i=1}^{k-1} \psi(\sigma(k-i), \sigma(n+k-i)) > \sum_{i=1}^{k-1} \psi(\sigma(n+k-i), \sigma(k-i))$$

$$1 > 0$$

For the remaining two terms, the direction of the inequality depends on the relationship between $\sigma(i-k)$ and $\sigma(n+k-i)$. In order for $\mathbf{P}(\mathbb{X} \cap \mathbb{A}) > \mathbf{P}(\mathbb{Y} \cap \mathbb{A})$, we need to have $\sigma(i-k) \le \sigma(n+k-i)$, i.e. $i - k \le n + k - i$ for all $i = k+1 \cdots n$.

$$i - k \le n + k - i \quad \forall i = k+1 \cdots n$$

$$\therefore 2k \ge 2i - n, \quad \forall i = k+1 \cdots n$$

$$\therefore k \ge i - \frac{n}{2} \quad \forall i = k+1 \cdots n$$

$$\therefore k \ge \frac{n}{2}$$

Therefore, if $k \ge n/2$, then the inequality $\mathbf{P}(\mathbb{X} \cap \mathbb{A}) > \mathbf{P}(\mathbb{Y} \cap \mathbb{A})$ holds. $\qquad \square$

**Lemma 3.** *If $k \le n/2 + 1$, then $\mathbf{P}(\mathbb{Y} \cap \mathbb{B}) > \mathbf{P}(\mathbb{X} \cap \mathbb{B})$*

*Proof.* From Claim 3 and Claim 4,

$$\mathbf{P}(\mathbb{X} \cap \mathbb{B}) = \frac{1}{2n} \left( \sum_{i=1}^{k-1} \psi(\sigma(n+i-k), \sigma(k-i)) + 0 + \sum_{i=k+1}^{n} \psi(\sigma(n+i-k), \sigma(i-k)) \right)$$

$$\mathbf{P}(\mathbb{Y} \cap \mathbb{B}) = \frac{1}{2n} \left( \sum_{i=1}^{k-1} \psi(\sigma(k-i), \sigma(n+i-k)) + 1 + \sum_{i=k+1}^{n} \psi(\sigma(i-k), \sigma(n+i-k)) \right)$$

The following inequalities trivially hold.

$$\sum_{i=k+1}^{n} \psi(\sigma(i-k), \sigma(n+i-k)) > \sum_{i=k+1}^{n} \psi(\sigma(n+i-k), \sigma(i-k))$$

$$1 > 0$$

For the remaining two terms, the direction of the inequality depends on the relationship between $\sigma(n+i-k)$ and $\sigma(k-i)$. In order for $\mathbf{P}(\mathbb{Y} \cap \mathbb{B}) > \mathbf{P}(\mathbb{X} \cap \mathbb{B})$, we need to have $\sigma(k-i) \le \sigma(n+i-k)$, i.e. $k - i \le n + i - k$ for all $i = 1 \cdots k - 1$.

$$k - i \le n + i - k \quad \forall i = 1 \cdots k - 1$$

$$\therefore 2k \le 2i + n, \quad \forall i = 1 \cdots k - 1$$

$$\therefore k \le i + \frac{n}{2} \quad \forall i = 1 \cdots k - 1$$

$$\therefore k \le \frac{n}{2} + 1$$

Therefore, if $k \le n/2 + 1$, then the inequality $\mathbf{P}(\mathbb{Y} \cap \mathbb{B}) > \mathbf{P}(\mathbb{X} \cap \mathbb{B})$ holds. $\qquad \square$

**Theorem 2.** *If $\sigma''(k)\sigma(k) < \sigma'(k)^2$, then $\mathbf{P}(\mathbb{X} \cap \mathbb{A}) + \mathbf{P}(\mathbb{Y} \cap \mathbb{B})$ is maximized when $k = n/2$.*

*Proof.* From Claim 1 and Claim 4,

$$\mathbf{P}(\mathbb{X} \cap \mathbb{A}) + \mathbf{P}(\mathbb{Y} \cap \mathbb{B})$$

$$= \frac{1}{2n}\left(\sum_{i=1}^{k-1}\psi(\sigma(k-i),\sigma(n+k-i)) + 1 + \sum_{i=k+1}^{n}\psi(\sigma(i-k),\sigma(n+k-i))\right)$$

$$+ \frac{1}{2n}\left(\sum_{i=1}^{k-1}\psi(\sigma(k-i),\sigma(n+i-k)) + 1 + \sum_{i=k+1}^{n}\psi(\sigma(i-k),\sigma(n+i-k))\right)$$

$$= \frac{1}{2n}\left(\sum_{i=1}^{k-1}\psi(\sigma(k-i),\sigma(n+k-i)) + 1 + \sum_{i=k+1}^{n}\psi(\sigma(i-k),\sigma(n+i-k))\right)$$

$$+ \frac{1}{2n}\left(\sum_{i=1}^{k-1}\psi(\sigma(k-i),\sigma(n+i-k)) + 1 + \sum_{i=k+1}^{n}\psi(\sigma(i-k),\sigma(n+k-i))\right)$$

For simplicity, we refer to the first term as $T1$, and the second term as $T2$. We now investigate which value of $k$ maximizes $T1$ and $T2$.

For $T2$, suppose the value of $k$ changes from $k$ to $k+1$.

$$T2_{k+1} - T2_k = \psi(\sigma(k),\sigma(n-k)) - \psi(\sigma(n-k),\sigma(k))$$

It can be easily seen that $\psi(\sigma(k),\sigma(n-k))$ is a decreasing function of $k$, and $\psi(\sigma(n-k),\sigma(k))$ is an increasing function of $k$. Therefore, $T2_{k+1} - T2_k$ is a decreasing function of $k$, and its value goes from a positive value when $k = 1$, and a negative value when $k = n-1$. Thus, $T2$ is maximized when $T2_{k+1} - T2_k = \psi(\sigma(k),\sigma(n-k)) - \psi(\sigma(n-k),\sigma(k)) = 0$. The value of $k$ where $\psi(\sigma(k),\sigma(n-k)) - \psi(\sigma(n-k),\sigma(k)) = 0$ is computed as follows.

$$k = n - k, \quad \therefore k = \frac{n}{2}$$

Therefore, $T2$ is maximized when $k = \frac{n}{2}$.

For $T1$, suppose the value of $k$ changes from $k$ to $k+1$.

$$T1_{k+1} - T1_k = \psi(\sigma(k),\sigma(n+k)) - \psi(\sigma(n-k),\sigma(2n-k))$$

Unlike $T2$, we cannot easily determine whether $\psi(\sigma(k),\sigma(n+k))$ or $\psi(\sigma(n-k),\sigma(2n-k))$ is an increasing or decreasing function of $k$. Therefore, we turn to the definition of $\psi(\sigma(\alpha),\sigma(\beta))$. From Lemma 1,

$$\psi(\sigma(\alpha),\sigma(\beta)) = 4\int_0^\infty \Phi\left(\frac{\sigma(\beta)}{\sigma(\alpha)}x\right)\cdot\frac{1}{\sqrt{2\pi}}e^{-\frac{x^2}{2}}\,dx - 1$$

Thus, the positivity of $T1_{k+1} - T1_k$ depends on the characteristics of the function $\sigma(n+k)/\sigma(k)$. If $\sigma(n+k)/\sigma(k)$ is a decreasing function of $k$, then $\psi(\sigma(k),\sigma(n+k))$ becomes a decreasing function of $k$, and inversely, $\psi(\sigma(n-k),\sigma(2n-k))$ becomes an increasing function of $k$. In this case, similarly to $T2$, we can show that $T1$ is maximized when $k = n/2$. In order to find the condition where $\sigma(n+k)/\sigma(k)$ becomes a decreasing function of $k$, we take its derivative in terms of $k$.

$$\frac{d}{dk}\frac{\sigma(n+k)}{\sigma(k)} = \frac{\sigma'(n+k)\sigma(k) - \sigma(n+k)\sigma'(k)}{\sigma(k)^2} < 0$$

$$\therefore \sigma'(n+k)\sigma(k) < \sigma(n+k)\sigma'(k), \quad \therefore \frac{\sigma'(n+k)}{\sigma(n+k)} < \frac{\sigma'(k)}{\sigma(k)}$$

Thus, we see that if $\sigma'(k)/\sigma(k)$ is a decreasing function of $k$, then $\sigma(n+k)/\sigma(k)$ also becomes a decreasing function of $k$. This condition is equivalent to the condition $\sigma''(k)\sigma(k) < \sigma'(k)^2$. The derivation is as follows.

$$\frac{d}{dk}\frac{\sigma'(k)}{\sigma(k)} < 0, \quad \therefore \frac{\sigma''(k)\sigma(k) - \sigma'(k)^2}{\sigma(k)^2} < 0$$

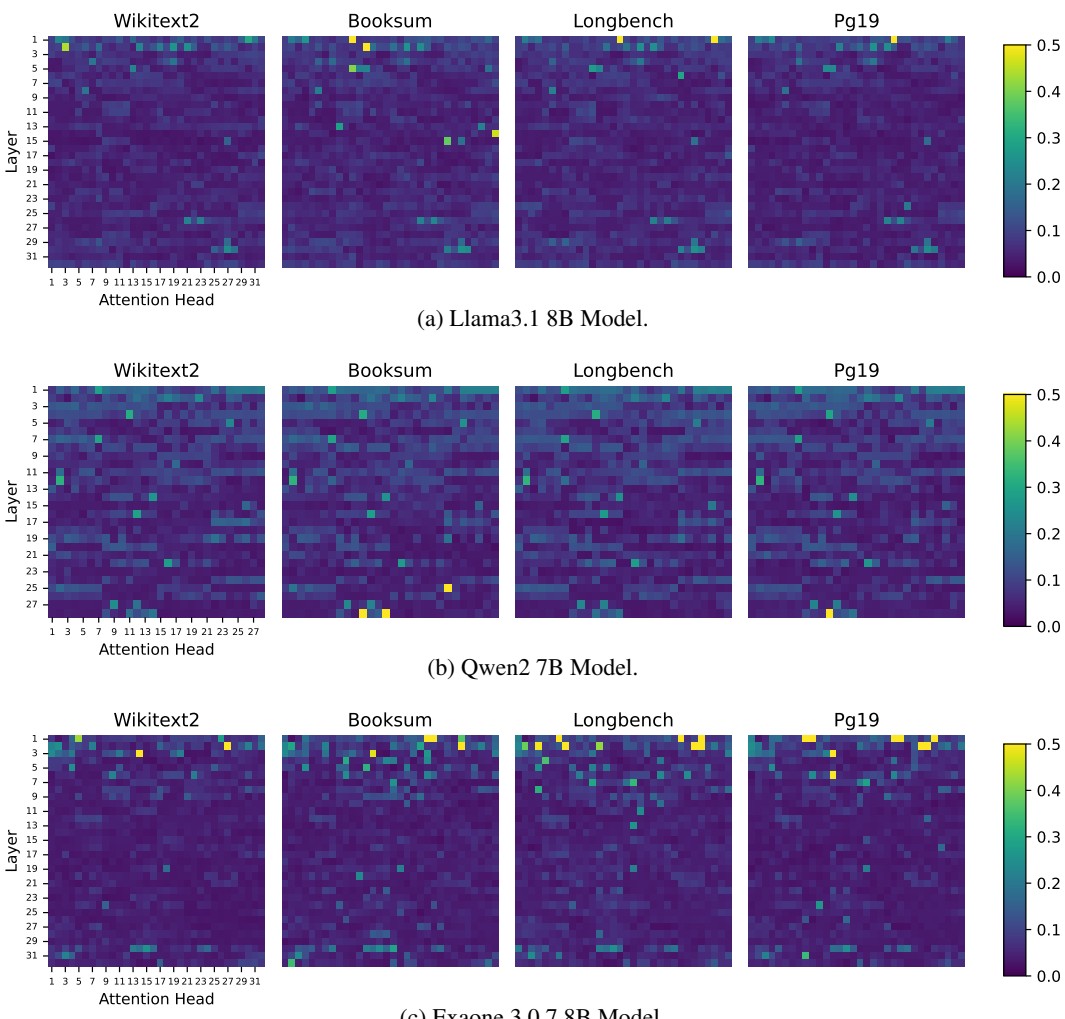

(a) Llama3.1 8B Model.

(b) Qwen2 7B Model.

(c) Exaone 3.0 7.8B Model.

Figure 11: **NRMSE of Fitting $\sigma(\Delta)$ as a Logarithmic Function.** The above figures show the NRMSE (Normalized Root Mean Squared Error) result of fitting the $\sigma(\Delta)$ function as a logarithmic function, as a continuation of Figure 12. For fitting the function, the general formula $y = a\log(x + b) + c$ was used. We demonstrate the experimental results on various datasets and LLM architectures.

$$\therefore \sigma''(k)\sigma(k) < \sigma'(k)^2$$

To summarize, if $\sigma''(k)\sigma(k) < \sigma'(k)^2$, then $\psi(\sigma(k), \sigma(n+k))$ becomes a decreasing function of $k$, and $\psi(\sigma(n-k), \sigma(2n-k))$ becomes an increasing function of $k$. Therefore, $T1_{k+1} - T1_k$ becomes a decreasing function of $k$, and its value goes from positive to negative as $k$ goes from 1 to $n-1$. The crossover point is where $k = n - k$, and $n + k = 2n - k$. Coincidentally, this is when $k = n/2$. Therefore, if $\sigma''(k)\sigma(k) < \sigma'(k)^2$ the value of $k$ which maximizes $T1$ is $k = n/2$.

To summarize, if $\sigma''(k)\sigma(k) < \sigma'(k)^2$, then $k = n/2$ maximizes both $T1$ and $T2$. This concludes the proof. $\qquad\square$

## A.3 REVISITING ASSUMPTIONS IN THEORETICAL ANALYSIS

In Axiom 1 and Lemma 1, we make two important assumptions: that $\delta_\Delta$ follows a normal distribution $\mathcal{N}(0, \sigma(\Delta)^2)$, and that $\delta_\alpha$ and $\delta_\beta$ are independent of each other. However, is this justifiable? In this section, we provide analysis and explanation on this issue by providing several empirical results collected across various datasets and LLM architectures, which justify our assumptions.

### A.3.1 THE DISTRIBUTION OF $\delta_\Delta$

First, we justify the assumption $\delta_\Delta \sim \mathcal{N}(0, \sigma(\Delta)^2)$ by validating three parts - that $\delta_\Delta$ does follow a normal distribution, that $\sigma(\Delta)$ is an increasing function of $\Delta$, and that the mean can be approximated as zero.

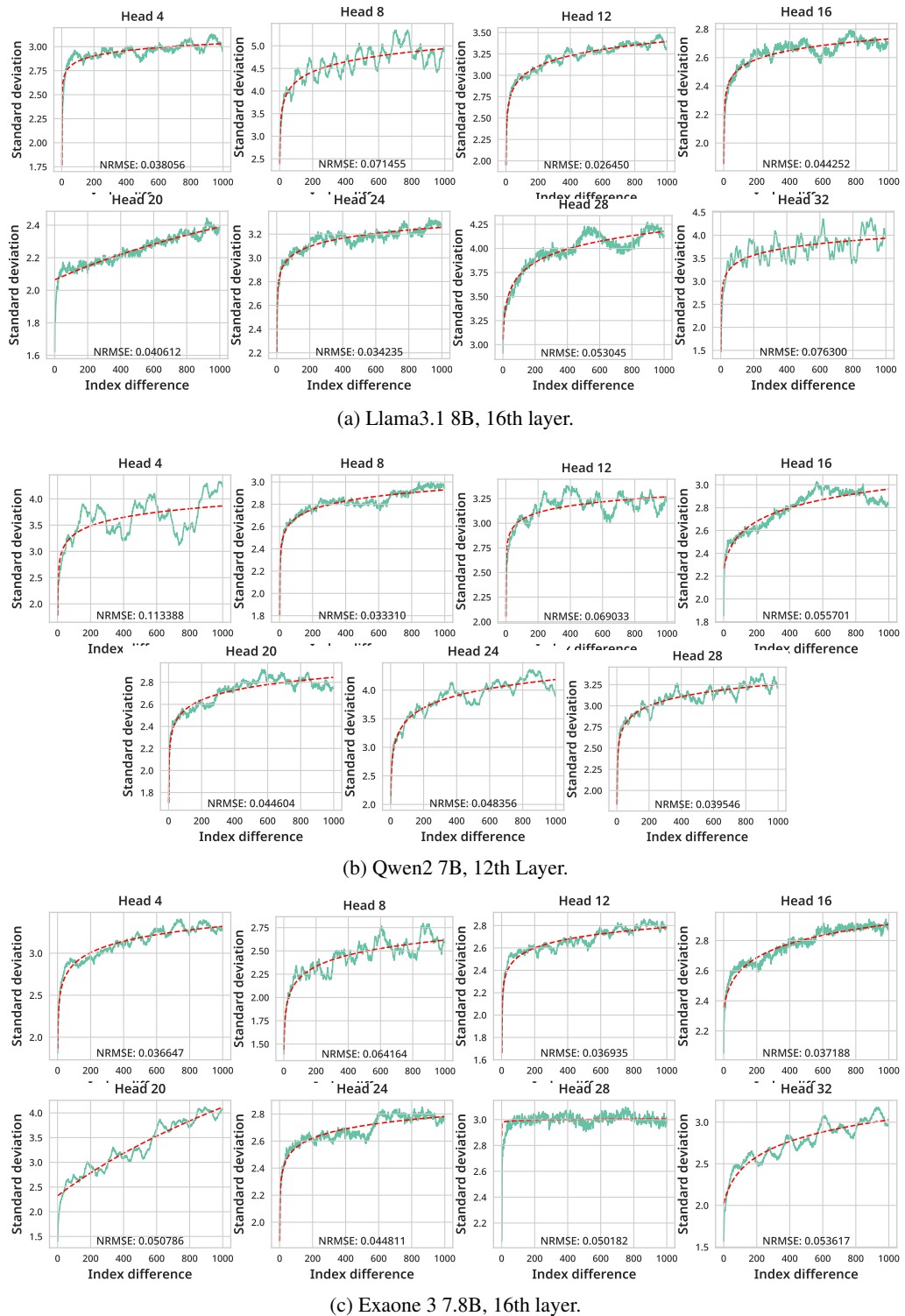

Figure 12: **Plots of $\sigma(\Delta)$ and their Normalized Root Mean Squared Error.** The above figures show the plots of $\sigma(\Delta)$, and the NRMSE errors for fitting as a logarithmic function across various LLM models and layers. For the input, we used the Wikitext2 dataset.

Figure 15 shows some examples of the distribution of $\delta_\Delta$ of various models. As can be seen in the figure, $\delta_\Delta$ clearly follows a normal distribution. In order to provide statistical evidence that $\delta_\Delta$ indeed follows a normal distribution, we compute the KL divergence between $\delta_\Delta$ and the normal distribution fitted onto $\delta_\Delta$. We experiment on all layers for various LLM models and $\Delta$ values, and show some of the results in Figure 16. As can be seen in Figure 16, almost all of the KL divergence values are extremely small and close to zero. In fact, with the small exception of the first layer

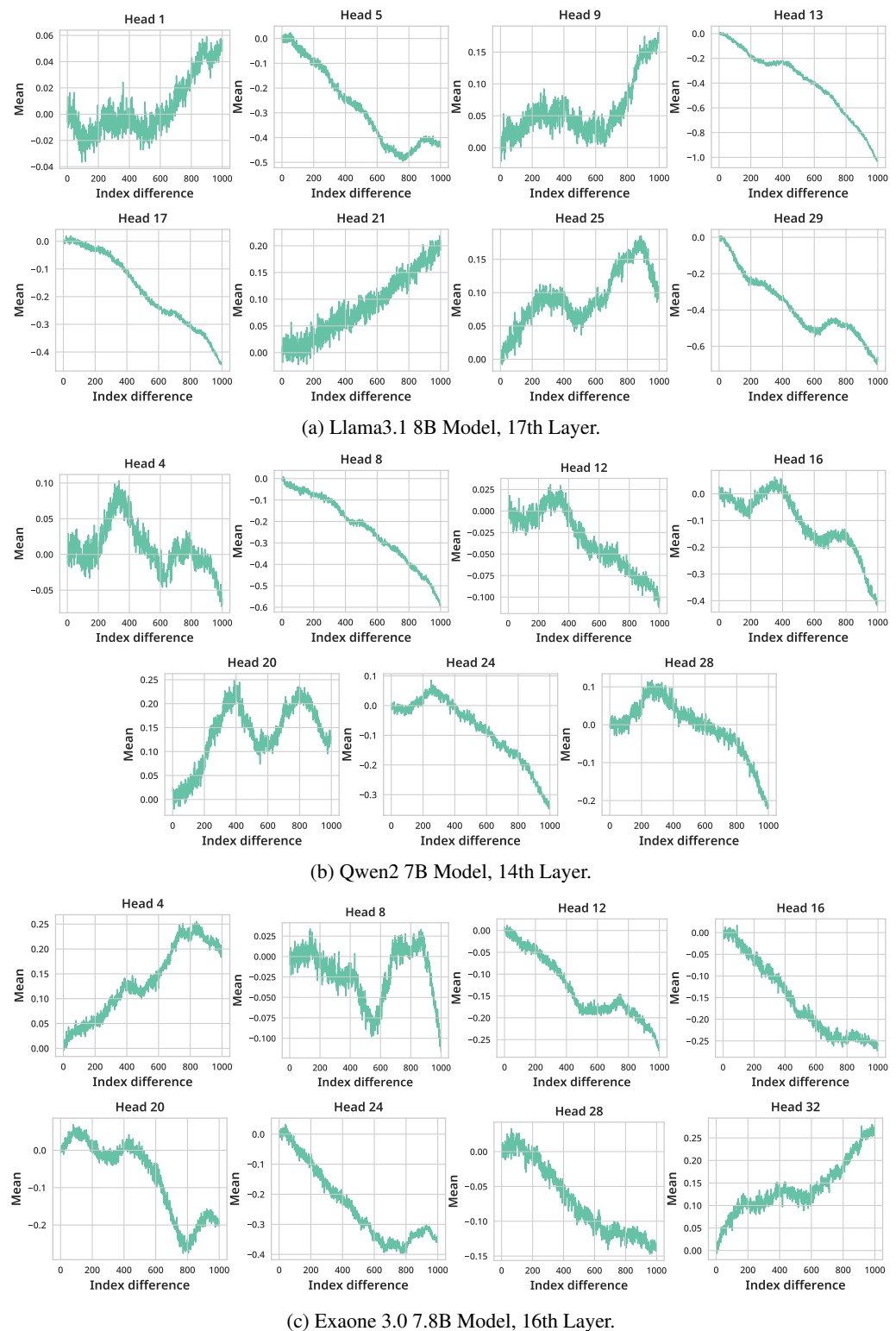

Figure 13: **Plots of the mean of $\delta_\Delta$.** The above figures show the plots of the mean of $\delta_\Delta$, as a function of $\Delta$ for various LLM models and layers. For the input, we used the LongBench dataset.

of the Qwen2 model, all of the KL divergence values are smaller than 0.1, which validates that $\delta_\Delta$ follows the normal distribution.

Although the KL divergence values in Figure 16 are very small, it appears that few attention heads stand out as extreme outliers in the first layer of the Qwen2 model. In Figure 14, we take a closer look at the distribution of $\delta_\Delta$ in one of such attention heads. This figure shows that the distribution of $\delta_\Delta$ in these outliers still largely resemble a Gaussian Distribution. However, the distribution appears

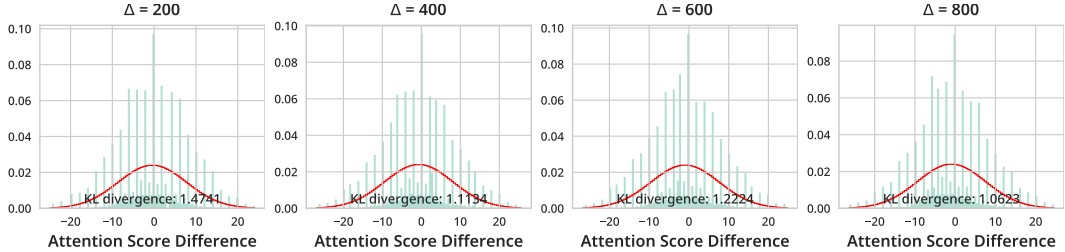

Figure 14: **Close Examination of $\delta_\Delta$ for the Qwen2 Model.** The above figures plot the distributions of $\delta_\Delta$ of the 3rd attention head of the first layer for the Qwen2 model, which was also shown in Figure 16. Close examination on these figures suggest that although the KL divergence scores are high, $\delta_\Delta$ still largely follows a Gaussian distribution.

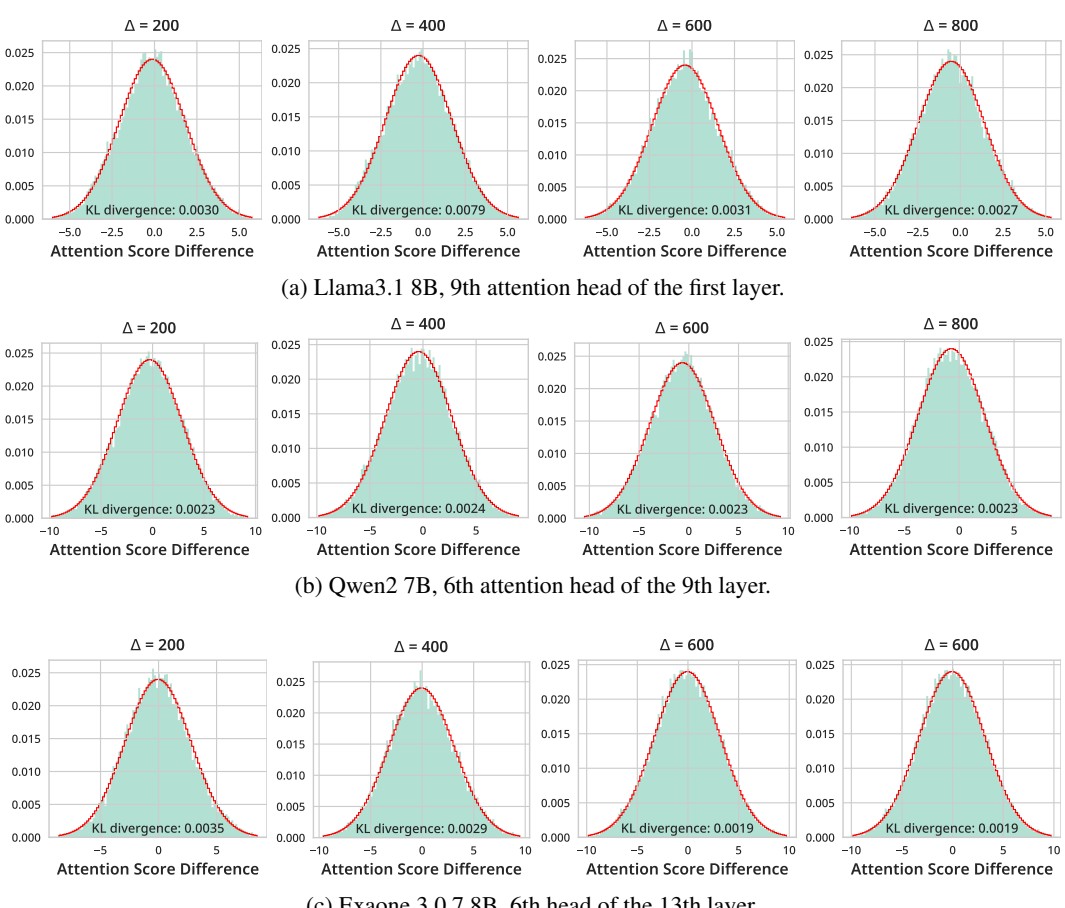

(a) Llama3.1 8B, 9th attention head of the first layer.

(b) Qwen2 7B, 6th attention head of the 9th layer.

(c) Exaone 3.0 7.8B, 6th head of the 13th layer.

Figure 15: **Distribution of $\delta_\Delta$.** The above figures show the distributions of $\delta_\Delta$ across various models for $\Delta$ values 200, 400, 600, and 800, respectively. The Wikitext2 dataset was used as input.

to be severely discontinuous and quantized, which is the reason why the KL divergence values were so high. We suspect that this is the result of some internal operations of the Qwen2 model, which we are not aware of. Although these few occasions stand out as anomalies, we believe that it does not compromise the integrity of our assumption for two reasons. First, it is an extremely rare occasion that only occurs in the few early layers of the Qwen2 model. Secondly, even though the results are discrete, the distribution of $\delta_\Delta$ still resembles a normal distribution. Therefore, our assumption of modeling $\delta_\Delta$ as a normal distribution is valid.

Next, we show that $\sigma(\Delta)$ is an increasing function of $\Delta$. We observe an interesting phenomenon, where the overall demeanor with which $\sigma(\Delta)$ increases resembles a logarithmic function, as can be seen in Figure 12. Although there may be some amount of minor deviations, the standard deviation

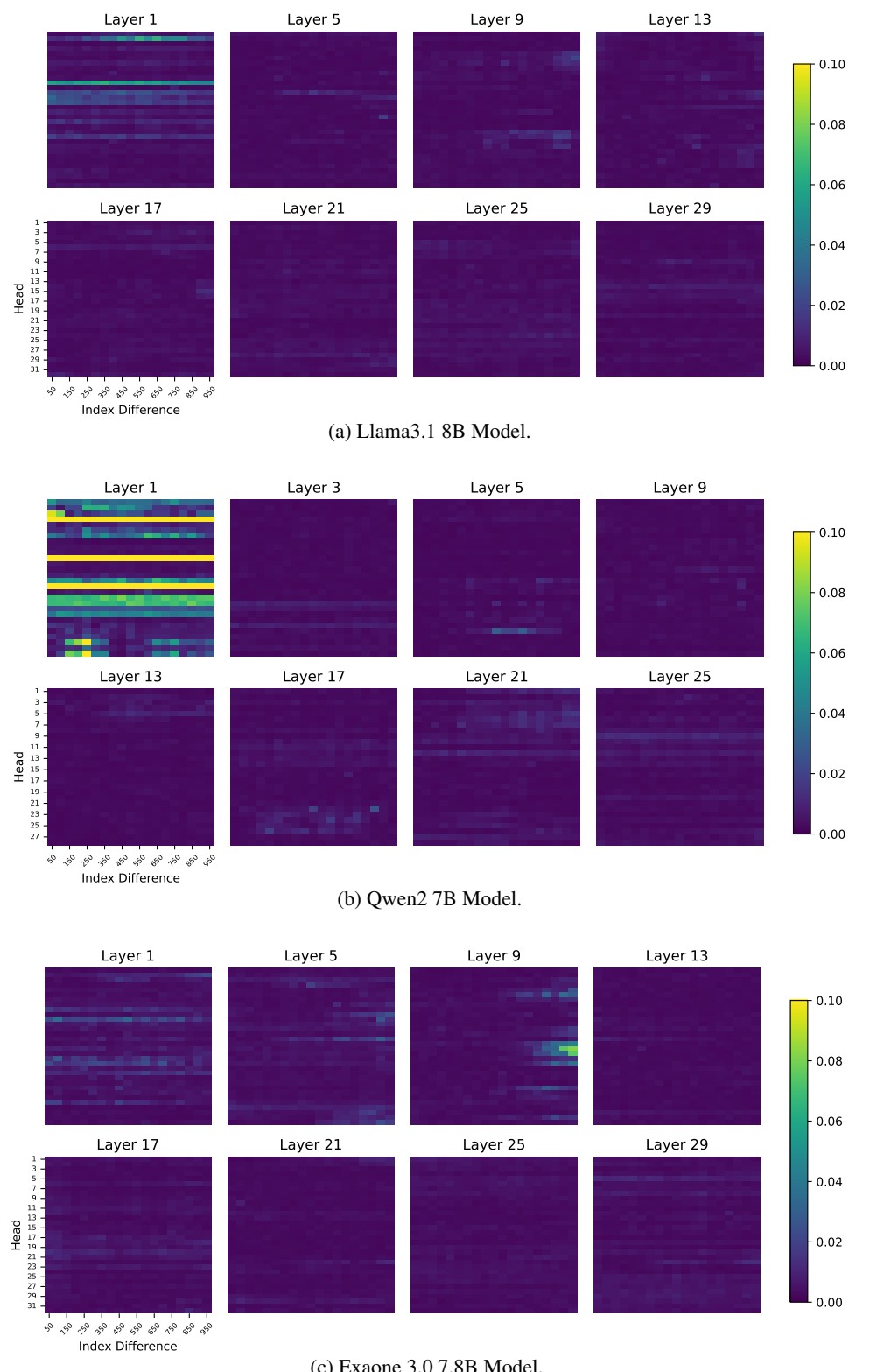

(a) Llama3.1 8B Model.

(b) Qwen2 7B Model.

(c) Exaone 3.0 7.8B Model.

Figure 16: **KL Divergence.** The above figures express the KL Divergence between $\delta_\Delta$ and its fitted normal distributions of various layers. The $y$ axis represents attention heads, and the $x$ axis represents $\Delta$, ranging from 50 to 950. We use the Wikitext2 dataset as input for the model.

$\sigma(\Delta)$ of $\delta_\Delta$ has consistently shown to resemble a logarithmic function for almost all layers and attention heads. In order to prove this claim, we show the results for fitting $\sigma(\Delta)$ as a logarithmic function for each attention head and layer. As can be seen in Figure 11, except for a few outliers (especially in the early layers), the results converge within NRMSE (Normalized Root Mean Squared

Error) of 0.1, which suggests a very good fit. This shows that $\sigma(\Delta)$ resembles a logarithmic function and, therefore, can be seen as an increasing function.

The above observation actually opens up the room for discussion about the optimality of HiP. In Theorem 2, $\mathbf{P}(\mathbb{X} \cap \mathbb{A}) + \mathbf{P}(\mathbb{Y} \cap \mathbb{B})$ is the probability of HiP making the right choice, regardless of the location of the maximum token. Therefore, Theorem 2 proves that if $\sigma''(\Delta)\sigma(\Delta) < \sigma'(\Delta)^2$, then not only does HiP perform better than random token selection, but it is also in fact optimal. If $\sigma(\Delta)$ resembles a logarithmic function like our observation, then $\sigma''(\Delta) < 0$, and therefore the inequality $\sigma''(\Delta)\sigma(\Delta) < 0 <= \sigma'(\Delta)^2$ holds. Thus, on top of HiP guaranteeing better performance than random token selection, there also is the possibility of HiP being, in fact, optimal.

However, we are very careful with making this claim because we do not yet have a logical explanation about why $\sigma(\Delta)$ displays a logarithmic increase. In the main paper, based on the assumption of locality, we only claim that $\sigma(\Delta)$ is an increasing function of $\Delta$. However, the assumption of locality does not cover the exact detailed behavior of $\sigma(\Delta)$. There is always the possibility that the logarithmic pattern we see in $\sigma(\Delta)$ may be the result of some model-specific traits, the overall training corpus, or some other reasons that we are not aware of. Therefore, although we find this observation interesting, we only leave it as a discussion topic in the appendix section. We plan to analyze the detailed characteristics of the $\sigma(\Delta)$ function in future work.

Finally, we discuss approximating the mean to zero. Unlike our assumption, the mean of which normal distribution $\delta_\Delta$ follows does not stay as zero. As can be seen in the following Figure 13, the mean of $\delta_\Delta$ consistently begins from zero when $\Delta = 0$, but it moves away from zero as $\Delta$ increases. The manner in which the mean gets further away from zero is not consistent. In some cases, the mean decreases negatively away from zero, and in some cases, it is the opposite. The exact behavior differs between different layers, or attention heads or the input dataset, so it seems that we cannot deterministically express the mean in terms of math.

However, this does not undermine the integrity of our mathematical analysis. Note that in our proof, we use the absolute attention score difference $|\delta_\Delta|$. The main logic of our proof is that since attention scores display locality, the representative token near the maximum token is likely to have a larger score than the other representative token and, hence, is more likely to be chosen by HiP's algorithm. This does not change, even if the mean of $\delta_\Delta$ displays the aforementioned nonzero characteristic. Since $\delta_\Delta$ moves away from zero as $\Delta$ increases, $|\delta_\Delta|$ is an increasing function of $\Delta$. This makes it even more likely that the representative token near the maximum token will have a larger score than the other representative token compared to when the mean is zero. For this reason, $\psi(\sigma(\alpha), \sigma(\beta))$ is still a decreasing function of $\alpha$ and an increasing function of $\beta$, even if the mean exhibits nonzero characteristics. Thus, our mathematical analysis remains valid.

We approximate the mean as zero mainly for three reasons. First, even though the mean gradually moves away from zero, regardless, it is still close to zero and much smaller than the standard deviation of $\delta_\Delta$. Secondly, the behavior of the mean is not consistent, which makes it very difficult to express it in terms of math. Finally, the approximation does not compromise the integrity of the mathematical analysis while making the overall derivation much simpler and easier to understand. These are the reasons why we approximate the mean as zero in this paper.

### A.3.2 THE INDEPENDENCE BETWEEN $\delta_\alpha$ AND $\delta_\beta$

Now, we talk about the independence between $\delta_\alpha$ and $\delta_\beta$. In Lemma 1, for simplicity, we have assumed that $\delta_\alpha$ and $\delta_\beta$ is independent of each other. However, this is not necessarily true. As can be seen in Figure 17, the correlation coefficient between $\delta_\alpha$ and $\delta_\beta$ is nonzero. From empirical analysis, we observe that the correlation coefficient between $\delta_\alpha$ and $\delta_\beta$ is actually dependent on $|\alpha - \beta|$ - the distance between the two tokens. As can be seen in Figure 18, the correlation coefficient decreases from around 0.7 when the distance is short to around 0.4 when the distance is long.

This suggests that $\delta_\alpha$ and $\delta_\beta$ have a strong positive correlation when $\alpha$ and $\beta$ are close to each other, and a weak positive correlation when they are far away. In other words, nearby tokens have a higher probability of having similar attention scores. This is exactly equivalent to the assumption of attention locality that we discussed in the main section of this paper, and this empirical data is the strongest evidence we have that supports this assumption. To summarize, due to attention locality, $\delta_\Delta$ is not independent, and nearby tokens are less independent than faraway tokens.

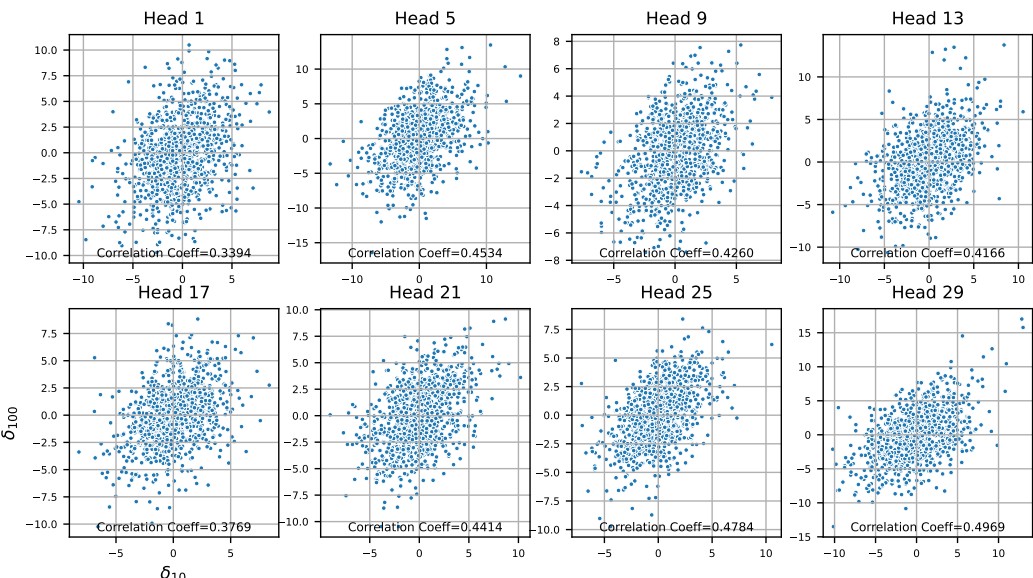

Figure 17: **Correlation Coefficient Between $\delta_{10}$ and $\delta_{100}$.** The above figures show the scatter plots of the joint distribution of $\delta_{10}$ and $\delta_{100}$. The nonzero correlation coefficient suggests that there exists some amount of dependency between $\delta_{10}$ and $\delta_{100}$. The 16[th] layer of the Llama3.1 8B model was used for sampling the values. We use Wikitext2 dataset for the input.

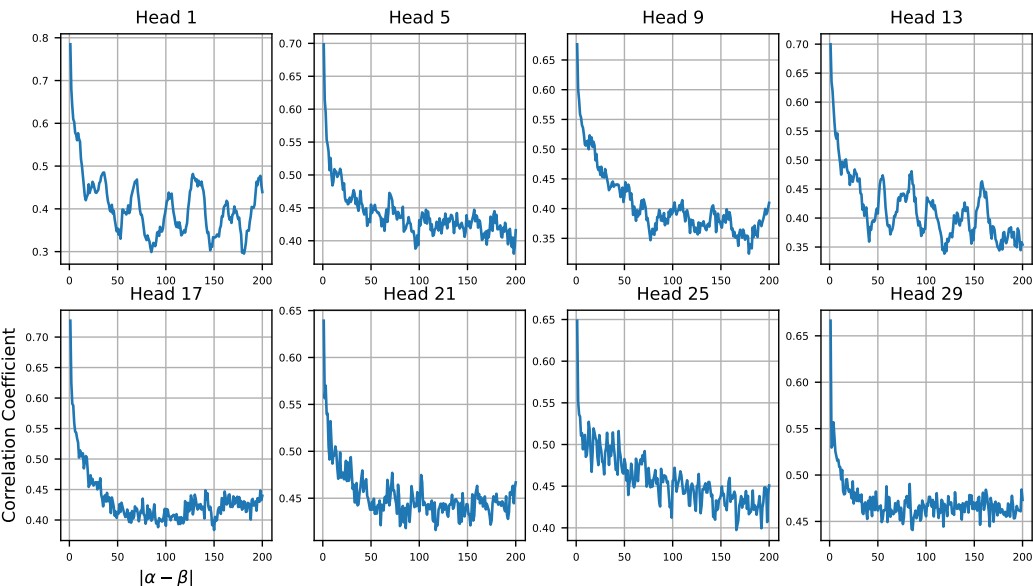

Figure 18: **Correlation Coefficient.** The above figures plot the relationship of the correlation coefficient between $\delta_{\alpha}$ and $\delta_{\beta}$, and the index difference $|\alpha - \beta|$ for different attention heads. In order to obtain a more clearer plot, the value of $\alpha$ was fixed to 10. As can be seen in the figures, the correlation coefficient consistently decreases from 0.7 to around 0.4, as the distance between $\alpha$ and $\beta$ increases. The values were sampled from layer 16 of the Llama3.1 8B model, using the Wikitext2 dataset as input.

If so, then does the dependency between $\delta_{\alpha}$ and $\delta_{\beta}$ disprove our analysis? In order to analyze this issue, we compare $\mathbf{P}(\delta_{\alpha} = x, \delta_{\beta} = y)$ when $\delta_{\Delta}$ are independent and dependent. When they are independent, then $\mathbf{P}(\delta_{\alpha} = x, \delta_{\beta} = y)$ is expressed as follows.

$$\mathbf{P}(\delta_{\alpha} = x, \delta_{\beta} = y) = \mathbf{P}(\delta_{\alpha} = x)\mathbf{P}(\delta_{\beta} = y) = \frac{1}{2\pi\sigma(\alpha)\sigma(\beta)} \exp - \left( \frac{x^2}{2\sigma(\alpha)^2} + \frac{y^2}{2\sigma(\beta)^2} \right).$$

When $\delta_\alpha$ and $\delta_\beta$ are dependent with the correlation coefficient being $\rho$, the probability can be expressed using the multivariate normal distribution as follows.

$$\mathbf{P}(\delta_\alpha = x, \delta_\beta = y)$$
$$= \frac{1}{2\pi\sigma(\alpha)\sigma(\beta)\sqrt{1-\rho^2}} \exp\left(-\frac{1}{2[1-\rho^2]}\left[\frac{x^2}{2\sigma(\alpha)^2} + \frac{y^2}{2\sigma(\beta)^2} - 2\rho\frac{xy}{\sigma(\alpha)\sigma(\beta)}\right]\right).$$

We argue that the critical difference between these two equations is the nonlinear term $1 - \rho^2$. When the tokens are far apart, i.e., $\rho \simeq 0.4$, then the $1 - \rho^2 = 0.85$, which is relatively close to 1. Therefore, we can largely approximate the dependent $\mathbf{P}(\delta_\alpha = x, \delta_\beta = y)$ equation with its independent version without much error. However, when the tokens are near to each other, i.e., $\rho \simeq 0.7$, then $1 - \rho^2 = 0.51$, which is significantly different from 1. In this case, we can no longer justify the approximation of independence.

This means that if the representative tokens are too close to each other, then our mathematical analysis, which proves that HiP performs better than random token selection, might no longer hold. Interestingly, we have actually observed experimental results that support this notion. In the last few iterations of HiP, we have noticed that HiP's binary search algorithm does not perform well, unlike previous iterations. The main reason behind this phenomenon is that in the last few iterations, the sections become too short and too close to each other. Since they are all close to each other, due to attention locality, there no longer exists a significant attention score difference between each section. Hence, HiP fails to successfully identify the sections with the largest attention scores, and deteriorates to almost random token selection level performance.

In order to resolve this issue, we tried to develop a simple patch that acts after the top-$k$ selection process. Recall that in order to foster the efficient usage of matrix multiplier units, HiP was developed with block granularity. Previously, after the hierarchical top-$k$ selection process, only the selected blocks are used for creating the sparse attention mask. Instead, we grabbed entire vicinity of tokens around the selected blocks, and used those for the sparse attention mask. The key idea was that due to attention locality, the tokens around the selected blocks would have the highest attention scores in the entire sequence. If HiP cannot choose among those tokens, the easiest workaround would be to use all of them. Therefore, by simply using the entire range around the selected blocks, we thought that we would be able to save the most important tokens from being discarded. During development, we used range sizes of 16 or 32.

The effects of this experimental revision definitely improved the overall performance. As can be seen in table Table 7, the relative performance compared to FlashAttention2 was enhanced by 5%. However, as this patch requires the algorithm to access much more key tokens than before, it is significantly detrimental to the overall latency. Due to this reason, we did not include this experimental patch in the final version of HiP. Yet, we believe that with further research, we may be able to find a method that can exploit the strengths of this approach while minimizing the additional latency. Thus, we reserve this topic as a future work.

Table 7: $b_k$ **Patch Results.** In order to resolve the dependency issue of $\delta_\Delta$, we test an experimental patch where the $b_k$ values are modified for the last few iterations. Results show that the patch is beneficial for overall performance yet comes at the cost of latency. The Relative Performance is measured compared to FlashAttention2. The unit of the prefill latency is milliseconds (ms). Also, note that we use 128k context length for experiments.

| | NQA | Qasper | HQA | 2WM | GR | MN | Rel. Perf | Prefill Latency $T$=128k |
|---|---|---|---|---|---|---|---|---|
| Flash Attention | 29.53 | 44.27 | 54.61 | 39.49 | 34.99 | 27.39 | 100.0% | 855.2 |
| HiP $k = 512$ | 21.44 | 41.73 | 50.36 | **45.17** | 30.09 | 26.01 | 92.40% | **105.7** |
| Range Size 16 | 25.46 | 44.12 | **52.87** | 42.30 | 33.40 | 26.89 | 97.51% | 181.4 |
| Range Size 32 | **26.82** | **44.30** | 52.85 | 41.23 | **33.71** | **27.18** | **97.88%** | 227.2 |

To summarize, when the representative tokens are far apart, the dependency between $\delta_\Delta$ is weak, so we can approximate them as independent. When the representative tokens are nearby, (i.e. last few iterations of HiP), the dependency of $\delta_\Delta$ can no longer be ignored, and thus HiP's binary search like algorithm no longer suffices. We acknowledge this issue, and have tried to develop an experimental

method that solves the problem. We find that while the method definitely helps performance, it comes at a large cost of latency. We plan on tackling this problem in future work.

# B    DETAILED METHODOLOGY DESCRIPTIONS

## B.1    HIERARCHICAL SPARSE ATTENTION MASK ESTIMATION ALGORITHM

---
**Algorithm 1** Hierarchical Sparse Attention Mask Estimation

---

**input**  Queries $\boldsymbol{Q} \in \mathbb{R}^{T \times d}$, Keys $\boldsymbol{K} \in \mathbb{R}^{T \times d}$, Sparsity constant $k$, Query block size $b_q$, Key block size $b_k$, Top-$r$ approximation constant $r$.

**output**  Estimated attention mask $\widehat{\boldsymbol{M}} \in \{0,1\}^{T \times T}$ which is represented by an array of indices $\mathcal{I} \in [1:T]^{T/b_q \times k/b_k}$.

1:  $\mathbf{Q}, \mathbf{K} = \mathrm{reshape}_{T/b_q \times b_q \times d}[\boldsymbol{Q}], \mathrm{reshape}_{T/b_k \times b_k \times d}[\boldsymbol{K}]$.

2:  Number of iterations $n_{it} = \lceil \log(T/b_k) \rceil$.

3:  **for each** query block index $q = 1 .. T/b_q$ **do**

4:  $\quad \left( f_{qj}^{(1)}, l_{qj}^{(1)} \right) = \left( \lfloor (j-1) \cdot \frac{T}{k} \rfloor + 1, \lfloor j \cdot \frac{T}{k} \rfloor \right)$ for $j = 1 .. k$.    ▷ *Set $k$ nodes' initial start and end indices*

5:  $\quad$ **for each** iteration $i = 1 .. n_{it}$ **do**

6:  $\quad\quad$ **for each** node index $j = 1 .. k$ **do**

7:  $\quad\quad\quad m_{qj}^{(i)} = \left\lfloor (f_{qj}^{(i)} + l_{qj}^{(i)})/2 \right\rfloor$.

8:  $\quad\quad\quad \left( \mathcal{B}_{q,2j-1}^{(i)}, \mathcal{B}_{q,2j}^{(i)} \right) = \left( (f_{qj}^{(i)} : m_{qj}^{(i)} - 1), (m_{qj}^{(i)} : l_{qj}^{(i)}) \right)$.

9:  $\quad\quad$ **for each** branch index $h = 1 .. 2k$ **do**

10:  $\quad\quad\quad$ Pick a center index $r_{qh}^{(i)}$ from the range $\mathcal{B}_{qh}^{(i)}$.

11:  $\quad\quad\quad$ Compute score $s_{qh}^{(i)} = \max_{m,n} \left( \mathrm{causal\_mask} \left( \mathbf{Q}_{q,m,:}^{\top} \mathbf{K}_{r_{qh}^{(i)}, n, :} \right) \right)$.

12:  $\quad\quad\quad$ Pick top-$k$ indices $\{t_1, \ldots, t_k\}$ of the sequence $s_{q,1}^{(i)}, \ldots, s_{q,2k}^{(i)}$.

13:  $\quad\quad\quad$ Update nodes $(f_{qj}^{(i+1)} : l_{qj}^{(i+1)}) := \mathcal{B}_{t_j}^{(i)}$ for $j = 1 .. k$.

14:  $\quad$ Set mask indices $\mathcal{I}_{qj} = f_{qj}^{(n_{it})}$ for $j = 1 .. k$.

---

In Algorithm 1, we describe the complete algorithm used for mask estimation. The peak amount of memory used during the mask estimation process is in $O(T)$, since at each iteration, only the immediately previous iteration's node indices are needed, and the rest can be discarded.

## B.2    HIP DECODING ALGORITHM

---
**Algorithm 2** HiP Decoding Algorithm

---

**input**  The model $\mathcal{M}$, the number of layers $L$, the mask estimation period $r_m$, number of initial dense layers $l_d$.

**output**  Generated sequence $y$.

1:  Initialize $y$ with an empty sequence.

2:  **while** generation has not ended **do**

3:  $\quad$ **for each** layer $l = 1 .. N$ **do**

4:  $\quad\quad$ **if** $l < l_d$ **then**

5:  $\quad\quad\quad$ Perform regular dense attention for the current layer.

6:  $\quad\quad$ **else**

7:  $\quad\quad\quad$ **if** the current generated sequence length is divisible by $r_m$ **then**

8:  $\quad\quad\quad\quad$ For each head, estimate the attention mask with Algorithm 1.    ▷ *$O(T \log T)$ time*

9:  $\quad\quad\quad\quad$ Cache the attention masks.    ▷ *$O(T)$ space*

10:  $\quad\quad\quad$ Perform fused sparse attention using the cached attention masks.    ▷ *$O(T)$ time & space*

11:  $\quad$ Sample the next token and append it to the sequence $y$.

---

In Algorithm 2, we show a rough sketch of the decoding process with HiP. In particular, note the function of the mask estimation period $r_m$ and the number of initial dense layers $l_d$, as well as the time and space complexities of each component.

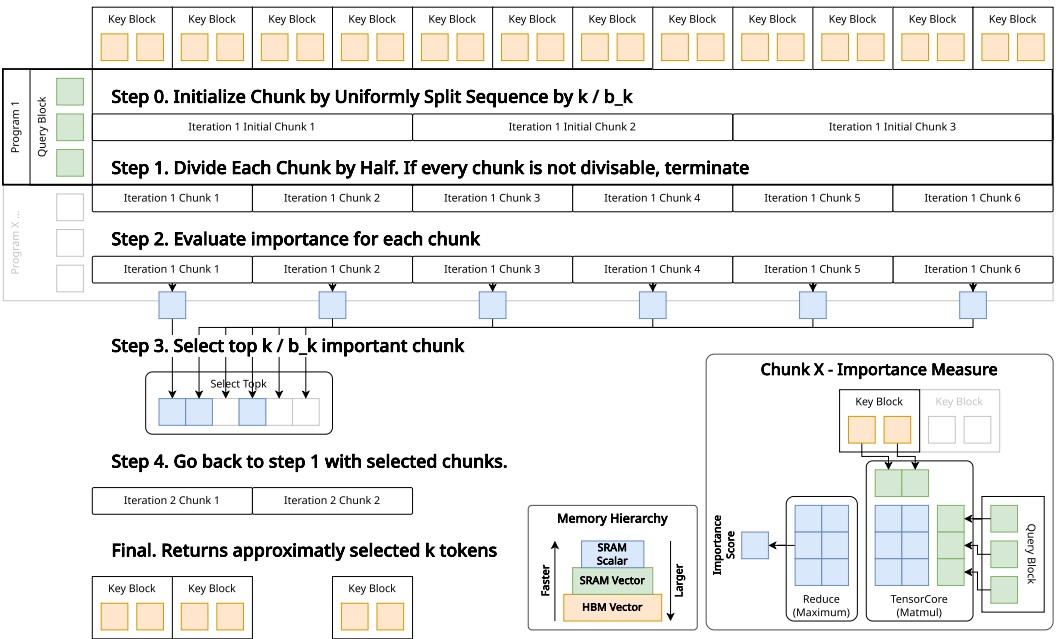

Figure 19: **Detailed Flowchart of Hierarchically Pruned Attention.** We illustrate the internal steps of each program of hierarchical attention pruning. We instantiate a single program for each attention row due to row-level synchronization on the top-k operation.

### B.3 Detailed Flow-diagram of HiP

In Figure 19, we illustrate hierarchical attention pruning step-by-step.

### B.4 Additional Optimization Techniques

#### B.4.1 Top-r Approximation

In order to reduce the cost of the mask estimator even further, we take inspiration from SparQ Attention (Ribar et al., 2013), where global memory (HBM or GDDR) transfer is reduced by selectively fetching only the most relevant components of the key vectors. Specifically, when computing the inequality condition in Equation (4), instead of fetching all $d$ components of the key vectors, we only fetch $r \ll d$ most prominent components estimated by the query vector $\boldsymbol{q}$. Thus, we compute the following as an approximation:

$$\boldsymbol{q}^\top \boldsymbol{k}_t \approx \sum_{l=1}^{r} \boldsymbol{q}_{p_l} \cdot \boldsymbol{k}_{p_l} \tag{6}$$

where $\{p_1, p_2, \ldots, p_r\} = \mathrm{argtop\_}r(|\boldsymbol{q}|)$. By using the top-$r$ approximation, the total number of global memory accesses in the mask estimation stage can be drastically reduced. However, we disable this approximation by default.

#### B.4.2 Block Sparse Flash Attention

We utilize the Flash Attention (Dao et al., 2022) mechanism to reduce the latency of sparse attention and use a small size sliding window to reduce performance degradation on the side of the sparse attention kernel. Following the receipt of StreamingLLM (Xiao et al., 2024), local sliding window and global sink attention are also added during block sparse flash attention operation.

### B.5 Training Downstream Tasks with HiP

In this section, we describe the HiP attention training strategy for downstream tasks. We discovered that direct fine-tuning after applying HiP could not achieve the performance of the fine-tuned vanilla

attention. Empirically, HiP's highly sparse attention matrices show excellent performance approximation during test time but not in train gradients. Since our method heavily prunes the attention matrix, the gradient cannot flow through dense attention probabilities. This incomplete and unstable gradient of the attention matrix leads to significant training performance degradation because HiP forces the model to have attention patterns similar to those of the pretrained model rather than adopting them for the downstream task.

However, we could achieve the same performance as vanilla by using the following two-step training strategy: (1) fine-tuning with vanilla first and (2) further fintuning after applying HiP (healing). First, we train the pretrained model with vanilla attention to the downstream task as usual. Then, we load the finetuned weight, apply HiP, and further finetuning with the downstream task dataset with just a few optimizer steps. We call this a healing process because we only finetune a few steps from the finetuned weight. For training details about each experiment, we describe hyperparameter and optimization setups in Appendix D.

## C  ADDITIONAL EXPERIMENTAL RESULTS

### C.1  LARGE MULTIMODAL MODEL WITH HIP

Table 8: **LMMs-Eval on LLaVA-1.6-13B.** We apply the efficient attention mechanism in only LLM backbone of LMM.

| Method | MME cog | MME percep | MMMU | DocVQA | GQA | TextVQA | Avg. |
|---|---|---|---|---|---|---|---|
| Flash Attn | 324 | 1575 | 36% | 77% | 65% | 67% | 100.0% |
| SLLM $k$=512 | 242 | 1094 | 31% | 25% | 52% | 27% | 63.7% |
| BigBird $k$=512 | 320 | 1526 | 36% | 53% | 64% | 55% | 90.7% |
| HiP $k$=256 | 303 | 1465 | 37% | 68% | 64% | 63% | 94.7% |
| HiP $k$=512 | 286 | 1500 | 36% | 74% | 64% | 65% | 95.9% |

Since large multi-modal models (LMM) (Liu et al., 2023b; 2024) leverage pre-trained LLMs as a backbone for their NLU ability, without changes, except for the tokenizer, we also evaluated our HiP on top of LLaVA-1.6-13B (Liu et al., 2024), a large multi-modal model, using LMMs-eval (Li* et al., 2024), which provides extensive benchmarks for large multimodal model suites. As we show in Table 8, our method scores 95.9% relative scores, which is similar to the performance recovery ratio of HiP on NLU tasks.

### C.2  MASSIVE MULTITASK LANGUAGE UNDERSTANDING (MMLU)

Figure 20: MMLU scores on Llama3.1-8B.

| Method | Humanities | STEM | Social Sciences | Other | Avg. |
|---|---|---|---|---|---|
| FlashAttn | 67.35 | 51.88 | 73.21 | 66.52 | 63.24 |
| SLLM | 66.65 | 52.22 | 73.98 | 65.68 | 63.16 |
| BigBird | 66.94 | 51.06 | 73.27 | 66.21 | 62.81 |
| HiP (Ours) | 67.64 | 51.50 | 73.19 | 66.32 | 63.13 |

Next, we evaluate HiP on the MMLU benchmark (Hendrycks et al., 2021) to show that our method does not negatively affect the NLU ability of the pretrained model. The results show that our HiP preserves the NLU performance of the original model. All tested methods are able to recover original MMLU scores without significant loss here. This is probably due to the nature of the MMLU task: the answer is only dependent on the most recent span of tokens rather than the entire prompt (which contains few-shot examples in the beginning).

### C.3  COMPARISON WITH REFORMER AND SEA

We compare the performance of HiP against Reformer (Kitaev et al., 2019) and SEA (Lee et al., 2023) using the Llama2-7b model and show the results in Table 9. Even though HiP requires no

Table 9: **Comparison of WikiText2 perplexity against Reformer and SEA.** For this comparison, we use the Llama2-7b model. The experimental setting for HiP is the same as the main experiment. For Reformer, we use 512 for the bucket size and replace the attention module in all layers except the first three for a fair comparison with HiP. For SEA, we use 512 for the attention predictor length and $k$ (the top $k$ sampling factor) as 64. We fine-tune both Reformer and SEA models using LoRA with rank = 128, the same as HiP HEAL, until convergence.

| Method | Perplexity |
|---|---|
| Vanilla PyTorch | 5.5933 |
| Reformer FT (Kitaev et al., 2019) | 45.5130 (+39.92) |
| SEA FT (Lee et al., 2023) | 23.1964 (+17.60) |
| **HiP** (k=512) PP | **5.7938** (+0.20) |
| **HiP** (k=512) HEAL | **5.6872** (+0.09) |

Table 10: **Self-Extend Result with HiP.** We apply Self-Extend with HiP $k$=256 on Gemma2-2B-it. We extend the maximum context length of Gemma2 from 8k up to 128k tokens. We measure the perplexity of Wiktiext2.

| Method | Self Extend | Sliding Window | $T$=1k | $T$=2k | $T$=4k | $T$=8k | $T$=16k | $T$=32k | $T$=64k | $T$=128k |
|---|---|---|---|---|---|---|---|---|---|---|
| FA2 | ✗ | ✓ | 18.90 | 14.48 | 12.35 | **11.40** | *46.66* | *167.40* | *363.02* | - |
| FA2 | ✗ | ✗ | 18.90 | 14.48 | 12.35 | *30.17* | *198.84* | *638.65* | *1310.26* | - |
| HiP | ✓ (x4) | ✗ | 18.98 | 14.77 | 13.00 | 12.14 | **12.00** | *46.51* | *277.54* | - |
| HiP | ✓ (x8) | ✗ | 18.98 | 14.90 | 13.30 | 12.57 | **12.22** | 12.63 | *49.26* | - |
| HiP | ✓ (x4-8) | ✗ | 18.98 | 14.81 | 13.15 | 12.36 | **12.06** | 13.62 | *116.18* | - |
| HiP | ✓ (x8-16) | ✗ | 18.98 | 15.00 | 13.51 | 12.83 | **12.53** | 12.55 | 14.32 | - |
| HiP | ✓ (x8) | ✓ | 19.28 | 14.86 | 12.87 | 11.95 | 11.53 | **11.36** | 11.69 | - |
| HiP | ✓ (x16) | ✓ | 19.28 | 14.98 | 13.11 | 12.25 | 11.86 | 11.69 | **11.54** | 11.68 |

fine-tuning, unlike Reformer and SEA, which need fine-tuning, our HiP attention is far superior in language modeling performance compared to these two baselines.

For a fair comparison, we fine-tune Reformer and SEA using LoRA (Low-rank adapters), which have the same rank as the healed version of HiP. Due to this, the Reformer and SEA's performance converges to a much-degraded value compared to the values reported in their respective original papers. We conclude that both methods need much modification to the original weights in order to perform well, whereas since HiP performs well even without any modification to the weights whatsoever, a small amount of LoRA training can even close this small gap.

## C.4 CONTEXT EXTENTION WITH SELF-EXTEND

Since our method has a considerable advantage in long-context decoding, we need the pre-trained long-context model. Unfortunately, however, not all pretrained transformer models support long contexts. Therefore, many previous studies (Peng et al., 2023; Jin et al., 2024) try to extend maximum position embeddings of the trained model to extend the context size of LLM. We adopt the SelfExtend (Jin et al., 2024) method into our method because it is also a training-free context extension method. We picked Gemma2 (Team, 2024) to target LLM and extend the context model because the model is a hybrid of sliding windows and dense attention. Therefore, it will have the advantage of a long context model with HiP by saving the KV cache of sliding window layers. The Gemma2 repeats the attention layer by repeating the stack of different attention blocks: sliding window and dense attention. To evaluate the effectiveness of the combination of HiP and SelfExtend, we apply them to attention layers.

We can observe Gemma2 explode after its pretrained context length, which is 8192 (First row of the Table 10). We can see the model fails after the sliding window context length, which is 4096 for the sliding window layer (Second row of the Table 10). Therefore, we know that treating sliding windows especially is quite essential for performance. In the third and fourth rows of the Table 10, we apply the same Self-Extend group size for every attention layer, including the layer that was originally a sliding window, before replacing it with HiP. We could observe the settings are struggling to

recover performance right after Self-Extend Group Size × Sliding Window Size, so we apply twice the larger Self Extend group size for the HiP layers originally sliding window. The modified group size application is in the fifth and sixth rows of the Table 10. We could observe that we can extend the context window as expected, not bounded by sliding window size. However, the above replacements are quite restarted setting because they are not even allowed to use sliding window attention, which is usually very efficient in practical LLM frameworks. Therefore, we replace only dense attention to HiP, and we could observe a significant performance boost from all layer replacements while extending context length up to 128k, as shown in the last two rows of the Table 10.

### C.5 ENSEMBLE HIERARCHICAL TOP-$k$ APPROXIMATION

For the first attempt to address the challenges in HiP described in Appendix E.9, we perform an ensemble of HiP masks by introducing slight randomness to the branching decisions within a given iteration. Our empirical results on Wikitext2 indicate that the ensemble of HiP with no dense layers ($l_d = 0$) achieves comparable perplexity to HiP with dense layers ($l_d = 3$) across context lengths from 4k to 16k. This highlights how considering different branching strategies within a given iteration can improve inference performance and suggests the potential for replacing the dense layers in HiP with comparable performance.

**Methodology.** First, we generate $n_e$ HiP attention mask samples by adding randomness $r_e$ to the branch direction when the middle iteration branches out to the next iteration. As $r_e$ slightly adjusts the selected node in the iteration, each sample can take a slightly different path during tree branching, leading to a diverse construction of attention masks.

Second, we count the number of agreements of the indices per query and retain the indices that exceed or equal to the agreement threshold $\theta_{\text{vote}}$. Therefore, $\theta_{\text{vote}} = 1$ functions as a union operator and $\theta_{\text{vote}} = n_e$ as an intersection operator. By adjusting $\theta_{\text{vote}}$ from 1 to $n_e$, we can perform an operation that lies between union and intersection, prioritizing indices with a higher number of votes. To prevent the union-like $\theta_{\text{vote}}$ increasing the number of active attention elements too much, we truncate the number of the final selected indices by original $k$ when $\tau \in \{0, 1\}$ is 1, prioritizing the indices having more votes.

Lastly, we perform the introduced ensemble method for the first $l_e$ layers, just like we do with $l_d$.

**Experiments.** We first provide experimental evidence on how ensemble enables end-to-end subquadratic complexity and then give details on our hyperparameter tuning process. The experiments are conducted with the LLaMA2-7B model.

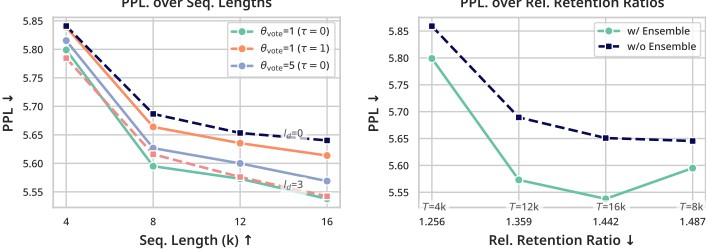

Figure 21: **Perplexity Evaluation on Long Context. (Left)** Perplexity result in Wikitext2 with different ensemble settings ($\theta_{\text{vote}}$, $\tau$) where $r_e = 5.0$, $l_e = all$. **(Right)** Perplexity comparison between full HiP ($l_d = 0$) and ensemble ($\theta_{\text{vote}} = 1$, $\tau = 0$, $r_e = 5.0$, $l_e = all$) using same relative retention ratio in each sequence length.

**Performance Comparison with Original HiP.** To show that ensemble enables end-to-end subquadratic complexity with comparable performance to our default HiP ($l_d = 3$), we compare the full HiP ($l_d = 0$), the default HiP ($l_d = 3$), and the ensemble with $l_d = 0$. We fix $r_e = 5.0$, $l_e = all$ that gave the best performance in $T = 4096$, as shown in Figure 22. The result indicates that ensemble with $\theta_{\text{vote}} = 1$, $\tau = 0$ outperforms both full HiP and default HiP, as shown in Figure 21 (Left), and therefore this suggests that ensemble could not only improve the performance but also replace the dense layers with comparable performance.

Moreover, we provide a comparison with full HiP ($l_d = 0$) at the same level of sparsity as the ensemble to demonstrate that the improvement is not solely due to the increased number of selected

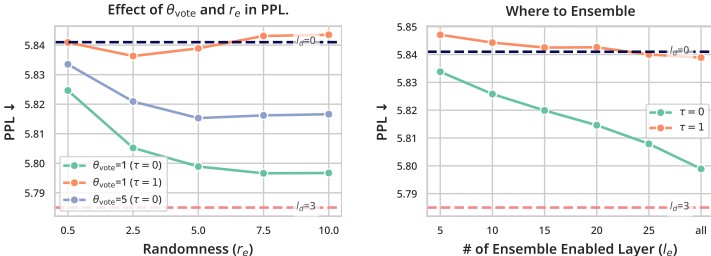

Figure 22: **Effect of Randomness in HiP sampling and Ensemble-Enabled Layers $l_e$.** Perplexity in Wikitext2 with $T = 4k$. **(Left)** We adjust randomness $r_e$ with fixed $l_e = all$. **(Right)** We adjust number of $l_e$ with fixed $\theta_{vote} = 1$, $r_e = 5.0$. The dashed horizontal line shows the performance of HiP ($l_d = 0$, $l_d = 3$).

indices resulting from our ensemble method. As shown in Figure 21 (Right), our ensemble method is Pareto frontier compared to HiP, with performance measured against the retention ratio.

**Latency of Ensemble.** Since we sample multiple HiP masks by $n_e$ and perform voting operations across $n_e \times k$ number of indices, the ensemble costs a few times more than the original HiP. However, since the cost of dense attention grows quadratically, the ensemble will become more efficient compared to the dense attention as the context length increases. Therefore, we think that the use of the ensemble method could be particularly advantageous in extremely long contexts.

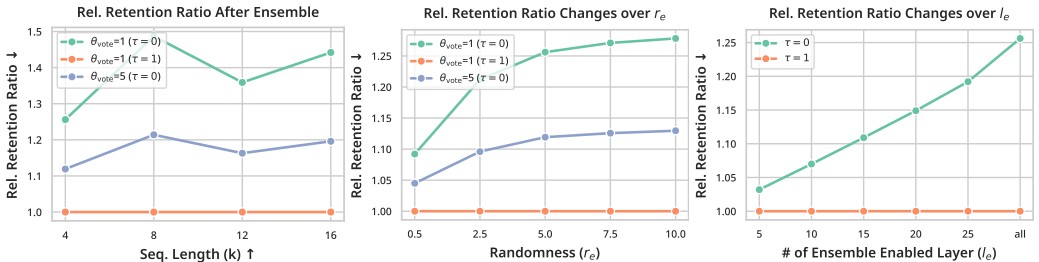

Figure 23: **Relative Retention Ratio and Ensemble Factors.** Relative retention ratio after mask ensemble. The ratio is computed by dividing the number of active indices after the ensemble by before. **(Left)** We adjust the sequence length $T$ with fixed $r_e = 5.0$, $l_e = all$. **(Center)** We adjust the randomness $r_e$ in $T = 4k$ with fixed $l_e = all$. **(Right)** We adjust the number of ensemble enabled layer $l_e$ in $T = 4k$ with fixed $\theta_{vote} = 1$, $r_e = 5.0$. Ensemble disabled layers are computed as 1.0.

**Ablation Study of Ensemble Hyperparameter: $\theta_{vote}$, $\tau$, $r_e$, $l_e$.** As shown in Figure 22 (Left), $\theta_{vote} = 1$ with $\tau = 0$, and $r_e >= 5.0$ gives the highest score. This indicates that performing the union operation with large randomness in sampling gives the highest performance. Also, in Figure 22 (Right), with $\theta_{vote} = 1$, we show that $l_e = all$ works the best, therefore if we want to replace the few layers of vanilla attention, then we would have to apply ensemble in every layer.

**Correlation Between the Relative Retention Ratio and Ensemble Factors.** In Figure 23, we show that the relative retention ratio shows no correlation with sequence length, while randomness shows a positive correlation. Moreover, when measuring the relative retention ratio changes over $l_e$ with $\theta_{vote} = 1$, since we treat the ensemble disabled layer as having a 1.0 relative retention ratio, more ensemble-enabled layers lead to a higher relative retention ratio.

**Analysis with Visualization.** In Figure 24, we provide a visual analysis to show how the ensemble selects indices that HiP missed to fill up a complete attention pattern and how it enables dynamic sparsity. We can see how the ensemble catches missed important indices such as diagonal, vertical, and stride attention patterns in (a), (b), and (c) of Figure 24. Moreover, compared to HiP (left), the union operation (center) enables dynamic sparsity per head. Especially in (c), we can see that the ensemble is effective for filling missed indices in a long sequence while providing dynamic sparsity for each row (red pixels are gradually frequent in the bottom). Lastly, in Figure 24 (center, right), we show how $\tau = 1$ selects indices that receive more votes compared to those selected by $\tau = 0$.

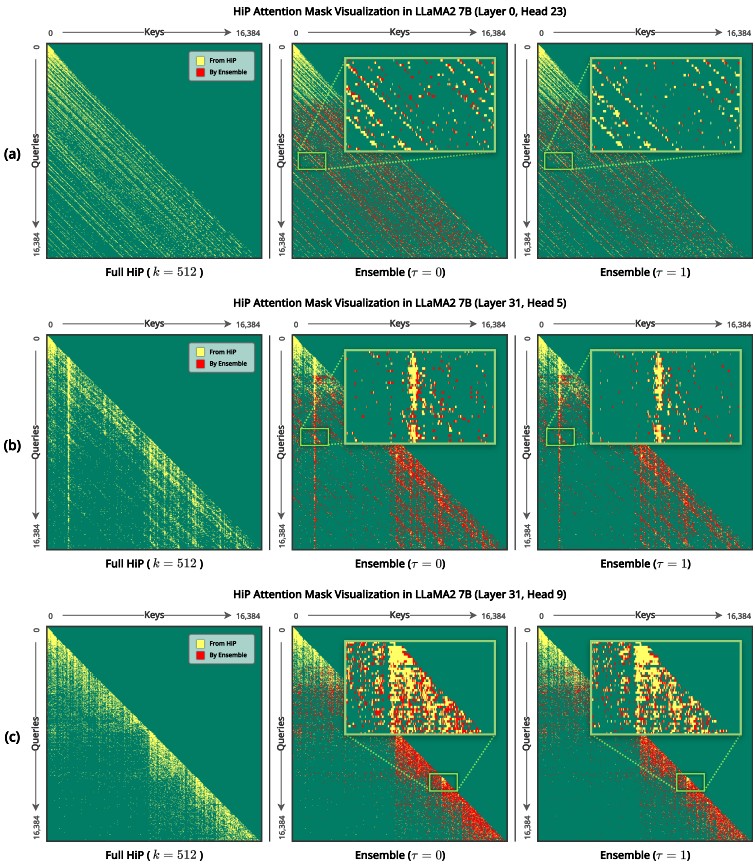

Figure 24: **Attention Mask Ensemble Visualization.** Visualization of attention mask in $T = 16k$ for (**Left**) HiP ($l_d = 0$). (**Center**) ensemble ($\theta_{\text{vote}} = 1, \tau = 0$). (**Right**) ensemble ($\theta_{\text{vote}} = 1, \tau = 1$). Red indicates indices added by our ensemble method, yellow indicates indices from HiP, and green indicates where attention will not be computed.

## D   DETAILED EXPERIMENTAL SETTINGS

**Computation Resources.** We use two machines to run experiments and training. (1) 4090 Machine. We use this local development machine. Most of the micro-benchmark and kernel optimization is done with this machine. (2x RTX 4090 24GB, 128GB DDR5-3600, Ryzen 7950x), (2) A100 Machine. We use this AWS cloud node as the main computation horse. All training and most long context benchmarks are measured with this machine. We use different GPU architectures for kernel development because we could not get an H100 AWS node due to a lack of quota. Therefore, our kernel's CUDA hyper-parameters, such as the number of warps per program and block sizes, are not optimal in the A100 machine. To overcome these mismatches between the development machine and the computation horse, we used `triton.autotune` as much as possible. However, for the above reasons, the optimization of the GPU kernel may not be optimal for every GPU architecture.

**Experiment Settings.** By default, for the HiP experiment, we use $b_q = 32, b_k = 2, k = 512, l_d = 3$ `if` 7B `else` $4, r_m = 8$. For StreamingLLM, we use num_sink = 4.

We show overall experiment settings, such as the number of GPUs and model IDs to which the experiment is introduced (e.g., the caption of the figure and table). To reference, we leave the huggingface model path in Table 11. We used an instruction model from the same provider for instruction following ability-required tasks such as passkey, LongBench, and RULER.

**Experiment Details.** We measure every latency measures with a single NVIDIA RTX 4090, PCIe 4.0 x8, 128GB DDR5-5600, and Ryzen 7950x. The batch size is 32 for decoding and 1 for prefilling. The official implementation of HyperAttention is not available for decoding; therefore, we did not measure the decoding latency and decoding benchmark.

Table 11: **Model IDs on Huggingface.** We use large language models trained on long context inputs in order to demonstrate our method's effectiveness at long context lengths.

| Model | Huggingface ID | Maximum context length |
|---|---|---|
| LLaMA2-7B | `togethercomputer/LLaMA-2-7B-32K` | 32K |
| LLaMA2-13B | `Yukang/Llama-2-13b-chat-longlora-32k-sft` | 32K |
| Qwen1.5-7B | `Qwen/Qwen1.5-7B-Chat` | 32K |
| Qwen1.5-14B | `Qwen/Qwen1.5-14B-Chat` | 32K |
| Luxia-21.4B | `saltlux/luxia-21.4b-alignment-v1.1` | 32K |
| Llama3.1-8B | `meta-llama/Meta-Llama-3.1-8B` | 128K |
| Llama3.1-8B-Instruct | `meta-llama/Meta-Llama-3.1-8B-Instruct` | 128K |
| Gemma2-2B-it | `google/gemma-2-2b-it` | 8K |
| Gemma2-9B-it | `google/gemma-2-9b-it` | 8K |
| Exaone3-7.8B | `LGAI-EXAONE/EXAONE-3.0-7.8B-Instruct` | 4K |

For FlashAttention, we use `flash_attn==2.6.3` for every baseline that requires a FlashAttention backend, such as vLLM, SGlang, and HyperAttention.

We do not use HyperAttention in another experiment because it fails to recover the perplexity of PG19, the most basic metric of language modeling. For HyperAttention (Han et al., 2024) we used `lsh_num_projs=7`, `block_size=64`, `sample_size=1024`, `min_seq_len=32`. We select `min_seq_len` to match the size of the MMU's block size (32) rather than 4096, which is suggested by the authors' code repository. Since in sortLSH (Han et al., 2024) it processes shorter block size of `min_seq_len` with vanilla attention. Therefore, we have to reduce its size to a smaller size than 4096.

Since StreamingLLM does not use block-wise computation just like HiP's block approximation, it cannot utilize TensorCore in GPU. This downside degrades the throughput significantly compared to HW-aware algorithms such as ours and Flash Attention (Dao, 2023). However, since the method requires a different RoPE index for every query-key dot product, we cannot easily adopt block sparsity on their method. This will slow down attention computation more than twice because the RoPE re-computation costs twice as much as attention score computation.

In the Figure 7, the latency is measured with our latency measure machine (RTX 4090). StreamingLLM shows OOM over 32k context due to the overhead of the COO sparse matrix. HyperAttention shows invalid kernel parameters over 32k context due to heavily nested tensors in a long context. It uses high-dimensional striding to perform reformer-style token clustering, but the backbone flash attention kernel does not support that high-dimensional striding with a larger tensor.

In the Figure 9, the latency is measured with our latency measure machine (RTX 4090). The machine has about 4GB of VRAM available for the KV cache, excluding model weight and temporary buffers. We limit the size of the CPU offloaded KV cache to 32GB. The tested model is Llama3.1-8B.

**Training Details (Healing HiP in Arxiv).** The HiP healing in Table 3 is done as follows. For the Llama3.1-8B model, after applying HiP, we fine-tune the pretrained model on the Arxiv dataset for 500 steps with AdamW optimizer with learning rate 1e-5, and with batch size 32, LoRA rank 256, and HiP's hyperparameters set to $b_k = 2, b_q = 64, b_sq = 2, b_sk = 1, k = 512, l_d = 3$. We use the Arxiv dataset in Redpajama (Computer, 2023). The inputs are truncated to a maximum of 8192 tokens to speed up the process.

The purpose of this fine-tuning, which we call healing, is to make the model adapt to the slight differences in the activations that appear when the original dense attention layers are replaced with sparse HiP attention. As shown in Table 3, the healed model performs slightly better than the plug-and-play (unhealed) model on LongBench. However, we emphasize that HiP is meant to be training-free, and healing is just an additional option for extra performance, as our method already works near-perfectly without training.

| | Complexity | | Is Training-free? | Is Dynamic Attention? |
|---|---|---|---|---|
| | Time | Space | | |
| **HiP (Ours)** | Log-linear | Linear | ✓ | ✓ |
| SEA (Lee et al., 2023) | Linear | Linear | ✗ | ✓ |
| FlashAttention (Dao, 2023) | Quadratic | Linear | ✓ | ✓ |
| StreamingLLM (Xiao et al., 2024) | Linear | Linear | ✓ | ✗ |
| HyperAttention (sortLSH) (Han et al., 2024) | Near-linear | Near-linear | ✓ | ✓ |
| Reformer (LSH) (Kitaev et al., 2019) | Log-linear | Log-linear | ✗ | ✓ |
| Performer (Choromanski et al., 2022) | Linear | Linear | ✗ | ✓ |
| Cosformer (Qin et al., 2022) | Linear | Linear | ✗ | ✓ |
| Longformer (Beltagy et al., 2020) | Linear | Linear | ✗ | ✗ |

Table 12: Comparison of time and space complexities of various efficient attention methods.

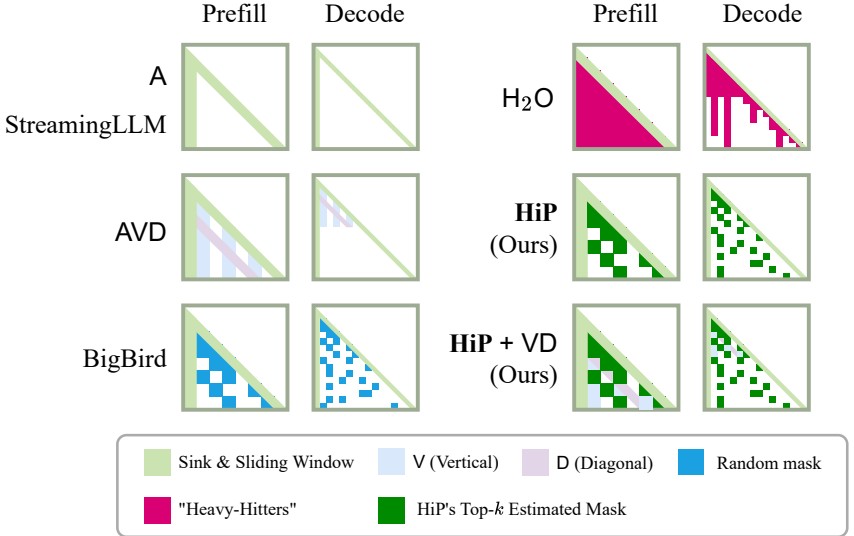

Figure 25: Visualization of sparse attention baselines' attention masks.

# E ADDITIONAL ANALYSIS

## E.1 MORE DISCUSSION ON RELATED WORKS

In Table 12, we compare efficient attention methods that we do not include in our main benchmarks. For each method, we show the time and space complexities, whether it is training-free, and whether it uses dynamic attention. Dynamic attention methods can attend to each other content dynamically rather than using static attention patterns such as a sliding window. Besides HiP, HyperAttention is the only method that satisfies all four criteria, but HyperAttention suffers from substantial performance degradation (see Figure 7). In Figure 25, we conceptually visualize the various sparse attention baselines' attention patterns for extra clarity.

**StreamingLLM (Xiao et al., 2024).** StreamingLLM uses a sliding window attention with an attention sink, which processes the input sequence in linear complexity without resetting the KV cache; they call this process 'streaming.' StreamingLLM introduces the attention sink, which is similar to the global attention token in Longformer (Beltagy et al., 2020), and streams the KV cache using RoPE indexing. However, due to the sliding window, the method cannot perform long-context knowledge retrieval. Therefore, this method cannot utilize the full context, and they do not extend the context window of the model by any amount. Since the method loses the key-value memory as time passes, it cannot take advantage of the Transformer's strength: its powerful past knowledge retrieval ability. Furthermore, since they use a different RoPE indexing for every query-key dot-product, they cannot utilize a MMU, which is a critical speedup factor in modern accelerators.

**HyperAttention (Han et al., 2024).** HyperAttention introduces *sortLSH*, improved version of LSH (Kitaev et al., 2019), to work as plug-and-play. The method uses block sparsity to utilize MMU. It is training-free, has sub-quadratic time complexity (near-linear), and has the ability to potentially access to every past key token, much like our method. However, HyperAttention struggles to recover vanilla performance when replacing most of the layers in the trained model in a training-free manner (see Figure 7).

**Sparse Linear Attention with Estimated Attention Mask (SEA) (Lee et al., 2023).** Inspired by SEA's framework, which introduces linear complexity attention estimation and sparse matrix interpolation, we aimed to improve its efficiency. SEA estimates each query's attention probabilities over the keys with a fixed-size vector, turns it into a sparse mask by selecting the top-k elements, and resizes it; this process is done with linear complexity. However, the method is difficult to implement efficiently due to its extra modules, mainly the estimator and sparse matrix interpolation. Furthermore, the method does not support block sparsity; thus, it cannot utilize the MMU. We were motivated to improve this work drastically by introducing a fused and train-free attention mask estimator, HiP.

## E.2 ANALYSIS OF SUMMARIZING RESULT BETWEEN STREAMINGLLM AND HIP

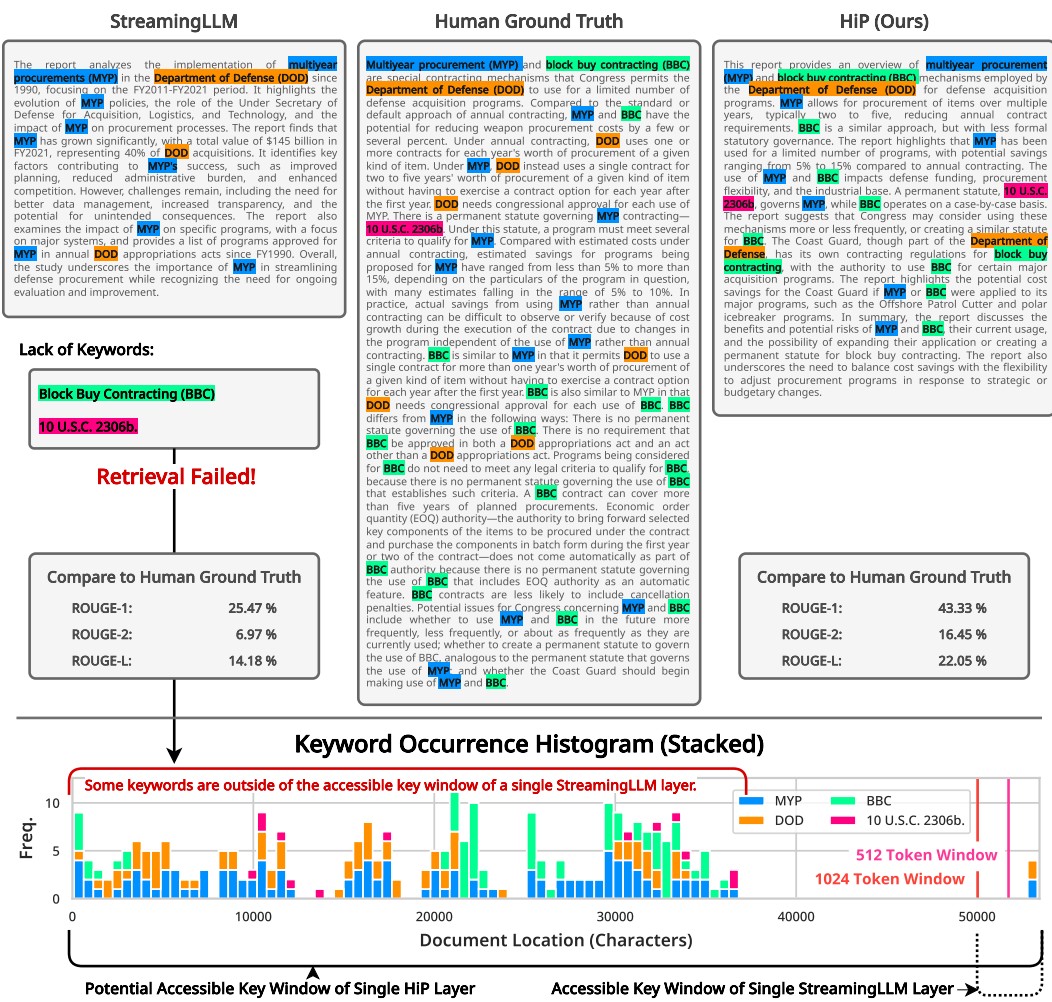

Figure 26: **Summarizing Example of GovReport Dataset from LongBench.** We sample random examples from GovReport summarization results with Qwen1.5-14B.

In Figure 26, we analyze one example generation result from GovReport in LongBench. We pick four important keywords from the human ground truth summary: *MYP*, *BBC*, *DOD*, and *10 U.S.C*

*2306b*. We pick two keywords (*MYP*, *DOD*) that appear in every summary and two keywords (*BBC*, *10 U.S.C 2306b*) that appear only in the ground truth and HiP result. The result clearly shows that StreamingLLM is struggling to gather information beyond its single-layer window size, $k = 1024$. StreamingLLM should be accessible to a much longer distance than $k$ because information keeps exchanging across time dimensions in each layer, like Mistral. In contrast to StreamingLLM, our proposed HiP attention shows successful knowledge retrieval in summary from long-range in-context documents, with the same *plug-and-play* manner. Also, quantitatively, ROUGE-* scores show that the summary generated by HiP is much better in coherence with ground truth than StreamingLLM.

### E.3 HIERARCHICAL ATTENTION MASK PRUNING VISUALIZATION

Figure 27: **Visualization of Hierarchical Attention Mask Pruning.** Yellow indicates a non-zero entry of the attention matrix, and green indicates an empty entry of the attention matrix. We use $k = 512, b_q = 32, b_k = 2, T = 4k$.

In Figure 27, we demonstrate real-world examples of hierarchical attention mask pruning. We sample the Q, K, and V tensors from the first layer of LLaMA2-7B with a random text sample from Wikitext-2. Note that each attention mask in the masking iteration is not the final attention mask. The final attention mask generated by this process is from iteration 3. In an earlier iteration, the sparsity of the mask is low because the group size of blocks is very large (8), so the $8 * 2$ key values are treated as single groups. The attention score of that group is represented by the attention score between the query and the group's first block ($b_k$).

### E.4 ABLATION STUDY ON BLOCK SIZE

Table 13: **Ablation Study on Block Size: Perplexity on WikiText2.** We use LLaMA-2-7B-32k model, $k = 512, dl = 3, T = 12k$. Average shows the average perplexity of each row and column. Red means a bad perplexity score, and green means a good perplexity score.

| | PPL. | $b_q$ | | | | | | |
| | | 1 | 2 | 4 | 8 | 16 | 32 | Avg. |
|---|---|---|---|---|---|---|---|---|
| $b_k$ | 1 | 5.5210 | 5.5172 | 5.4996 | 5.4952 | 5.5064 | 5.5331 | 5.5121 |
| | 2 | 5.6032 | 5.6060 | 5.5863 | 5.5599 | 5.5577 | 5.5901 | 5.5839 |
| | 4 | 5.6316 | 5.6280 | 5.5971 | 5.5872 | 5.5838 | 5.5870 | 5.6024 |
| | 8 | 5.5807 | 5.5928 | 5.5553 | 5.5238 | 5.5100 | 5.5212 | 5.5473 |
| | Avg. | 5.5841 | 5.5860 | 5.5596 | 5.5415 | 5.5395 | 5.5578 | 5.5614 |

We perform an ablation study on block sizes $(b_q, b_k)$ using our method. Block size $b_q$ determines how many queries are grouped into the block during the masking iteration and sparse attention. And block size $b_k$ determines how many keys are grouped. Block size is a really important factor in utilizing MMU (e.g., NVIDIA TensorCore) in modern accelerators. MMU enables matrix multiplication and tensor operations to be performed in single or fewer cycles rather than processing one by one

Table 14: **Ablation Study on Block Size: Attention latency on decoding phase with a single query.** We use same setting with Table 13. Batch size is 96 and $r_m = 1$ and sequence length is 12k. Average shows the average latency of each row and column. Red means bad latency, and green means good latency. The unit of latency is milliseconds. We use RTX-4090 to measure the latency of attention operation. Pytorch shows 20.7215 ms and FlashAttention2 shows 20.2527 ms.

|  | Lat. | $b_q$ |  |  |  |  |  |  |
|---|---|---|---|---|---|---|---|---|
|  |  | 1 | 2 | 4 | 8 | 16 | 32 | Avg. |
| $b_k$ | 1 | 1.6416 | 1.6397 | 1.6397 | 1.6398 | 1.6397 | 1.6401 | 1.6401 |
|  | 2 | 1.3543 | 1.3542 | 1.3530 | 1.3533 | 1.3535 | 1.3534 | 1.3536 |
|  | 4 | 1.2312 | 1.2311 | 1.2296 | 1.2305 | 1.2322 | 1.2306 | 1.2309 |
|  | 8 | 1.2836 | 1.2842 | 1.2830 | 1.2827 | 1.2830 | 1.2835 | 1.2833 |
|  | Avg. | 1.3777 | 1.3773 | 1.3763 | 1.3766 | 1.3771 | 1.3769 | 1.3770 |

Table 15: **Ablation Study on Block Size: Attention latency on decoding phase with multiple queries (Speculative Decoding).** We use same setting with Table 14. We use 32 query tokens to mimic speculative decoding scenarios. PyTorch shows 31.0632 ms and FlashAttention2 shows 20.3378 ms.

|  | Lat. | $b_q$ |  |  |  |  |  |  |
|---|---|---|---|---|---|---|---|---|
|  |  | 1 | 2 | 4 | 8 | 16 | 32 | Avg. |
| $b_k$ | 1 | 152.4423 | 77.2624 | 39.2077 | 20.1341 | 10.4925 | 5.5461 | 50.8475 |
|  | 2 | 85.3140 | 43.8207 | 22.7708 | 12.1814 | 6.4987 | 3.5056 | 29.0152 |
|  | 4 | 54.2530 | 27.7025 | 14.3960 | 7.9660 | 4.5839 | 2.6156 | 18.5862 |
|  | 8 | 41.5079 | 21.3036 | 11.5369 | 7.4481 | 5.0634 | 3.0337 | 14.9823 |
|  | Avg. | 83.3793 | 42.5223 | 21.9779 | 11.9324 | 6.6596 | 3.6753 | 28.3578 |

using floating point multiplication and addition. This kind of accelerator trend leads to mismatching of wall-clock latency and FLOPs in modern hardware. Therefore, we check the performance and latency trade-off among grouping queries and keys by block size $b_q, b_k$.

In Table 13, we show that perplexity gets better as $b_q$ increases while it gets worse as $b_k$ increases. It is not intuitive that increasing $b_q$ shows better perplexity than before because they lose the resolution across the query dimension in attention mask estimation. However, the result shows that more block size (more averaging) across the query (time) dimension shows better performance. In contrast to this observation, $b_k$ works as expected, like that less resolution in key (past knowledge or memory) dimension leads to worse performance.

This phenomenon makes our method speed up without any performance loss, even achieving better performance. In Table 14, we measure the micro latency benchmark of our attention operation during the decoding stage, which feeds a single query into the attention operator. With a single query, we cannot utilize the MMU fully because, during sparse attention and attention score estimation in masking iteration, we cannot matrix multiply between the $Q$ group and $K$ group. We have a single query vector; therefore, we need a vector-matrix multiplier instead of matrix-matrix multiplication, which is the main key feature of MMU. However, in Table 15, we measure the micro latency benchmark of our attention operation during the decoding stage with a speculative decoding strategy, which feeds multiple queries into the attention operator. We feed 32 query vectors within a query dimension in the input tensor; therefore, now we can utilize a matrix-matrix multiplier in an MMU. With these multiple queries and MMU utilization, our method could achieve a 10.23 times speedup on 12k sequence length compared to PyTorch naive implementation (using bmm).

We use $b_q = 32, b_k = 2$ by default, according to the ablation study across the block sizes. We choose $b_q = 32$ because increasing $b_q$ leads to better latency and perplexity. However, we stopped increasing $b_q$ by 32 because the current modern GPU, especially the NVIDIA Ampere series, usually does not support matrix-matrix multiplication larger than 32. And maybe in the future, some variants will support larger matrix multiplication, just like Google TPU. However, larger blocks need more register allocation for block masking and address calculation. Therefore, considering implementation limitations, we think there is no benefit to increasing $b_q$ infinitely. Also, from a performance perspective, we do not think this trend will keep over $b_q > 32$. We choose $b_k = 2$ because latency speedup from $b_k = 1$ to $b_k = 2$ is huge respect to perplexity loss.

Additionally, we measure the latency with $r_m = 1$, which means without mask caching. Therefore, this speedup will be amplified with $r_m$ in a practical setting.

## E.5 ABLATION STUDY ON DENSE LAYER CHOICE

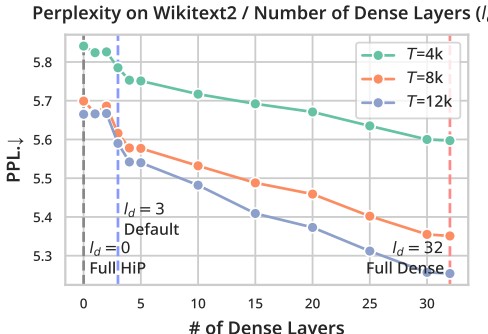

Figure 28: **How Many Layers Should be Remained as Dense Layer** $l_d$**?** We use $l_d = 3$ as the default value. The Y-axis means how many first layers of the Transformer model are kept as dense attention layers rather than replaced with HiP. We use Llama2 7B 32k for PPL evaluation on Wikitext2

We do not replace the first few layers ($l_d$) of the Transformer because the first few layers tend to have dense attention probabilities rather than sparse attention probabilities. This phenomenon is well described in previous results (Ribar et al., 2013). The first few layers exchange information globally and uniformly across the tokens.

Therefore, in Figure 28, we perform an ablation study on how many first layers should remain as dense attention. We observe that the first layers are kept as dense attention and then more perplexity. In other words, if we replace the original transformer block with HiP attention, we could minimize the performance degradation. However, for maximum practical speedup, we want to minimize the number of dense layers for the experiment. Therefore, we run the ablations study on different numbers of dense layers, and we choose 3. For 2 to 3, the performance (perplexity) improvement is maximized. In conclusion, the practical effect of dense layers on latency is minimal because the number of dense layers (e.g., 3, 4) is small compared to the number of HiP layers (e.g., 29, 36). We show that we could achieve practical end-to-end speedup compared to baselines in Figure 8.

## E.6 ABLATION STUDY ON REPRESENTATIVE TOKEN LOCATION

In the theoretical analysis section, we have mathematically proved that selecting the middle token as the representative token of a section guarantees better performance than random selection. In this subsection, we perform a brief ablation study on the location of the representative token to see if this claim is true. In the following experiment, we measured the perplexity values of the Llama3.1 8B model for various representative token locations. We used the PG19 dataset as the input for the model.

Table 16: **Ablation Study of Representative Token Location.** Perplexity is measured on the first 3.2M tokens in the PG19 dataset with a 128k context length. We use Llama3.1-8B for evaluation.

| Position | First | 0.2 | 0.4 | Middle | 0.6 | 0.8 | Last |
|---|---|---|---|---|---|---|---|
| PG19 Perplexity | 8.851 | 8.752 | 8.691 | **8.685** | 8.818 | 8.934 | 8.812 |

Table 16 shows the experimental results. As can be seen in the table, the perplexity value is minimized when the middle token is selected as the representative token. This closely matches our expectations, and it asserts that our theoretical analysis of the representative token location is valid. Therefore, this ablation study justifies the selection of the middle token as the representative token.

### E.7 Discussion about KV Cache Eviction and Compression Strategy

We think the KV eviction and compression strategy is an orthogonal method to our proposed HiP method, and we can cooperate with KV cache strategies. Users can use sparse linear attention methods like ours with a KV eviction strategy. If the KV eviction strategy is careful enough, our method should retain the same performance as vanilla attention.

Also, the typical retention ratio ($512/32000 = 1.6\%$) of our method is much more extreme than state-of-art eviction strategies (10 to 20% (Zhang et al., 2023)). Moreover, the KV eviction strategy loses information permanently, which should be a problem. We think we can solve the memory pressure from the KV cache should be solved with the memory hierarchy of the computer system. NVMe storage should be large enough to store everything. We think KV eviction has limitations because we cannot estimate which information will be important in the future. Therefore, we should store every important piece of knowledge somewhere in our memory. During the storage of the KV cache, we can utilize a partial KV cache eviction strategy.

We believe KV cache offloading is the future direction to tackle the memory limitation of the attention mechanism, as we proposed in the main section.

### E.8 Discussion about Speculative Decoding

We think that HiP can cooperate with many other speculative decoding strategies orthogonal (Leviathan et al., 2023; Miao et al., 2024; Fu et al., 2024; Cai et al., 2024) because they are working with the output of LLM, which is logits. Also, the speculative decoding method tries to decode multiple queries simultaneously to verify the speculative generation candidates. This characteristic of speculative decoding will take advantage of additional speedup with the large batches in HiP.

### E.9 Remaining Challenges in HiP and Potential Solutions

Although HiP successfully replaces the existing vanilla attention, there is room for improvement as follows:

- While HiP outperforms the baselines and performs similarly with Flash Attention in long context evaluation, HiP still underperforms Flash Attention with smaller $k$ (lower compute budget).
- As shown in Appendix E.5, HiP uses a few layers ($l_d$) of quadratic dense attention, which could lead to a higher cost as the context length increases.
- Since HiP enforces every row in the attention matrix to have the same sparsity, this is not optimal to handle dynamic sparsity of attention (Ribar et al., 2013; Lee et al., 2023).

As shown in Figure 2, because HiP discards the bottom chunks for unselected branches, it is impossible to select tokens from the actual top-$k$ set if they happen to be within those discarded chunks. We refer to this as 'branch early termination' in HiP, and addressing this issue could help resolve above improvement points. Therefore, we propose two possible research directions that tackle the branch early termination problem while also enabling dynamic sparsity: (1) an ensemble hierarchical top-$k$ approximation and (2) an improved tree traverse strategy.

First, for the ensemble hierarchical top-$k$ approximation, we generate multiple HiP masks by injecting randomness into branching decisions during a specific iteration and create a final ensemble mask by aggregating indices from these masks. The ensemble method demonstrates that applying different branching in a given iteration can enhance inference accuracy and indicates the potential to replace the dense layers $l_d$ in HiP with comparable performance, as shown in Appendix C.5. Moreover, Figure 24 illustrates how the ensemble method enables dynamic sparsity across layers and heads, addressing the limitation of uniform sparsity in HiP.

Second, we could explore more diverse traversal methods, rather than strictly relying on binary branching and selecting the top-$k$ tokens at each iteration. In Appendix C.5, we examine the effectiveness of applying ensemble techniques to HiP masks with slight randomness. However, this approach incurs additional computational costs due to the oversampling of the mask, which can be quite expensive. Therefore, to achieve a similar effect, we could diversify the tree branching beyond

the binary structure, similar to an n-beam search. Another potential solution specifically tailored to HiP is to apply multi-branching in a certain iteration and oversample the chunks in subsequent iterations, maintaining multiple paths until the end. By doing so, the final iteration would include more than $k$ candidates, resolving the branch early termination issue and allowing us to decide how many tokens to select for dynamic sparsity across the layers.

## E.10 Unique GPU Resource Demand Pattern of HiP Compared to Flash Attention

Our novel HiP attention is quite different from any other attention implementations. We do not heavily rely on ALUs (floating operations) like Flash Attention. Also, we do not heavily rely on memory bandwidth like previous sparse attention methods like $H_2O$ (it has to store attention scores in global memory). The HiP relies on thread resources because of the top-$k$ operator in between HiP iterations. Moreover, we have highly complex algorithms compared to traditional GPU applications like graphics shaders. Due to this highly complex algorithm, we heavily rely on thread resources, even if we are heavily optimized with MMU for attention score sampling.

Thanks to the reduced complexity of HiP, $O(T \log T)$, we are winning every configuration with long context prefill and decoding. Since the decoding phase is highly memory-dependent for Flash Attention, we also always win in most practical context lengths. However, we sometimes lose if Flash Attention is way too much faster because of the trillions of floating operation specifications of high-end server-grade GPUs. Moreover, Flash Attention 2 and 3 utilize special floating point valuation resources in GPU, especially on H100; FA3 is way too much faster in another setting. Therefore, we are starting to lose in short context ($T$=32k) to FA2 and FA3 because of the speedup of Flash Attention on H100.

This phenomenon is disappointing to us. Therefore, we try to investigate why HiP in H100 is so slower than others, even compared to consumer-grade GPUs such as RTX 4090. We think the high demand for CUDA thread resources is due to our internal sorting algorithm. Since we must remain top-$k$ blocks in every iteration, we must perform $O(k \log k)$ cost sorting operation $O(\log T)$ times. Therefore, as $k$ grows, we are staving to allocate worker thread for score comparison.

Table 17: **Prefill Latency Speedup of HiP by Removing Sorting on RTX 4090 and H100 with different $k$ and $T$.** The time unit is milliseconds.

| Device | 4090 | | | | | | H100 | | | | | |
|---|---|---|---|---|---|---|---|---|---|---|---|---|
| Context Length | 32k | | | 128k | | | 32k | | | 128k | | |
| Prefill Latency | w/o Sort | w/ Sort | Speedup | w/o Sort | w/ Sort | Speedup | w/o Sort | w/ Sort | Speedup | w/o Sort | w/ Sort | Speedup |
| Flash Attention | 56.0 | | | 855.2 | | | 24.05 | | | 430.79 | | |
| HiP k=512 | 16.48 | 19.33 | **1.173** | 85.62 | 103.13 | **1.204** | 16.55 | 19.80 | **1.196** | 86.65 | 108.38 | **1.251** |
| HiP k=1024 | 26.96 | 33.70 | **1.250** | 150.90 | 190.98 | **1.266** | 27.56 | 36.30 | **1.317** | 153.41 | 210.78 | **1.374** |
| HiP k=2048 | 44.12 | 62.27 | **1.411** | 263.60 | 386.97 | **1.468** | 43.74 | 69.54 | **1.590** | 260.22 | 434.30 | **1.669** |

Table 18: **Technical Specifications of 4090 and H100.** We put a comparison of prefill latency compared to RTX 4090.

| | Rel. CUDA Core | Rel. TFLOPs | Rel. Mem. Bandwidth | Rel. Clock Speed | Rel. HiP Speedup | Rel. FA2 Speedup |
|---|---|---|---|---|---|---|
| RTX 4090 | 1.00 | 1.00 | 1.00 | 1.00 | 1.00 | 1.00 |
| H100 | 1.00 | 3.66 | 3.32 | 0.71 | 0.89 | 1.99 |

We want to show that the elimination of the sorting operation will speed up our top-$k$ estimation. To do so, we replace sorting with an identity function. So, in this version, we always select the first half blocks to pass the next HiP iteration. As shown in Table 17, eliminating sorting speed up our HiP significantly. In 4090, we could observe 46.8% speedup, and in H100, we could observe 66.9%. We can see the high relation between (CUDA cores + relative clock speed) and HiP speed as shown in Table 18. So, we will try to investigate removing the sorting and replacing it with some approximations for more practicality of HiP.

This characteristic is quite good for cost-effectiveness. Nvidia does not usually reduce CUDA cores on consumer-grade GPUs; therefore, we could achieve the same speed as H100 while reducing GPU costs more than ten times. Even in server-grade GPUs, there are some great cost-effective alternatives. For example, L40s has more ALU than 4090 and the same amount of CUDA core. Therefore, L40s will offer A100-level linear layer computation while offering H100-level attention,

which is way more after than flash attention on L40s. In the L40s, flash attention will be extremely slow, like in A100 and 4090, because they have similar FLOPs due to the price tag. We wanted to test L40s during submission, but unfortunately, we could not find any possible option to get L40s.

The lower-grade GPUs often struggle with the small size of VRAM. However, the tiny memory of lower-grade GPU is not a problem with our method due to the powerful KV cache offloading feature without decoding throughput degradation. We have already shown that we can serve 64K context length with a single RTX 4090 card, and if you put 8 of them together, then we can serve around 512K context length with high decoding throughput. For example, the tinygrad company offers 8x 4090 workstations with only 40,000$ (tinygrad, 2024) *(we are not them, just for clarification)*. The price is almost similar to a single H100 card, but you can serve 512K context length with more than twice TFLOPs! This means that if you have two nodes of that machine, you can actually run Google Gemini class (Google, 2024) long context LLM in the home. And if the tensor parallelism is linearly scaled with two nodes, you can decode 1,527 tokens with 64k context length. Since our method is almost a logarithm scale with context length during decoding, we can expect to decode around 1K tokens per second with a one million context length. So, we are really excited to introduce our KV cache offloading feature with HiP in many practical aspects.

## F  POTENTIAL NEGATIVE SOCIAL IMPACT

In this paper, we do not perform a careful investigation on LLM alignment performance with HiP. There is the potential that HiP might break the LLM safety guard. However, as far as we observed during the experiment, the HiP could preserve most of the behavior of trained LLM. Furthermore, we could always adopt the third-party LLM safety guard model such as LLaMA Guard (Inan et al., 2023).

