# OpenReview forum: "A Training-Free Sub-quadratic Cost Transformer Model Serving Framework with Hierarchically Pruned Attention"
_ICLR.cc/2025/Conference — ICLR 2025 Poster_

### Official Review · Reviewer_KprR · 2024-10-27

**Soundness:** 2
**Presentation:** 3
**Contribution:** 3
**Rating:** 6
**Confidence:** 3

**Summary:**

The paper presents HiP (Hierarchically Pruned Attention), a novel approach aimed at reducing the time and space complexity of the attention mechanism in Large Language Models (LLMs). HiP leverages the observation that tokens close together tend to have similar attention scores to estimate the top-k key tokens for a given query on the fly. This results in sub-quadratic time complexity (O(T log T)) and linear space complexity (O(T)), where T is the sequence length.  Experimental results show that HiP significantly reduces both prefill and decoding latencies while maintaining high performance on benchmarks such as LongBench.

**Strengths:**

1. HiP does not require retraining, making it easy to apply to pre-trained models. In addition, by reducing the time complexity to O(T log T) and space complexity to O(T), HiP enables the use of longer context lengths without the associated quadratic cost.
2. Through KV-cache offloading, HiP optimizes GPU memory usage, which is especially beneficial for large models.
3. HiP can dynamically adjust to different sequence lengths, making it suitable for a variety of tasks that involve long contexts.

**Weaknesses:**

1. The effectiveness of HiP is contingent upon the presence of "attention localities," which might vary across different LLM architectures or tasks. The robustness of this operation deserves more discussions.
2. While HiP reduces latency in the decoding phase, the iterative pruning process might introduce overhead in the initial stages. How does the proposed method balance these two parts.
3. The authors have already shown improvements for certain sequence lengths, while it is expected to thoroughly explore how HiP scales to extremely long sequences or very large models. The current configuration on the sequence length (128k) and the model size (8B) are relatively small for a paper studies on the efficient decoding. Alternatively speaking, the scalability of this method deserves more experiments to support.

**Questions:**

See the weakness for details.

---

> ### Author Response · Authors · 2024-11-14
> **Author Rebuttal to the Reviewer KprR (1/2)**
>
> Thank you for taking the time to review our work and for providing insightful and supportive feedback. We have addressed your concerns and questions below.
>
> ---
>
> **Weaknesses:**
> > W1. The effectiveness of HiP is contingent upon the presence of "attention localities," which might vary across different LLM architectures or tasks. The robustness of this operation deserves more discussions.
>
> We tested attention locality in various tasks and pretrained models. Due to space layout, we omitted the empirical results from various models, but we updated the revision to have them now, so please **refer to Figure 11-16** in Appendix A. The updated figures in Appendix A now include empirical data showing that attention locality is prevalent across various LLM architectures. As suggested, we revised the section more, so please give us more comments if you have any concerns.
>
> > W2. While HiP reduces latency in the decoding phase, the iterative pruning process might introduce overhead in the initial stages. How does the proposed method balance these two parts.
>
> This is a misunderstanding. **Our method does not cause latency degradation in both the prefill (initial) stage and the decoding stage**. We always report the prefill and decode attention latencies (including masking and sparse attention) in Tables 1, 2, and 3 in the bottom row. The sorting overhead is negligible compared to Flash Attention, as shown in Figure 8. As we reported in Figure 7, our method outperforms FlashAttention2 in terms of prefilling and decoding attention latency.
>
> Please be aware our method provides latency speedup on every phase of the Transformer model in a plug-and-play manner without any architecture modification or further training.
>
> ---
> Continue to the following comment.

---

> ### Author Response · Authors · 2024-11-14
> **Author Rebuttal to the Reviewer KprR (2/2)**
>
> > W3. The authors have already shown improvements for certain sequence lengths, while it is expected to thoroughly explore how HiP scales to extremely long sequences or very large models. The current configuration on the sequence length (128k) and the model size (8B) are relatively small for a paper studies on the efficient decoding. Alternatively speaking, the scalability of this method deserves more experiments to support.
>
> This is a misunderstanding; we have already performed extremely long (512k) latency reports, as shown in Table 6. We already showed the long context extendability in Table 10. To sum up, with our method, every LLM can extend their context window without being bottlenecked by memory and latency.
>
> Our method not only gains speedup in extremely long contexts, but also in shorter contexts such as 32k and 128k which are used in LongBench and other benchmarks. In Tables 1, 2, and 3, we already use appropriate context length to measure attention speedup (Longbench: T=32k, Passkey, RULER: T=128k). Therefore, we could show the speedup with the tested dataset context length. We think the 128k context window size of Llama3.1 is quite long enough to show the speedup of HiP.
>
> Moreover, in the KV cache offloading experiment of Table 6, we try to extend the context length up to a 512k context window. Therefore, we believe we have already shown the effectiveness of the HiP framework in the long context in terms of latency and scalability.
>
> Regarding performance scalability, we have already verified that the original LLM's long-context ability was recovered using RULER. Therefore, we can expect HiP to be scalable in an extremely long context model in the future, too (if you scale the cost budget appropriately). We hope this resolves the concerns about HiP's performance scalability.
>
> To show the context extendability of HiP, we used Gemma 2 in Appendix Section C.4 by extending its pretrained context window 16 times longer using HiP with SelfExtend. This shows that we think our method is also scalable when we extend the context window extremely.
>
> **Table 1. Updated Latency Results.**
> Prefill Latency (4090, ms) | 64k | 128k
> -|-|-
> FA2	| 219.430 | 893.360
> HiP (k=512) | 83.038 | 188.641
> HiP (k=1k)	| 155.583 | 364.060
> HiP (latest, low-preset) | 139.761 | 311.747
> HiP (latest, mid-preset) | 201.032 | 439.410
>
> **Table 2. Passkey Result (Llama 3.1 8B, Pretrained on 128k)**
> Passkey (Acc.)          | 32k	| 64k	| 128k	| 256k	| 512k	| Average.
> -|-|-|-|-|-|-
> HiP (no-extend, 1k)	    | 100.0	| 95.0	| 85.0	| 25.3	| 0.0	| 61.1
> HiP (latest, low-preset)| 100.0	| 100.0	| 90.0	| 97.5	| 91.9	| 95.9
> HiP (latest, mid-preset)| 100.0	| 100.0	| 97.5	| 98.8	| 98.8	| **99.0**
>
> **Table 3. InfiniteBench Result (Llama 3.1 8B, Pretrained on 128k).**
> InfiniteBench `En.MC` Acc.   | 128k	| 256k	| 384k
> -|-|-|-
> FA2	                    | **0.629**	| 0.550	| 0.550
> HiP (latest, low-preset)| 0.590	| 0.568	| 0.563
> HiP (latest, mid-preset)| 0.611	| **0.638**	| **0.638**
>
> **Table 4. InfiniteBench Result (Exaone 3 7.8B, Pretrained on 4k).**
> InfiniteBench `En.MC` Acc.   | 4k    | 8k    | 16k    | 32k    | 64k    | 96k    | 128k
> -|-|-|-|-|-|-|-
> FA2	| **0.3362**	| -	| -	| -	| -	| -	| -
> HiP (latest, mid-preset)| 0.3188 | 0.3144 | **0.3362** | **0.3668** | **0.3450** | **0.3843** | **0.3930**
>
> To further support context extendability, in our latest research after submission, we aim to enhance the ability of HiP to extend the context while maintaining the overall framework and latency, as shown in Table 1. We increased flexibility in our framework, so now the users freely control each iteration (how many chunks should be selected, how to break down each chunk, and how to select the representative tokens). We could increase long-context performance by searching appropriate iteration configurations. However, since the latest development is still under construction, we leave this for future work (of course, we will go to open source everything, so you can check our source code in real-time if you are wondering how we built it). We could achieve 99.0 percent passkey accuracy result in 512k context length with Llama3.1 8B (Table 2) and even outperform FA2 in InfiniteBench 384k (Table 3). As shown in the table (Table 4), we could increase performance improvement by increasing context length more than 32 times than the pre-trained.
>
> Therefore, we think a hierarchically pruned attention framework is a promising research direction and has great value to share with the research community to stimulate the following creative studies by community members who are trying to build extremely long context models using various approaches.
>
> ---
>
> Thank you once again for taking the time to review our work and provide insightful feedback.
>
> We hope to address and resolve your concerns during the discussion period. If our rebuttal successfully clarifies your questions, we kindly ask you to consider updating your score accordingly.

---

> ### Comment · Reviewer_KprR · 2024-11-26
> **Reply to authors**
>
> Thanks for the response by the authors, which makes the performance reported by this paper more convincing. Given the original score I gave,  I decide to keep my score.

---

> > ### Author Response · Authors · 2024-11-27
> >
> > Thank you for taking the time to review our work and rebuttal. If there are any additional concerns that we could address or further clarifications we could provide, we would be glad to hear them.
> >
> > Again, thank you for providing an insightful and constructive review.

---

### Official Review · Reviewer_NLBm · 2024-11-03

**Soundness:** 3
**Presentation:** 4
**Contribution:** 2
**Rating:** 5
**Confidence:** 4

**Summary:**

This paper proposes a novel, training-free attention mechanism called Hierarchically Pruned Attention (HiP) to accelerate the serving of pre-trained Transformer-based large language models (LLMs) for long-context tasks. HiP addresses the computational challenges posed by the quadratic time and space complexity of standard attention mechanisms in handling long sequences.

**Strengths:**

1. HiP's training-free nature eliminates the need for costly retraining of large LLMs, making it readily applicable to existing models.
2. HiP's hierarchical pruning significantly reduces the computational complexity of attention from quadratic to log-linear, enabling efficient handling of long sequences.
3. KV cache offloading effectively addresses the memory limitations of GPUs, allowing HiP to scale to much longer context lengths.
4. The paper provides thorough experimental results on diverse benchmarks, showcasing HiP's effectiveness in terms of speedup, performance preservation, and context length extension.
5. The paper includes a theoretical analysis of HiP's hierarchical pruning algorithm, providing insights into its superior performance compared to random key selection.

**Weaknesses:**

1. HiP's effectiveness relies on the assumption of attention locality. While this assumption generally holds, there might be cases where it's violated, potentially impacting performance.
2. HiP enforces the same sparsity across all rows of the attention matrix. Exploring mechanisms for dynamic sparsity, where sparsity varies based on the input, could further enhance efficiency and performance.
3. The implementation and optimization of HiP, particularly the KV cache offloading, are tailored for specific hardware platforms (e.g., RTX 4090). Performance and optimal configurations might vary across different hardware accelerators.
4. The paper doesn't specifically address potential LLM alignment issues that might arise from applying HiP. Further investigation is needed to ensure HiP's safety and robustness in practical deployments.

**Questions:**

1. How does the choice of chunk sizes in HiP's hierarchical pruning affect the trade-off between accuracy and efficiency?
2. What strategies can be employed to further optimize KV cache offloading, such as using different memory tiers or compression techniques?
3. How well does HiP integrate with other efficiency techniques, such as quantization or model pruning, to further improve serving efficiency?
4. Can HiP be effectively applied to other Transformer architectures beyond the specific LLM model used in the paper?

---

> ### Author Response · Authors · 2024-11-14
> **Author Rebuttal to the Reviewer NLBm (1/2)**
>
> Thank you for taking the time to review our work and for providing insightful feedback. We have addressed your concerns and questions below.
>
> ---
>
> **Weaknesses:**
> > W1. HiP's effectiveness relies on the assumption of attention locality. While this assumption generally holds, there might be cases where it's violated, potentially impacting performance.
>
> Please see Appendix A for more detailed discussions about attention locality with more theoretical analysis and empirical results. We have already tested various tasks (please refer to Fig. 11) to show that attention locality exists in general for a wide range of tasks. Plus, we have added empirical results from additional models, namely Qwen2 7B and Exaone 3.0 7.8B, to the figures in Appendix A.3 in the revision. (Please check the updated Fig. 11-16).
>
> > W2. HiP enforces the same sparsity across all rows of the attention matrix. Exploring mechanisms for dynamic sparsity, where sparsity varies based on the input, could further enhance efficiency and performance.
>
> In this study, we are focused on proposing and validating the HiP (Hierarchically Pruned attention) framework. However, as we discussed in Appendix E.9, we agree that dynamic sparsity is a promising approach for future research.
>
> In fact, our implementation of HiP already has support for tweaking the amount of sparsity on-the-fly, so future researchers can easily utilize our implementation for their own dynamic sparsity version of HiP. The block sparse attention kernel receives the non-zero entries' indices and the number of non-zero entries (commonly referred to as `nnz` in many sparse matrix data structure implementations) for each row. By controlling the `nnz`, the user can control the cost of the block sparse attention very easily. However, in this study, we opt for the simplicity of design since dynamically determining the sparsity based on the input is out of the scope of this study.
>
> > W3. The implementation and optimization of HiP, particularly the KV cache offloading, are tailored for specific hardware platforms (e.g., RTX 4090). Performance and optimal configurations might vary across different hardware accelerators.
>
> This is a factual misunderstanding -- We have not tailored our method to specific hardware such as the 4090. In Table 6, we benchmark our KV Offloading on both 4090 and A100. Also, in Appendix E.10, we compare HiP's latency on H100 and analyze its hardware resource demand patterns. As discussed there, HiP utilizes the GPU's thread resources more heavily instead of floating point arithmetic than the matmul-heavy FlashAttention2. However, this does not mean that our kernel only works efficiently in a specific HW.
>
> We understand that a HW difference may lead to non-optimal performance with the default hyperparameters. To alleviate this problem, in our implementation, we have added a built-in autotuner of kernel hyperparameters (e.g., block sizes (*not $b_k$, $b_q$, but the memory tile block size*), number of warps, number of registers, etc.). Since we built an autotuner using grid search, we will add a pre-built library of our HiP kernels in future code releases.
>
> > W4. The paper doesn't specifically address potential LLM alignment issues that might arise from applying HiP. Further investigation is needed to ensure HiP's safety and robustness in practical deployments.
>
> We believe that this is out of the scope of our research. Although we agree that LLM safety is also a serious problem, we are proposing an unrelated orthogonal approach, an efficient attention mechanism, in this work.
> Since our extensive empirical analysis shows we can recover most LLM behavior, we expect the HiP applied model also to be aligned with guidelines well like the original LLM.
>
> **Questions:**
> > Q1. How does the choice of chunk sizes in HiP's hierarchical pruning affect the trade-off between accuracy and efficiency?
>
> We will assume that the term "chunk size" you mentioned refers to the block sizes $b_k$ and $b_q$. In Appendix E.4, we perform an ablation study on block sizes in HiP and discuss the results. We find that the perplexity gets better as $b_q$ increases, whereas it gets worse as $b_k$ increases. Please refer to Appendix E.4 for more details.
>
> > Q2. What strategies can be employed to further optimize KV cache offloading, such as using different memory tiers or compression techniques?
>
> Please see Appendix E.7 and E.10 for a discussion of the possibility of additional memory tiers as a future direction. To summarize the discussion in that section, we believe we can add additional cache offloading tiers to the two-level hierarchy between GPU VRAM and CPU DRAM in order to further relax the memory pressure under extremely long sequences. We are looking forward to extending our offloading system to contain various levels, such as NVMe local storage and network storage, in future work.
>
> Regarding compression techniques, we will answer this in more detail in Q3.
>
> ---
> Continue on the following comment.

---

> ### Author Response · Authors · 2024-11-14
> **Author Rebuttal to the Reviewer NLBm (2/2)**
>
> > Q3. How well does HiP integrate with other efficiency techniques, such as quantization or model pruning, to further improve serving efficiency?
>
> Our method is completely orthogonal to many other efficiency techniques, as HiP is a simple drop-in replacement for the Flash Attention call in whatever framework the user is operating in, such as vLLM, SGlang, and Huggingface Transformers.
>
> We have already provided vLLM and SGlang integration in the supplementary files that are built with interoperability with other efficiency techniques in mind, such as Paged Attention, model quantization (GPTQ, AWQ), speculative decoding, prompt caching, structured generation, and more. Furthermore, our method is compatible with KV cache compression or eviction methods as long as they are still calling the regular attention mechanism internally.
>
> We are looking forward to contributing to the open source, especially vLLM, and SGlang, so our method can be easily integrated with future efficiency techniques that are compatible with vLLM and SGLang, which is the most widely used LLM inference framework.
>
> > Q4. Can HiP be effectively applied to other Transformer architectures beyond the specific LLM model used in the paper?
>
> Yes, HiP can be effectively applied everywhere Flash Attention is used and attention locality exists. We already applied HiP to various Transformer models such as LLM (Llama3.1, Llama2, Gemma2) and LMM (Llava1.6, section C.1) in the manuscript and the appendices.
>
> ---
>
> Thank you once again for taking the time to review our work and provide insightful feedback.
>
> We hope to address and resolve your concerns during the discussion period. If our rebuttal successfully clarifies your questions, we kindly ask you to consider updating your score accordingly.

---

> ### Comment · Reviewer_NLBm · 2024-11-26
>
> I thank the authors for the explanations. The explanations given here are much better and I suggest the authors to make revisions in the main manuscript accordingly. I have raised my score.

---

> > ### Author Response · Authors · 2024-11-26
> >
> > Thank you for taking the time to review our work. We appreciate your positive remarks about our explanations and are glad to hear that our rebuttal has addressed your concerns. We will ensure that the revisions to the main manuscript reflect the clarity and detail you found helpful in our responses.
> >
> > We are also grateful for your decision to raise the score. If there are any additional concerns that we could address or further clarifications we could provide, we kindly ask for the opportunity to do so in the hope that it may justify raising the score further to more accurately reflect the contributions of our work.

---

### Official Review · Reviewer_T2wK · 2024-11-03

**Soundness:** 4
**Presentation:** 4
**Contribution:** 3
**Rating:** 8
**Confidence:** 3

**Summary:**

The paper introduces Hierarchically Pruned Attention (HiP) which reduces time and space complexity of the attention mechanism. HiP exploits "attention locality" to estimate top-k key tokens (w/ theoretical justifications as insights) for a given query and does so in a hardware aware way. Moreover, the paper introduces a further KV cache offloading steps which reduces space complexity further.

**Strengths:**

* The method being training-free means it can be used as a drop-in to already trained models.
* The paper does careful complexity analysis of its claims but strikes a balance on introducing information in a way to aid presentation (thinking of the informal theorem) while still being rigorous later.
* It is extremely valuable to have code examples and implementation released.
* The paper is extremely comprehensive when understanding its metrics across different hardware and comparing with the many version of flash attention.

**Weaknesses:**

* It would be great to have the long context benchmarks also for different models -- gemma and mistral are both open source and around similar sizes.

**Questions:**

* Gemma 2 uses this mix between sliding window and normal attention. it would be great to understand if there are any degradation on non-llama architectures.
* The method does show that MMLU is not degraded when using this method. It would be interesting to see this for a broader set of metrics if possible.

---

> ### Author Response · Authors · 2024-11-15
> **Author Rebuttal to the Reviewer T2wK (1/2)**
>
> Thank you for taking the time to review our work and for providing insightful and supportive feedback. We have addressed your concerns and questions below.
>
> ---
>
> **Weaknesses:**
> > W1. It would be great to have the long context benchmarks also for different models -- gemma and mistral are both open source and around similar sizes.
>
> Please refer to section C.4. You can find additional Transformer models such as Llama-2 and Llava1.6 in our appendix sections E.4, E.4, and C.1. We already have results with Gemma 2 with a training-free context extrapolation method to test the 128k context window, which is 16 times larger than its pretrained context window. Here is a summarized result of Gemma 2 from Appendix Table 10.
>
> **Table 1. Gemma 2 Results from Appendix Table 10**
> Perplexity (Wikitext2) | 8k | 64k
> -|-|-
> FA2 | 11.40 | 363.02
> HiP | 12.25 | 11.54
>
> As for our model selection, we choose to use Llama3.1 8B to test our method in various tasks and baseline methods. Since our benchmark suite is quite huge (Passkey, Perplexity, RULER, LongBench, LMMS-Eval, MMLU), we could not perform all of the experiments on as many models as we would have liked.
>
> In our experience, our method generally shows similar performance with Llama 3.1 in other well-known LLMs because the attention locality is prevalent across various LLM architectures, such as Qwen2 and Exaone3. We have updated the figures in Appendix A.3 to include empirical results across various LLM models, which shows that attention locality exists in other various models as well. (Please check the updated Figures 11-16).
>
> ---
>
> Continue on to the next comment.

---

> ### Author Response · Authors · 2024-11-15
> **Author Rebuttal to the Reviewer T2wK (2/2)**
>
> **Questions:**
> > Q1. Gemma 2 uses this mix between sliding window and normal attention. it would be great to understand if there are any degradation on non-llama architectures.
>
> We also think Gemma 2 is quite an interesting architecture to test, so we include a performance evaluation on Gemma 2 in Appendix C.4. In that section, we tested the effect of sliding window and HiP attention up to 128k context window using HiP and SelfExtend. Our method showed good performance recovery (Please check Table 2 below); sometimes, HiP even improves upon the original performance due to longer context being available on context-extended Gemma 2 (Please check Appendix Table 10).
>
> **Table 2. Gemma 2 Sliding Window Ablation.**
> Perplexity (Wikitext2) | Sliding? | 4k | 8k | 128k
> -|-|-|-|-
> FA2 | X | 12.35 | 30.17 | 1310.26
> FA2 | O | 12.35 | 11.40 | 363.02
> HiP (k=512) | X | 13.51 | 12.83 | 14.32
> HiP (k=512) | O | 13.11 | 12.25 | 11.54
>
> Moreover, we observe noticeable performance improvements with sliding window attentions in longer contexts. We think the sliding window layer helps stabilize the HiP-replaced layer. HiP specializes in retrieving long-range information, and the sliding window specializes in retrieving local information. Therefore, we think that there is some synergy effect in combination of two types of attention mechanisms.
>
> **Table 3. Updated Latency Results.**
> Prefill Latency (4090, ms) | 64k | 128k
> -|-|-
> FA2	| 219.430 | 893.360
> HiP (k=512) | 83.038 | 188.641
> HiP (k=1k)	| 155.583 | 364.060
> HiP (latest, low-preset) | 139.761 | 311.747
> HiP (latest, mid-preset) | 201.032 | 439.410
>
> **Table 4. InifniteBench with the latest HiP**
> `En.MC` Acc. | 8k | 16k	| 32k | 64k	| 128k | 256k
> -|-|-|-|-|-|-
> Gemma2 9B (latest, mid-preset)	| 0.4541	| **0.5240**	| **0.5852**	| 0.6026	| **0.6550**	| **0.6550**
> Llama3.1 8B (latest, mid-preset)	| **0.4585**	| 0.4978	| 0.5677	| **0.6114**	| 0.6114	| 0.6376
>
> `En.QA` F1. | 8k | 16k	| 32k | 64k	| 128k | 256k
> -|-|-|-|-|-|-
> Gemma2 9B (latest, mid-preset) 	| **0.1599**	| **0.1954**	| **0.2285**	| **0.2613**	| **0.2647**	| **0.2533**
> Llama3.1 8B (latest, mid-preset) | 0.1019	| 0.1171	| 0.1622	| 0.1797	| 0.2007	| 0.2009
>
> **Table 5. Passkey with the latest HiP**
> Passkey (Acc.)	| 8k | 16k | 32k | 64k | 128k | 256k | 448k | 704k | 960k
> -|-|-|-|-|-|-|-|-|-
> Gemma 2 9B (latest, mid-preset) | 100 | 100 | 100 | 100 | 100 | 100 | 90 | 100 | 100
>
> In our latest HiP research, after submission, we could engineer HiP further to enhance the ability to extend the context window while maintaining overall framework and latency, as shown in Table 3. We increased flexibility in our framework, so now the users freely control each iteration (how many chunks should be selected, how to break down each chunk, and how to select the representative tokens). We could increase long-context performance by searching appropriate iteration configurations. However, since the latest development is still under construction, we leave this for future work (of course, we will go to open source everything, so you can check our source code in real-time if you are wondering how we built it). With the latest version, we could extend the Gemma 2's context window from 8k to 960k on passkey evaluation (above Table 4).
>
> Therefore, we think a hierarchically pruned attention framework is a promising research direction and has great value to share with the research community to stimulate the following creative studies by community members who are trying to build extremely long context models using various approaches.
>
> > Q2. The method does show that MMLU is not degraded when using this method. It would be interesting to see this for a broader set of metrics if possible.
>
> Please specify a set of metrics you would like us to evaluate our method in, if possible. However, please note that we already have various sets of metrics that examine context utilization (Passkey, RULER), long-context reasoning performance (LongBench), and multi-modality functions of LMM (LMMS-eval). We hope our extensive large-scale empirical evaluation helps verify the performance of our work.
>
> ---
>
> Thank you once again for taking the time to review our work and provide insightful feedback.
>
> We hope to address and resolve your concerns during the discussion period. If our rebuttal successfully clarifies your questions, we kindly ask you to consider updating your score accordingly.

---

### Official Review · Reviewer_RsGC · 2024-11-08

**Soundness:** 3
**Presentation:** 3
**Contribution:** 3
**Rating:** 6
**Confidence:** 3

**Summary:**

This paper proposes Hierarchically Pruned Attention (HiP) to reduce the time complexity of attention to O(T logT) and space complexity to O(T) where T is the sequence length. By exploiting the continuity of token sequence (tokens close together tend to have similar scores), for each query, the HiP use a tree-search like algorithm to approximately search the top k key tokens that yield large attention weights. Further, the author developed a KV cache offloading scheme to offload KV cache to host memory and reduce the GPU memory usage.

**Strengths:**

1. The proposed method uses iterative refinement to dynamically and approximately locate top k tokens, which is interesting.
2. HiP shows promising efficiency improvement with only small performance drop.
3. The appendix provides a lot of ablation study to study the behavior of the proposed method.
4. The method is training free.

**Weaknesses:**

see question section

**Questions:**

1. It would be easier to understand to have a figure illustration showing the tree search (or improve figure 2, the figure 2 step 1 is a bit confusing) for section 3.1.
2. The algorithm divides the sequence into k segments, and one token will be selected in each segment. Is it correct? If so, then the top k tokens must be distributed in the sequence, and cannot be concentrated on certain regions. Why do you make this design choice?
3. I am aware of some literatures that also use iterative refinement to dynamically calculate attention for efficiency. Have the authors tried to compare to these literatures?

---

> ### Author Response · Authors · 2024-11-14
> **Author Rebuttal to the Reviewer RsGC**
>
> Thank you for taking the time to review our work and for providing insightful and supportive feedback. We have addressed your concerns and questions below.
>
> ---
>
> **Questions:**
> > Q1. It would be easier to understand to have a figure illustration showing the tree search (or improve figure 2, the figure 2 step 1 is a bit confusing) for section 3.1.
>
> Thank you for your advice. In this discussion period, we have added a detailed figure on tree search in the appendix (**please check Figure 19**: *Detailed Flowchart of Hierarchically Pruned Attention.*), referenced from Figure 2 in the main paper due to space limitations in the main paper. Please review the new figures and let us know if you have any further comments.
>
> > Q2. The algorithm divides the sequence into k segments, and one token will be selected in each segment. Is it correct? If so, then the top k tokens must be distributed in the sequence, and cannot be concentrated on certain regions. Why do you make this design choice?
>
> We would like to respectfully clarify a factual misunderstanding. We select top-k tokens from the whole sequence rather than select top-1 from each segment (chunk) individually. We believe this may be a similar misunderstanding to Q1. Our method performs each iteration as follows (*We omit block approximation here for clarity.*):
>
> 1. (Initialization step) Uniformly split the sequence of length $T$ into $k$ chunks.
> 2. Split each chunk in half, resulting in $2k$ chunks.
> 3. For each chunk, select a representative key token (the center of the chunk) and compute the dot product between the query and key token. This dot product score serves as the importance score for each chunk.
> 4. Select the top-$k$ most important chunks and return to step 2. If the chunk size is equal to or smaller than $b_k$, terminate the process.
>
> Since we globally select the top-$k$ chunks from the $2k$ chunks in each iteration (step 4), our method can concentrate the non-zero entries in specific, high-importance sections of the sequence. Please refer to the Appendix Figure 27 for the attention mask visualization on each iteration step.
>
> > Q3. I am aware of some literatures that also use iterative refinement to dynamically calculate attention for efficiency. Have the authors tried to compare to these literatures?
>
> Can you provide specific references to the related literature? We can try to compare them with our method.
>
> ---
>
> Thank you once again for taking the time to review our work and provide insightful feedback.
>
> We hope to address and resolve your concerns during the discussion period. If our rebuttal successfully clarifies your questions, we kindly ask you to consider updating your score accordingly.

---

### Author Response · Authors · 2024-11-25
**Reminder: Only two days are left before the Author-Reviewer discussion period ends**

Dear Reviewers,


This is a kind reminder that **we only have two days (about 50 hours) before the discussion period ends**.

**We are politely asking reviewers to review our rebuttal messages and leave responses.** We are still looking forward to having a discussion between authors and reviewers during the discussion period. We look forward to resolving existing concerns and comments that can improve our work with our rebuttal.

Again, we sincerely thank you for reviewing our work and providing insightful feedback.


Authors.

---

### Meta-Review · Area_Chair_u9t5 · 2024-12-25

**Metareview:**

Reviewers find the paper makes a solid contribution towards faster attention computation and solid results on models like Llama and Gemma. Main concerns were around limited models considered for experiments and the generality of the locality hypothesis. Authors presented more results during discussion answering the concerns. Overall I am happy to recommend acceptance.

**Additional Comments On Reviewer Discussion:**

Main concerns were around limited models considered and generality of the results. Authors presented additional experimental results answering these concerns.

---

### Decision · Program_Chairs · 2025-01-22

Accept (Poster)